# Towards Characterizing Domain Counterfactuals for Invertible Latent Causal Models

**Zeyu Zhou**\*, **Ruqi Bai**\*, **Sean Kulinski**\*, **Murat Kocaoglu, David I. Inouye**
Elmore Family School of Electrical and Computer Engineering
Purdue University
{zhou1059, bai116, skulinsk, mkocaoglu, dinouye}@purdue.edu

## ABSTRACT

Answering counterfactual queries has important applications such as explainability, robustness, and fairness but is challenging when the causal variables are unobserved and the observations are non-linear mixtures of these latent variables, such as pixels in images. One approach is to recover the latent Structural Causal Model (SCM), which may be infeasible in practice due to requiring strong assumptions, e.g., linearity of the causal mechanisms or perfect atomic interventions. Meanwhile, more practical ML-based approaches using naïve domain translation models to generate counterfactual samples lack theoretical grounding and may construct invalid counterfactuals. In this work, we strive to strike a balance between practicality and theoretical guarantees by analyzing a specific type of causal query called *domain counterfactuals*, which hypothesizes what a sample would have looked like if it had been generated in a different domain (or environment). We show that recovering the latent SCM is unnecessary for estimating domain counterfactuals, thereby sidestepping some of the theoretic challenges. By assuming invertibility and sparsity of intervention, we prove domain counterfactual estimation error can be bounded by a data fit term and intervention sparsity term. Building upon our theoretical results, we develop a theoretically grounded practical algorithm that simplifies the modeling process to generative model estimation under autoregressive and shared parameter constraints that enforce intervention sparsity. Finally, we show an improvement in counterfactual estimation over baseline methods through extensive simulated and image-based experiments.

## 1 INTRODUCTION

Causal reasoning and machine learning, two fields which historically evolved disconnected from each other, have recently started to merge with several recent results leveraging the available causal knowledge to develop better ML solutions (Kusner et al., 2017; Moraffah et al., 2020; Nemirovsky et al., 2022; Calderon et al., 2022). One such setting is causal representation learning (Schölkopf et al., 2021; Brehmer et al., 2022), which aims to take data from a complex observed space (e.g., images) and learn the *latent* causal factors that generate the data. A common scenario is when we have access to diverse datasets from different domains, where from a causal perspective, each domain is generated via an *unknown* intervention on some domain-specific latent causal mechanisms. With this in mind, we focus on a specific causal query called a *domain counterfactual* (DCF), which hypothesizes: "What would this sample look like if it had been generated in a different domain (or environment)?" For example, given a patient's medical imaging from Hospital A, what would it look like if it had been taken at Hospital B? Answering this DCF query could have applications in fairness, explainability, and model robustness.

A naïve ML approach to answering this query is to simply train generative models to map between the two distributions without any causal assumptions or causal constraints (e.g., Kulinski and Inouye (2023)); however, this lacks theoretic grounding and may produce invalid counterfactuals. One common causal approach for answering such a counterfactual query would be a two-stage method of first recovering the causal structure and then estimating the counterfactual examples (Kocaoglu

---

\*Equal contribution. Listing order is random.

Table 1: This table of related causal representation learning works, focuses mostly on works that study learning a *latent* SCM, shows that most prior works in this area aim for identifiability of the (latent) SCM, and thus require strong technical assumptions which may not hold in real-world scenarios (e.g., perfect single-node interventions for each variable).

| | SCM type | Observ. Function | Other Assumptions | Observ. Function Identifiability | Characterization of Counterfactual Equiv. |
|---|---|---|---|---|---|
| Nasr-Esfahany et al. (2023) | Invertible observed | N/A | 1) Access to ground-truth DAG | N/A | Single mechanism counterfactuals under specific contexts |
| Brehmer et al. (2022) | Invertible latent | Invertible | 1) Atomic stochastic hard interv 2) Training set is counterfactuals pairs | Mixing and elementwise | N/A - Counterfactuals as input |
| Squires et al. (2023) | Linear latent | Linear | 1) Atomic hard interv. | Scaling | No |
| Liu et al. (2022a) | Linear latent | Non-linear | 1) Significant causal weights variation | Mixing and scaling | No |
| Varici et al. (2023) | Latent non-linear | Linear | 1) Atomic stochastic hard interv. | Mixing or scaling | No |
| Khemakhem et al. (2021) | Invertible observed (implicit) | Affine | 1) Bivariate requirement for identifiability | Full (bivariate only) | No |
| Ours | Invertible latent | Invertible | 1) Access to domain labels | No | Domain counterfactual |

et al., 2018; Sauer and Geiger, 2021; Nemirovsky et al., 2022). However, most of the existing methods for causal structure learning either assume the causal variable to be observed (as opposed to our setting where the causal variables are latent) or require restrictive assumptions for recovering the latent causal structure, such as atomic interventions (Brehmer et al., 2022; Squires et al., 2023; Varici et al., 2023), or access to counterfactual pairs (Brehmer et al., 2022), or assume model structures like linearity or polynomial (Khemakhem et al., 2021; Squires et al., 2023), which often do not hold in practice. A summary of existing works can be found in Table 1. In this paper, we strive to balance practicality and theoretical guarantees by answering the question: "Can we theoretically and practically estimate domain counterfactuals without the need to recover the ground-truth causal structure?"

With weak assumptions about the true causal model and available data, we analyze invertible latent causal models and show that it is possible to estimate domain counterfactuals both theoretically and practically, where the estimation error depends on the intervention sparsity. We summarize our contributions as follows:

**C1** For a class of invertible latent domain causal models (ILD), we show that recovering the true ILD model is unnecessary for estimating domain counterfactuals by proving a necessary and sufficient characterization of domain counterfactual equivalence.

**C2** We prove a bound on the domain counterfactual estimation error which decomposes into a data fit term and intervention sparsity term. If the true intervention sparsity is small, this bound suggests adding a sparsity constraint for DCF estimation.

**C3** Towards practical implementation, we prove that *any* ILD model with intervention sparsity $k$ can be written in a *canonical* form where only the last $k$ variables are intervened. This significantly reduces the modeling search space from $\binom{m}{k}$ causal structures to only one.

**C4** In light of these theoretic results, we propose an algorithm for estimating domain counterfactuals by searching over canonical ILD models while restricting intervention sparsity (inspired by **C2** and **C3**). We validate our algorithm on both simulated and image-based experiments [1].

**Notation** We denote function equality between two functions $f : \mathcal{X} \to \mathcal{Y}$ and $f' : \mathcal{X} \to \mathcal{Y}$ as simply $f = f'$, which more formally can be stated as $\forall \boldsymbol{x} \in \mathcal{X}, f(\boldsymbol{x}) = f'(\boldsymbol{x})$. Similarly, $f \neq f'$ means that there exists $\boldsymbol{x} \in \mathcal{X}, f(\boldsymbol{x}) \neq f'(\boldsymbol{x})$. We use $\circ$ to denote function composition, e.g., $g(f(\boldsymbol{x})) = g \circ f(\boldsymbol{x})$ or simply $h = g \circ f$. We use subscripts to denote particular indices (e.g., $x_j \in \mathbb{R}$ is the $j$-th value of the vector $\boldsymbol{x}$ and $\boldsymbol{x}_{<j} \in \mathbb{R}^{j-1}$ is the subvector corresponding to the indices 1 to $j - 1$). For function outputs, we use bracket notation to select a single item (e.g., $[f(\boldsymbol{x})]_j \in \mathbb{R}$ refers to the $j$-th output of $f(\boldsymbol{x})$) or subvector (e.g., $[f(\boldsymbol{x})]_{\leq j} \in \mathbb{R}^j$ refers to the subvector for indices 1 to $j$ inclusive). Similarly, for (unbound) functions, let $[f]_j : \mathbb{R}^m \to \mathbb{R}$ refer to the scalar function corresponding to the $j$-th output or $[f]_{\leq j} : \mathbb{R}^m \to \mathbb{R}^j$ refer to the vector function corresponding to first $j$ outputs. For any positive integer $m$, we define $[m] \triangleq \{1, \ldots, m\}$. We denote $N_d$ as number of domains in the ILD model.

---

[1] Code can be found in https://github.com/inouye-lab/ild-domain-counterfactuals.

## 2 Domain Counterfactuals with Invertible Latent Domain Causal Models

Given a set of domains (or environments), a domain counterfactual (DCF) asks the question: "What would a sample from one domain look like if it had (counterfactually) been generated from a different domain?" Each domain represents different causal model on the same set of causal variables, i.e., they can be viewed as interventions of a baseline causal model. If we let $D$ be an auxiliary indicator variable denoting the domain, a DCF can be formalized as the counterfactual query $p(X_{D=d'}|X = \boldsymbol{x}, D = d)$, where $x$ is the observed evidence, $d$ is the original domain, and $X_{D=d'}$ is the counterfactual random variable when forcing the domain to be $d'$. In this work, we aim to find DCF for a class of invertible models (which we define in Section 2.1) and we will assume that the causal variables are unobserved (i.e., latent). To compare, Causal Representation Learning (CRL) has a similar latent causal model setup (Schölkopf et al., 2021). However, most CRL methods aim for identifiability of the latent representations, which is unsurprisingly very challenging. In contrast, we show that estimating DCFs is easier than estimating the latent causal representations and may require fewer assumptions in Section 2.2.

### 2.1 ILD Model

We now define the causal model based primarily on the assumption of invertibility. First, we assume that the observation function (or mixing function) shared between all domains is invertible (as in Liu et al. (2022a); Zhang et al. (2023); von Kügelgen et al. (2023)). This means that the latent causal variables are invertible functions of the observed variables. Second, we assume that the latent SCMs for each domain are also invertible with univariate exogenous noise terms per causal variable. We assume the standard Directed Acyclic Graph (DAG) constraint on the SCMs. For notational simplicity, we will assume w.l.o.g. that the DAG is a complete graph (i.e., it includes all possible edges), but some edges could represent a zero dependency which is functionally equivalent to the edge being missing. Given the topological ordering respecting the complete DAG, we prove that an invertible SCM can be written as a unique autoregressive invertible function that maps from all the exogenous noises to the latent endogenous causal variables (See Appendix B.1). Note that the SCM invertibility assumption excludes causal models where causal variables have multivariate exogenous noise. Given all this, we now define our ILD model class that joins together the shared mixing function and the latent SCMs for each domain.

**Definition 1** (Invertible Latent Domain Causal Model). *An invertible latent domain causal model (ILD), denoted by $(g, \mathcal{F})$, combines a shared invertible mixing function $g : \mathcal{Z} \to \mathcal{X}$ with a set of $N_d$ domain-specific latent SCMs $\mathcal{F} \triangleq \{f_d : \mathbb{R}^m \to \mathcal{Z}\}_{d=1}^{N_d}$, where $f_d$ are invertible and autoregressive. The exogenous noise is assumed to have a standard normal distribution, i.e., $\boldsymbol{\epsilon} \sim \mathcal{N}(0, I_m)$.*

While we discuss the model in depth in Appendix A, we first briefly discuss why the autoregressive and standard normal exogenous noise assumptions are not restrictive. For any model that violates the topological ordering, an equivalent ILD model can be constructed by merging the original mixing function with a variable permutation. Similarly, for any continuous exogenous distribution, we can construct an equivalent Gaussian noise-based ILD model via merging the original SCM with the Rosenblatt transform (Rosenblatt, 1952) and inverse element-wise normal CDF transformation. Moreover, we prove in the appendix that for any observed domain distributions, there exists an ILD model that could match these domain distributions. Therefore, these two assumptions are not critical but will simplify theoretical analysis.

Given our definition, we note that interventions between two ILDs are *implicitly* defined by the difference between two domain-specific causal models and the intervention set is denoted by $\mathcal{I}(f_d, f_{d'}) \subseteq [m]$, which is the set of the intervened causal variables' indices. In Appendix B.3 in the appendix, we prove that the standard notion of causal intervention is equivalent to checking if the inverse subfunctions are equal, i.e., $j \in \mathcal{I}(f_d, f_{d'}) \Leftrightarrow \left[f_d^{-1}\right]_j \neq \left[f_{d'}^{-1}\right]_j$. We further define the ILD intervention set as the union over all pairs of domains, i.e., $\mathcal{I}(\mathcal{F}) \triangleq \bigcup_{f_d, f_{d'} \in \mathcal{F}} \mathcal{I}(f_d, f_{d'}) = \bigcup_{d \leq N_d} \mathcal{I}(f_1, f_d)$. These implicit ILD interventions could be a hard intervention (i.e., remove dependence on parents) or a soft intervention (i.e., merely change the dependence structure with parents). Because any intervened causal mechanism is invertible by our definition, ILD interventions must be stochastic rather than do-style interventions, which would break the invertibility of the latent SCM.

Finally, we define a notion of two ILD models being equivalent with respect to their observed distributions based on the change of variables formula. This notion, which is a true equivalence relation because the equation in (2) has the properties of reflexivity, symmetry, and transitivity by the properties of the equality of measures, will be important for defining an upper bound on DCF estimation in Section 3.1 and for developing practical algorithms that minimize the divergence between the ILD observed distribution and the training data in Section 3.3.

**Definition 2** (Distribution Equivalence). *Two ILDs $(g, \mathcal{F})$ and $(g', \mathcal{F}')$ are distributionally equivalent, denoted by $(g, \mathcal{F}) \simeq_D (g', \mathcal{F}')$, if the induced domain distributions are equal, i.e.,*

$$\forall d, \quad p_{\mathcal{N}}\left(f_d^{-1} \circ g^{-1}(\boldsymbol{x})\right)|J_{f_d^{-1} \circ g^{-1}}(\boldsymbol{x})| = p_{\mathcal{N}}\left(f'^{-1}_d \circ g'^{-1}(\boldsymbol{x})\right)|J_{f'^{-1}_d \circ g'^{-1}}(\boldsymbol{x})|.$$

## 2.2 ILD DOMAIN COUNTERFACTUALS

With our ILD model defined, we now formalize a DCF query for our ILD model. For that, we remember the three steps for computing (domain) counterfactuals (Pearl, 2009, Chapter 1.4.4): abduction, action, and prediction. The first step is to infer the exogenous noise from the evidence. For ILD models, this simplifies to a deterministic function that inverts the mixing function and latent SCM, i.e., $\boldsymbol{\epsilon} = f_d^{-1} \circ g^{-1}(\boldsymbol{x})$. The second step and third steps are to perform the target intervention and run the exogenous noise through the intervened mechanisms. For ILD, this is simply applying the other domain's causal model and the shared mixing function, i.e., $\boldsymbol{x}_{d \to d'} = g \circ f_{d'}(\boldsymbol{\epsilon})$. Combining these steps yields the simple form of a DCF for ILD models: $\boldsymbol{x}_{d \to d'} \triangleq g \circ f_{d'} \circ f_d^{-1} \circ g^{-1}(\boldsymbol{x})$, where $f_d, f_{d'} \in \mathcal{F}$. DCF for ILD models are *deterministic counterfactuals* (de Lara et al., 2023) since they have a unique mapping, i.e., given the evidence $\boldsymbol{x}$ from $d$, the counterfactual $\boldsymbol{x}_{d \to d'}$ is deterministic. We now provide a notion that will define which ILDs have the same DCFs (see Appendix B.4 for the equivalence relation proof).

**Definition 3** (Domain Counterfactual Equivalence). *Two ILDs $(g, \mathcal{F})$ and $(g', \mathcal{F}')$ are domain counterfactually equivalent, denoted by $(g, \mathcal{F}) \simeq_C (g', \mathcal{F}')$, if all domain counterfactuals are equal, i.e., for all $d, d' : g \circ f_{d'} \circ f_d^{-1} \circ g^{-1} = g' \circ f'_{d'} \circ f'^{-1}_d \circ g'^{-1}$ .*

While Definition 3 succinctly defines the equivalence classes of ILDs, it does not give much insight into the structure of the equivalence classes. To fill this gap, we now present one of our main theoretic results which characterizes a *necessary and sufficient* condition for being domain counterfactually equivalent and relates proves that their intervention set size must be equal.

**Theorem 1** (Characterization of Counterfactual Equivalence). *Two ILDs are domain counterfactually equivalent, i.e., $(g, \mathcal{F}) \simeq_C (g', \mathcal{F}')$ if and only if:*

$$\exists h_1, h_2 \in \mathcal{F}_I \text{ s.t. } g' = g \circ h_1^{-1} \in \mathcal{F}_I \text{ and } f'_d = h_1 \circ f_d \circ h_2 \in \mathcal{F}_A, \forall d, \tag{1}$$

*and moreover, counterfactually equivalent models share the same intervention set size, i.e., if $(g, \mathcal{F}) \simeq_C (g', \mathcal{F}')$, then $|\mathcal{I}(\mathcal{F})| = |\mathcal{I}(\mathcal{F}')|$.*

See Appendix B.5 for proofs. Importantly, Theorem 1 can be used to *construct* domain counterfactually equivalent models and *verify* if two models are domain counterfactually equivalent (or determine they are not equivalent). In fact, for *any* two invertible functions $h_1$ and $h_2$ that satisfy the implicit autoregressive constraint, i.e., for all $d, h_1 \circ f_d \circ h_2 \in \mathcal{F}_A$, we can construct a counterfactually equivalent model—which can have arbitrarily different latent representations defined by $g' = g \circ h_1^{-1}$ since $h_1$ can be an arbitrary invertible function. Ultimately, this result implies that to estimate domain counterfactuals, we indeed *do not require the recovery of the latent representations or the full causal model*.

## 3 ESTIMATING ILD DOMAIN COUNTERFACTUALS IN PRACTICE

While the previous section proved that recovering the latent causal representations is not necessary for DCFs, this section seeks to design a practical method for estimating DCFs. Since we only assume access to i.i.d. data from each domain, one natural idea is to fit an ILD model that is distributionally equivalent to the observed domain distributions. Yet, distribution equivalence is only a distribution-level property while counterfactual equivalence is a point-wise property, i.e., the domain distributions can match while the counterfactuals could be different. Indeed, we show in Theorem 2

that even under the constraint of distribution equivalence, the counterfactual error can be very large. To mitigate this issue, we choose a relatively weak assumption called the Sparse Mechanism Shift (SMS) hypothesis (Schölkopf et al., 2021), which states that the differences between domain distributions are caused by a small number of intervened variables. Given this assumption about the true ILD model, it is natural to impose this intervention sparsity on the estimated ILD model. Therefore, we now have two components to ILD estimation: a distribution equivalence term and a sparsity constraint which are based on the dataset and our assumption respectively. We first prove that both of these components are important for DCF estimation by providing a bound on the counterfactual error (defined below). Then, we prove that the sparsity constraint can be enforced by only optimizing over a canonical version of ILD models, which have all intervened variables last in a topological ordering. This greatly simplifies the practical optimization algorithm since only one sparsity structure is needed than the potentially $\binom{m}{k}$ different sparsity structures, where $k$ is the sparsity level. Finally, we bring all of this together to form a practical optimization objective with sparsity constraints.

### 3.1 DOMAIN COUNTERFACTUAL ERROR BOUND

In this section, we will prove a bound on counterfactual error that depends on both distribution equivalence and intervention sparsity. Towards this end, let us first define a counterfactual pseudo-metric between ILD models via RMSE (proof of pseudo-metric in Lemma 6 in the appendix).

**Definition 4** (Counterfactual Pseudo-Metric for ILD Models). *Given a joint distribution $p(\boldsymbol{x}, d)$, the counterfactual pseudo metric between two ILDs $(g, \mathcal{F})$ and $(g', \mathcal{F}')$ is defined as the RMSE over all counterfactuals, i.e.,*

$$d_{\mathrm{C}}((g, \mathcal{F}), (g', \mathcal{F}')) \triangleq \sqrt{\mathbb{E}_{p(\boldsymbol{x}, d)p(d')}[\|g \circ f_{d'} \circ f_d{}^{-1} \circ g^{-1}(\boldsymbol{x}) - g' \circ f'_{d'} \circ f_d'^{-1} \circ g'^{-1}(\boldsymbol{x})\|_2^2]},$$

*where $p(d') = p(d)$ is the marginal distribution of the domain labels.*

Given this pseudo-metric, we can now derive a bound on the counterfactual error between an estimated ILD $(\hat{g}, \hat{\mathcal{F}})$ and the true ILD $(g^*, \mathcal{F}^*)$ defined as $\varepsilon(\hat{g}, \hat{\mathcal{F}}) \triangleq d_{\mathrm{C}}((\hat{g}, \hat{\mathcal{F}}), (g^*, \mathcal{F}^*))$.

**Theorem 2** (Counterfactual Error Bound Decomposition). *Given a max intervention sparsity $k \geq 0$ and letting $\mathcal{M}(k) \triangleq \{(g, \mathcal{F}) : (g, \mathcal{F}) \simeq_D (g^*, \mathcal{F}^*), |\mathcal{I}(\mathcal{F})| \leq \max\{k, |\mathcal{I}(\mathcal{F}^*)|\}\}$, the counterfactual error can be upper bounded as follows:*

$$\varepsilon(\hat{g}, \hat{\mathcal{F}}) \leq \underbrace{\min_{(g', \mathcal{F}') \in \mathcal{M}(k)} d_{\mathrm{C}}((\hat{g}, \hat{\mathcal{F}}), (g', \mathcal{F}'))}_{\text{(A) Error due to lack of distribution equivalence}} + \underbrace{\max_{(\tilde{g}, \widetilde{\mathcal{F}}) \in \mathcal{M}(k)} d_{\mathrm{C}}((\tilde{g}, \widetilde{\mathcal{F}}), (g^*, \mathcal{F}^*))}_{\text{(B) Worst-case error given distribution equivalence}} . \tag{2}$$

*Furthermore, if we assume that the ILD mixing functions are Lipschitz continuous, we can bound the worst-case error (B) as follows:*

$$(B) \leq \left[ \underbrace{\max_{(\tilde{g}, \widetilde{\mathcal{F}}) \in \mathcal{M}(k)} \widetilde{k} L_{\tilde{g}}^2 \max_{i \in [m]} \mathbb{E}\big[[\widetilde{f}_d(\boldsymbol{\epsilon}) - \widetilde{f}_{d'}(\boldsymbol{\epsilon})]_i^2\big]}_{\text{Error depends on } k \text{ since } \widetilde{k} \leq \max\{k, k^*\}} + \underbrace{k^* L_{g^*}^2 \max_{i \in [m]} \mathbb{E}\big[[f_d^*(\boldsymbol{\epsilon}) - f_{d'}^*(\boldsymbol{\epsilon})]_i^2\big]}_{\text{Error only depends on ground truth model}} \right]^{1/2},$$

*where $\widetilde{k} \equiv |\mathcal{I}(\widetilde{\mathcal{F}})|$ and $k^* \equiv |\mathcal{I}(\mathcal{F}^*)|$, $L_g$ is the Lipchitz constant of $g$, and the expectation is over $p(d, d', \boldsymbol{\epsilon}) \triangleq p(d)p(d')p(\boldsymbol{\epsilon})$.*

Please check proof in Appendix B.6. The first term (A) corresponds to a data fit term and could be reduced by minimizing the divergence between the ILD model and the observed distributions. If the estimated ILD already matches the ground truth distribution, then this term would be zero. The second term (B), however, does not involve the data distribution and cannot be explicitly reduced. Yet, the bound on this second error term shows that it can be implicitly controlled by constraining the target intervention sparsity $k$ of the estimated model. Informally, the (B) term depends on the intervention sparsity, Lipschitz constant, and a term that corresponds to the largest feature difference between domain SCMs. This last term can be interpreted as the worst case single-feature difference between *latent* counterfactuals. We do not claim this bound is tight, but rather simply aim to show that the domain counterfactual error depends on the target intervention sparsity $k$ such that reducing $k$ (as long as $k \geq k^*$) can improve DCF estimation. Therefore, our error bound elucidates that both data fit and intervention sparsity are needed for DCF estimation.

## 3.2 CANONICAL ILD MODEL

While the last section showed that imposing intervention sparsity helps control the counterfactual error, imposing this sparsity constraint can be challenging. In particular, the ground truth sparsity pattern, i.e., which of $k$ causal mechanisms are intervened, is unknown. A naïve solution would be to optimize an ILD model for all possible $\binom{m}{k}$ sparsity patterns. In this section, we prove that we only need to optimize one sparsity pattern without loss of generality. In particular, we can assume that all intervened mechanisms are on the last $k$ variables. We refer to such a model as a canonical ILD model which we formalize next.

**Definition 5** (Canonical Domain Counterfactual Model). *An ILD $(g, \mathcal{F})$ is a* canonical domain counterfactual model (canonical ILD), *denoted by $(g, \mathcal{F}) \in \mathcal{C}$, if and only if the last variables are intervened, i.e., $(g, \mathcal{F}) \in \mathcal{C} \Leftrightarrow \mathcal{I}(\mathcal{F}) = \{m - j : 0 \leq j < |\mathcal{I}(\mathcal{F})|\}$.*

While this definition may seem quite restrictive, we prove that (perhaps surprisingly) *any* ILD can be transformed to an equivalent *canonical* ILD.

**Theorem 3** (Existence of Equivalent Canonical ILD). *Given an ILD $(g, \mathcal{F})$, there exists a canonical ILD that is both counterfactually and distributionally equivalent to $(g, \mathcal{F})$ while maintaining the size of the intervention set, i.e., $\forall (g, \mathcal{F}), \exists (g', \mathcal{F}') \in \mathcal{C}$ s.t. $(g', \mathcal{F}') \simeq_{C,D} (g, \mathcal{F})$ and $|\mathcal{I}(\mathcal{F})| = |\mathcal{I}(\mathcal{F}')|$.*

See Appendix B.7 for full proof and Example 1 in the appendix for a toy example. This result is helpful for theoretic analysis and, more importantly, it has great practical significance as now we can merely search over canonical ILD models.

## 3.3 PROPOSED ILD ESTIMATION ALGORITHM

Given the error bound in Theorem 2, the natural approach is to minimize the divergence between the observed domain distributions (represented by the training data) and the model's induced distributions while constraining to $k$ interventions. From Theorem 3, we can simply optimize over canonical ILD models without loss of generality. Therefore, we optimize the following constrained objective given a target intervention size $k$:

$$\min_{g, \mathcal{F}} \mathbb{E}_{p(\boldsymbol{x}, d)}[-\log q_{g, \mathcal{F}}(\boldsymbol{x}, d)] \quad \text{s.t.} \quad [f_d]_{\leq m-k} = [f_{d'}]_{\leq m-k}, \forall d \neq d' . \tag{3}$$

Concretely, the practical algorithm means training a normalizing flow for each domain while sharing most (but not all) parameters and enforcing autoregressiveness for part of the model. The non-shared domain-specific parameters correspond to the intervened variable(s). For higher dimensional data, we also relax the strict invertibility constraint and implement this design using VAEs.

## 4 RELATED WORK

**Causal Representation Learning** Causal representation learning is a rapidly developing field that aims to discover the underlying causal mechanisms that drive observed patterns in data and learn representations of data that are causally informative (Schölkopf et al., 2021). This is in contrast to traditional representation learning, which does not consider the causal relationships between variables. As this is a highly difficult task, most works make assumptions on the problem structure, such as access to atomic interventions, the graph structure (e.g., pure children assumptions), or model structure (e.g., linearity) (Yang et al., 2022; Huang et al., 2022; Xie et al., 2022; Squires et al., 2023; Zhang et al., 2023; Sturma et al., 2023; Jiang and Aragam, 2023; Liu et al., 2022a). Other works such as (Brehmer et al., 2022; Ahuja et al., 2022; Von Kügelgen et al., 2021) assume a weakly-supervised setting where one can train on counterfactual pairs $(x, \tilde{x})$ during training. In our work, we aim to maximize the practicality of our assumptions while still maintaining our theoretical goal of equivalent domain counterfactuals (as seen in Table 1).

**Counterfactual Generation** A line of works focus on the identifiability of counterfactual queries (Shpitser and Pearl, 2008; Shah et al., 2022). For example, given knowledge of the ground-truth causal structure, Nasr-Esfahany et al. (2023) are able to recover the structural causal models up to equivalence. However, they do not consider the latent causal setting and assume some prior knowledge of underlying causal structures such as the backdoor criterion. There is a weaker form of

counterfactual generation without explicit causal reasoning but instead using generative models Zhu et al. (2017); Nemirovsky et al. (2022). These typically involve training a generative model with a meaningful latent representation that can be intervened on to guide a counterfactual generation (Ilse et al., 2020). As these works do not directly incorporate causal learning in their frameworks, we consider them out of scope for this paper. Another branch of works estimate causal effect without trying to learn the underlying causal structure, which typically assume all variables are observable(Louizos et al., 2017). An expanded related work section is in Appendix F.

## 5 EXPERIMENTS

We have shown theoretically the benefit of our canonical ILD characterization and restriction of intervention sparsity. In this section, we empirically test whether our theory could guide us to design better models for producing domain counterfactuals while only having access to observational data $x$ and the corresponding domain label $d$. In our simulated experiment, under the scenario where all of our modeling assumptions hold, we try to answer the following questions: (1) When we know the ground truth sparsity, does sparse canonical ILD lead to better domain counterfactual generation over naïve ML approaches (dense models)? (2) What would happen if there is a mismatch of sparsity between the dataset and modeling and what is a good model design strategy in practice? After this simulated experiment, we perform experiments on image datasets to determine if sparse canonical models are still advantageous in this more realistic setting. In this case, we assume the latent causal model lies in a lower dimensional space than the observed space and thus we use autoencoders to approximate an observation function that is invertible on a lower-dimensional manifold.

### 5.1 SIMULATED DATASET

**Experiment Setup** To extensively address our questions against diverse causal mechanism settings, for each experiment, we generate 10 distinct ground truth ILDs. The ground truth latent SCM $f_d^* \in \mathcal{F}_{IA}$ takes the form $f_d^*(\epsilon) = F_d^* \epsilon + b_d^* \mathbb{1}_{\mathcal{I}}$ where $F_d^* = (I - L_d^*)^{-1}, L_d^* \in \mathbb{R}^{m \times m}$ is a domain-specific lower triangular matrix that satisfies the sparsity constraint, $b_d^* \in \mathbb{R}$ is a domain-specific bias, $\mathbb{1}_{\mathcal{I}}$ is an indicator vector where entries corresponding to the intervention set are 1, and $L_d^*$ and $b_d^*$ are randomly generated for each experiment. The observation function takes the form $g^*(x) = G^*$ LeakyReLU $(x)$ where $G^* \in \mathbb{R}^{m \times m}$ and the slope of LeakyReLU is $0.5$. We use maximum likelihood estimation to train two ILDs (like training of a normalizing flow): *ILD-Can* as introduced in Section 3.2 and a baseline model, *ILD-Dense*, which has no sparsity restrictions on its latent SCM. To evaluate the models, we compute the mean square error between the estimated counterfactual and ground truth counterfactual. More details on datasets and models, and illustrating figures of the models can be found in Appendix C.1.

**Result** To answer whether sparse canonical ILD provides any benefit in domain counterfactual generation, we first look at the simplest case where the latent causal structure of the dataset and our model exactly match. In Figure 1a, we notice that when the grounth truth intervention set $\mathcal{I}^*$ is $\{5, 6\}$ (i.e. the last two nodes), *ILD-Can* significantly outperforms *ILD-Dense*. Then we create a few harder and more practical tasks where the intervention set size is still 2 but not constrained to the last few nodes. Again, in Figure 1a, we observe that no matter which two nodes are intervened on, *ILD-Can* performs much better than the naïve ML approach *ILD-Dense*. This first checks that restricting model structure to the specific canonical form does not harm the optimization even though the ground truth structure is different. Furthermore, it validates the benefit of our model design for domain counterfactual generation. More results with different number of domains and latent dimensions can be found in Appendix C.2, which all show that *ILD-Can* consistently perform better than *ILD-Dense*. We also include an illustrating figure visualizing how *ILD-Can* achieves lower counterfactual error. We then transition to the more practical scenario where the true sparsity $|\mathcal{I}^*|$ is unknown. In Figure 1b, at first glance, we observe a trend of the decrease in counterfactaul error as we decrease $|\mathcal{I}|$. For the case where $|\mathcal{I}| \geq |\mathcal{I}^*|$ (i.e. when $|\mathcal{I}| = 2, 3, 4$), this aligns with our intuition that the smaller search space of *ILD-Can* leads to a higher chance of finding model with low counterfactual error. For the case where $|\mathcal{I}| = 1$, we notice that it performs better than the canonical model that matches the true sparsity. Though it cannot reach distribution equivalence, the reduction in worst-case error (see Theorem 2) seems to be enough to enable comparable or better counterfactuals on average. We further check the performance of the data fitting and see a significant decrease in the fit of *ILD-Can* once $|\mathcal{I}| < |\mathcal{I}^*|$, which supports that the performance in data

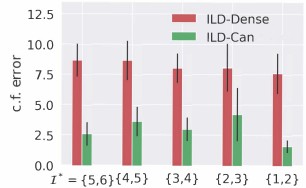 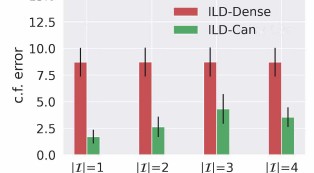

(a) With knowledge of $|\mathcal{I}^*|$ and $|\mathcal{I}^*| = |\mathcal{I}| = 2$.

(b) Without knowledge of $|\mathcal{I}^*|$ and $\mathcal{I}^* = \{5, 6\}$

Figure 1: Simulated experiment results ($N_d = 3$) averaged over 10 runs with different ground truth SCMs and the error bar represents the standard error. (a) This shows *ILD-Can* is consistently better than *ILD-Dense* regardless of intervened nodes in the dataset. (b) Here we test varying $|\mathcal{I}|$ while holding $\mathcal{I}^*$ fixed. The performance of *ILD-Can* approaches to that of *ILD-Dense* as we increase $|\mathcal{I}|$. An unexpected result is that *ILD-Can* performs best when $|\mathcal{I}| = 1$ and that results from a worse data fitting which is more carefully investigated in Appendix C.2.

fitting can be used as an indicator for whether we found the appropriate $|\mathcal{I}|$. Additional results on data fitting performance and experiments with different setups, including more complex $g$ based on normalizing flows and VAEs, can be found in Appendix C, and they all lead to the conclusion that *ILD-Can* produces better counterfactuals than *ILD-Dense* even though we do not know $|\mathcal{I}^*|$.

## 5.2 IMAGE-BASED COUNTERFACTUAL EXPERIMENTS

Here we seek to learn domain counterfactuals in the more realistic image regime. Following the manifold hypothesis (Gorban and Tyukin, 2018; Schölkopf et al., 2021), we assume that the causal interactions in this regime happen through lower-dimensional semantic latent factors as opposed to high-dimensional pixel-level interactions. To allow for learning of the lower dimensional latent space, we relax the invertibility constraint of our image-based ILD to only require pseudoinvertibility and test our models in this practical setting.

**High-dim ILD Modeling** We modify the ILD models from Section 5.1 to fit a VAE (Kingma and Welling, 2013) structure where the variational encoder, $(g^+, \mathcal{F}^+)$, first projects to the latent space via $g^+$ to produce the latent encoding $z$, which is then passed to two domain-specific latent causal models $f_{d,\mu}^+, f_{d,\sigma}^+$ which produce the parameters of posterior noise distribution. The decoder, $(g, \mathcal{F})$, follows the typical ILD structure: $g \circ f_d$, where, $g$ and $f_d$ can be viewed as pseudoinverse of $f_{d,\mu}^+$ and $g^+$. A detailed description and diagram of the models can be found in Figure 19, but informally, these modified ILD models can be seen as training a VAE *per* domain with the restriction that each VAE shares parameters for its initial encoder and final decoder layers (i.e. $g$ is shared). As an additional baseline, we compare against the naïve setup, which we call *ILD-Independent*, where each VAE has no shared parameters (i.e. a separate $g$ is learned for each domain). These models were trained using the $\beta$-VAE framework (Higgins et al., 2017). Further details can be found in the Appendix D.4. After training, we can perform domain counterfactuals as described in Section 2.2.

**Dataset** We apply our methods to five image-based datasets: Rotated MNIST (RMNIST), Rotated FashionMNIST (RFMNIST)(Xiao et al., 2017), Colored Rotated MNIST (CRMNIST), 3D Shapes (Burgess and Kim, 2018) and Causal3DIdent (Von Kügelgen et al., 2021), which all have both domain information (e.g., the rotation of the MNIST digit) and class information (e.g., the digit number). For each dataset, we split the data into disjoint domains (e.g., each rotation in CRMNIST constitutes a different domain) and define class variables which are generated independently of domains (e.g., digit class in CRMNIST), to evaluate our model's capability of generating domain counterfactuals. Specifically, for RMNIST, RFMNIST and 3D Shapes, all latent variables are independently generated, and for CRMNIST and Causal3DIdent, there is a more complicated causal graph containing the domain, class and other latent variables. Further details on each dataset and (assumed) ground-truth latent causal graphs could be found in Appendix D.1 and Appendix D.3.

**Metrics** Inspired by the work in Monteiro et al. (2023), we evaluate the image-based counterfactuals with latent SCMs via the following metrics, where $h_{\text{domain}}$ and $h_{\text{class}}$ represents pretrained domain classifier and class classifier respectively: (1) *Effectiveness* - whether the counterfactual truly changes the domain defined as $\mathbb{P}(h_{\text{domain}}(\hat{x}_{d \to d'}) = d')$; (2) *Preservation* - whether the domain counterfactual *only* changes domain-specific information defined as $\mathbb{P}(h_{\text{class}}(\hat{x}_{d \to d'}) = y)$; (3) *Composition* - whether the counterfactual model is invertible defined as $\mathbb{P}(h_{\text{class}}(\hat{x}_{d \to d}) = y)$; and (4)

Table 2: Quantitative result for **Composition** (Comp.), **Reversibility** (Rev.), **Preservation** (Pre.), and **Effectiveness** (Eff.), where higher is better. CRMNIST, 3D Shapes, Causal3DIdent are averaged 20, 5, 10 runs respectively. Best models are bold (within 1 standard deviation) and due to space constraints, expanded tables with additional datasets and standard deviation are in Appendix D.5.

| | CRMNIST | | | | 3D Shapes | | | | Causal3DIdent | | | |
|---|---|---|---|---|---|---|---|---|---|---|---|---|
| | Comp. | Rev. | Eff. | Pre. | Comp. | Rev. | Eff. | Pre. | Comp. | Rev. | Eff. | Pre. |
| *ILD-Independent* | **87.24** | 59.88 | **94.65** | 60.39 | **99.79** | 32.56 | **94.97** | 32.49 | **88.15** | 51.43 | **91.05** | 51.94 |
| *ILD-Dense* | **88.18** | 62.29 | **92.72** | 59.60 | **99.76** | 32.60 | 80.92 | 32.64 | 83.59 | 49.17 | **92.17** | 48.83 |
| *ILD-Can* | **92.10** | **85.74** | **94.48** | **72.95** | **99.85** | **79.84** | **96.72** | **64.99** | 86.00 | **79.73** | 84.15 | **79.73** |

(a) 3D Shapes

(b) CausalIdent

Figure 2: Domain counterfactuals with 3D Shapes and CausalIdent. Expanded figures can be found in Appendix D.5 (a) For 3D Shapes, only the object shape should change with domain counterfactuals – the other latent factors such as the hue of object, floor, background, should not change. (b) For CausalIdent, as the domain changes, the color of the background should change while holding all else unchanged. *ILD-Can* clearly performs better than the baseline *ILD-Dense* in terms of preserving non-domain features while changing domains for all datasets.

*Reversibility* - whether the counterfactual model is cycle-consistent defined as $\mathbb{P}(h_{\text{class}}(\hat{x}_{d \to d' \to d}) = y)$. For example, in the case of CRMNIST, a model might be able to rotate the image but cannot preserve the digit class during rotation, which would be high in effectiveness but low in preservation score. Details on the computation of these metrics and causal interpretations can be found in Appendix D.2 and Appendix D.3 respectively.

**Result** Due to space constraints, we put all results with RMNIST and RFMNIST in Appendix D.5. In Figure 2 we can see examples of domain counterfactuals for both *ILD-Dense* and *ILD-Can*. We note that no latent information other than the domain label was seen during training, thus suggesting the intervention sparsity is what allowed the canonical models to preserve important non-domain-specific information such as class information when generating domain counterfactuals. In Table 2, we include quantitative results using our metrics, which shows *ILD-Can* having significantly better reversibility and preservation while maintaining similar levels of counterfactual effectiveness and composition than the non-sparse counterparts. In Appendix D.5, we further investigate our model's sensitivity to the choice of sparsity by tracking how each metric change w.r.t. $|\mathcal{I}|$. We observe that reversibility and preservation tends to decrease while effectiveness tends to increase as we increase $|\mathcal{I}|$, which aligns with our findings here as *ILD-Dense* is equivalent to making $\mathcal{I}$ contain all latent nodes. In summary, our results here indicate our theory-inspired model design leads to better domain counterfactual generation in the practical pseudo-invertible setting.

## 6 CONCLUSION

In this paper, we show that estimating domain counterfactuals given only i.i.d. data from each domain is feasible without recovering the latent causal structure. We theoretically analyzed the DCF problem for a particular invertible causal model class and proved a bound on estimation error that depends on both a data fit term and an intervention sparsity term. Inspired by these results, we implemented a practical likelihood-based algorithm under intervention sparsity constraints that demonstrated better DCF estimation than baselines across experimental conditions. We discuss the limitations of our methods in Appendix E. We hope our findings can inspire simpler causal queries that are useful yet practically feasible to estimate and begin bridging the gap between causality and machine learning.

ACKNOWLEDGEMENT

Z.Z., R.B., S.K., and D.I. acknowledge support from NSF (IIS-2212097), ARL (W911NF-2020-221), and ONR (N00014-23-C-1016). M.K. acknowledges support from NSF CAREER 2239375.

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

# Appendix

## Table of Contents

## A  DISCUSSION OF INVERTIBLE LATENT DOMAIN CAUSAL MODEL

This section gives further discussion and details about our ILD model and serves as an expanded version of Section 2. We first remind the reader of the definition of an SCM using our notation.

**Definition 6** (Structural Causal Model). *A structural causal model (SCM) considers $m$ endogenous (causal) variables $z_j$ and $m$ exogenous noises $\epsilon_j$, $j \in [m]$, where each variable is a deterministic function of its parents and independent exogenous noise. Formally, each endogenous variable has form $z_j \triangleq f(\epsilon_j, z_{Pa_j})$, for all $j \in [m]$.*

Note that the SCM is a set of equations for each endogenous variables, where the exogenous noises could be multivariate and even infinite dimensional.

### A.1  INVERTIBLE SCM AS A GLOBAL INVERTIBLE AUTOREGRESSIVE FUNCTION

For theoretic analysis, our main SCM assumption is that the exogenous variables can be uniquely recovered from the endogenous variables, i.e., the SCM is invertible. This invertibility assumption will mean that domain counterfactuals are unique rather than being distributions over possible counterfactuals.

**Definition 7** (Invertible SCM). *We say that an SCM is invertible if the exogenous noise values can be uniquely recovered from the endogenous random variables, i.e., there is a one-to-one mapping between exogenous variables and endogenous variables.*

This invertibility assumption implies that the exogenous noises must be scalars[2] unlike standard SCMs, which can have multivariate exogenous noise variables.

We will now prove that all the SCM mechanisms can be represented by an vector to vector invertible autoregressive function (up to a relabeling), denoted as $f \in \mathcal{F}_{IA}$, and we call this the SCM *global function*. We first define an autoregressive function below.

**Definition 8** (Autoregressive Function). *A function $f : \mathbb{R}^m \to \mathbb{R}^m$ is autoregressive, denoted by $f \in \mathcal{F}_A$, if for all $i$, the $i$-th output can be written as a function of its corresponding input predecessors, i.e.,*

$$f \in \mathcal{F}_A \Leftrightarrow \forall j, \exists f^{(j)} \ \ s.t. \ \ [f(\epsilon)]_j \triangleq f^{(j)}(\epsilon_{\leq j}), \ where \ \epsilon \in \mathbb{R}^m. \tag{4}$$

Given this definition, we can now state our proposition that an invertible SCM can be represented by a single global invertible autoregressive function.

**Proposition 1** (SCM Global Function Representation). *An invertible SCM $\{\widetilde{f}^{(j)}\}_{j=1}^m$ that is topologically ordered, i.e., the parents have smaller index than the children, can be uniquely represented by an autoregressive invertible function $f \in \mathcal{F}_{IA}$ and vice versa.*

See Appendix B.1 for proof. From here on, we will simply use $f \in \mathcal{F}_{IA}$ to represent a invertible SCM.

While invertible SCMs do not subsume generic SCMs, we note that for any observed distribution of endogenous variables, there exist an invertible SCM that matches the observed distribution formalized as follows.

**Proposition 2** (Existence of Invertible SCM for Any Distribution). *Given any observed continuous distribution, there exists an invertible SCM with continuous exogenous noise whose observed distribution matches the given observed distribution.*

The full proof is in Appendix B.2. This means that invertible SCMs can model any continuous distribution, but they are not as general as generic SCMs. In practice, invertibility can be relaxed using pseudo-invertible or approximately invertible functions, as seen with a VAE in Section 5.2. We assume the exogenous noise distribution is standard Gaussian, which is made mostly for convenience and can be made without loss of generality due to the invertible Rosenblatt transformation (Rosenblatt, 1952; Melchers and Beck, 2018, Chapter B).

## A.2    Interventions for Invertible SCMs

If two SCMs are defined on the same space, then they could be regarded as soft interventions of each other, i.e., one SCM can be viewed as the observational SCM and the other as the intervened SCM, or vice versa. Thus, using the standard notion of intervention in which the causal mechanism is different, we can define the intervention set between two invertible SCMs in a symmetric way.

**Definition 9** (Intervention Set). *The intervention set between $f, f' \in \mathcal{F}_{IA}$ defined on the same sample space is the indices of the intervened variables of the corresponding unique SCMs derived from Proposition 1 represented by the equivalent individual SCM mechanisms $\widetilde{f}^{(j)}$ and $\widetilde{f'}^{(j)}$ respectively, i.e.,*

$$\mathcal{I}(f, f') \triangleq \mathcal{I}\big(\{\widetilde{f}^{(j)}\}_{j=1}^m, \{\widetilde{f'}^{(j)}\}_{j=1}^m\big) \triangleq \big\{ j : \widetilde{f}^{(j)} \neq \widetilde{f'}^{(j)} \big\}. \tag{5}$$

In the following, we show how to determine the intervention set using the SCM global functions $f$ and $f'$ directly instead of having to convert to the corresponding individual SCM mechanisms as in

---

[2]The proof is simple by contradiction. Suppose one mechanism had non-scalar exogenous noise. If the random variables are not perfectly dependent, then it would be impossible to recover more than one exogenous noise variable from a single endogenous noise variable and the parents. If the random variables were deterministic functions of each other (i.e., perfectly dependent), then the exogenous noise variables could be collapsed into a single exogenous noise without loss of generality.

the definition above. This will aid in the theoretic analysis and simplifies the analysis of intervention sparsity. See Appendix B.3 for proof.

**Proposition 3** (Intervention set characterized by SCM global function). *The intervention set between two SCM $f, f' \in \mathcal{F}_{IA}$ is equivalent to the set of variables where the inverse sub functions are different, i.e., $\mathcal{I}(f, f') = \left\{ j : \left[ f^{-1} \right]_j \neq \left[ f'^{-1} \right]_j \right\}$.*

## A.3 INVERTIBLE LATENT DOMAIN CAUSAL MODEL (ILD)

In this section, we propose the invertible latent domain causal model to capture multiple latent SCMs that are emerged through intervention. The data generated by one SCM forms a *domain*.

**Definition 1** (Invertible Latent Domain Causal Model). *An invertible latent domain causal model (ILD), denoted by $(g, \mathcal{F})$, combines a shared invertible mixing function $g : \mathcal{Z} \to \mathcal{X}$ with a set of $N_d$ domain-specific latent SCMs $\mathcal{F} \triangleq \{ f_d : \mathbb{R}^m \to \mathcal{Z} \}_{d=1}^{N_d}$, where $f_d$ are invertible and autoregressive. The exogenous noise is assumed to have a standard normal distribution, i.e., $\epsilon \sim \mathcal{N}(0, I_m)$.*

ILD induces the following data generating process: for the $d$-th domain, $\boldsymbol{z} = f_d(\epsilon)$ and $\boldsymbol{x} = g(\boldsymbol{z})$. Because $f_d$ and $g$ are invertible, we can write the observed distribution using the change of variables formula as: $p_d(\boldsymbol{x}) = p_{\mathcal{N}} \left( f_d^{-1} \circ g^{-1}(\boldsymbol{x}) \right) |J_{f_d^{-1} \circ g^{-1}}(\boldsymbol{x})|$. We now note that assuming a topological ordering of the latent variables does not restrict the ILD model class.

**Remark 1.** *The latent SCMs in ILD can be assumed to be topologically ordered without loss of generality.*

Because the latent variables are all unobserved, the labeling is arbitrary, thus we could relabel them in a way that preserves topological order and add a permutation to the observation function $g$. Essentially, given a non-autoregressive ILD, we could convert to an equivalent autoregressive ILD.

We now remember the distribution equivalence between two ILDs. The distributional equivalence defines a true equivalence relation because the equation in (2) has the properties of reflexivity, symmetry, and transitivity by the properties of the equality of measure.

**Definition 2** (Distribution Equivalence). *Two ILDs $(g, \mathcal{F})$ and $(g', \mathcal{F}')$ are distributionally equivalent, denoted by $(g, \mathcal{F}) \simeq_D (g', \mathcal{F}')$, if the induced domain distributions are equal, i.e.,*

$$\forall d, \quad p_{\mathcal{N}} \left( f_d^{-1} \circ g^{-1}(\boldsymbol{x}) \right) |J_{f_d^{-1} \circ g^{-1}}(\boldsymbol{x})| = p_{\mathcal{N}} \left( f'^{-1}_d \circ g'^{-1}(\boldsymbol{x}) \right) |J_{f'^{-1}_d \circ g'^{-1}}(\boldsymbol{x})|.$$

## B PROOFS AND AUXILIARY RESULTS

In this section, we prove propositions in Appendix A about the ILD model and the other results in the main paper. Before proving Proposition 2, we first introduce another lemma that is useful later in proving Proposition 1.

**Lemma 1** (Invertible Upper Subfunctions). *The upper subfunctions of $f \in \mathcal{F}_I \cap \mathcal{F}_A$ are also invertible, i.e., $\bar{f}_j(\epsilon_{\leq j}) \triangleq [f(\epsilon_{\leq j}, \cdot)]_{\leq j}$ is an invertible function of $\epsilon_{\leq j}$.*

*Proof.* We will prove this by induction on $k$ where $j = m - k$. For $k = 0$, it is trivial because $\bar{f}_{\leq m} \equiv f \in \mathcal{F}_I$. We will prove the inductive step by contradiction. Suppose $\bar{f}_{\leq m-k}$ is not invertible. This would mean it is not injective and/or not surjective.

If $\bar{f}_j$ is not injective, then $\exists \epsilon_{\leq j} \neq \epsilon'_{\leq j}$ such that $\bar{f}_{\leq j}(\epsilon_{\leq j}) = \bar{f}_{\leq j}(\epsilon'_{\leq j})$. We would then have for some $\epsilon_{>j}$ (e.g., all zeros):

$$\begin{aligned}
&\bar{f}_{\leq j+1}(\epsilon_{\leq j}, \epsilon_{j+1}) \\
&= [\bar{f}_{\leq j}(\epsilon_{\leq j}), [f(\epsilon_{\leq j}, \epsilon_{>j})]_{j+1}]^{\top} \\
&= [\bar{f}_{\leq j}(\epsilon'_{\leq j}), [f(\epsilon_{\leq j}, \epsilon_{>j})]_{j+1}]^{\top} \\
&= \bar{f}_{\leq j+1}(\epsilon'_{\leq j}, \epsilon_{j+1}),
\end{aligned} \tag{6}$$

but this would contradict the fact that $\bar{f}_{\leq j+1}$ is invertible by the inductive hypothesis.

If $\bar{f}_{\leq j}$ is not surjective, then $\exists\, \boldsymbol{x}_{\leq j}$ such that $\forall \epsilon_{\leq j}, \bar{f}_{\leq j}(\epsilon_{\leq j}) \neq \boldsymbol{x}_{\leq j}$. We would then have that $\forall \epsilon_{\leq j}, \epsilon_{>j}$

$$\bar{f}_{j+1}(\epsilon_{\leq j}, \epsilon_{j+1}) = [\bar{f}_j(\epsilon_{\leq j}), [f(\epsilon_{>j})]_{j+1}]^\top \neq [\boldsymbol{x}_{\leq j}, x_{j+1}]^\top. \tag{7}$$

but this would contradict the fact inductive hypothesis that $\bar{f}_{j+1}$ is surjective. Therefore, $\bar{f}_j$ must be invertible for all $j \in [m]$. $\qquad\square$

### B.1    PROOF OF PROPOSITION 1

**Proposition 1** (SCM Global Function Representation). *An invertible SCM $\{\widetilde{f}^{(j)}\}_{j=1}^m$ that is topologically ordered, i.e., the parents have smaller index than the children, can be uniquely represented by an autoregressive invertible function $f \in \mathcal{F}_{IA}$ and vice versa.*

We first note that because of topologically ordering, we can write the causal mechanisms $\{\widetilde{f}^{(j)}\}_{j=1}^m$ using different notation w.l.o.g. as: $z_j = \widehat{f}^{(j)}(\epsilon_j, \boldsymbol{z}_{<j}) \triangleq \widetilde{f}^{(j)}(\epsilon_j, \boldsymbol{z}_{z_{\mathrm{Pa}_j}})$ with $\widehat{f}^{(j)}(\cdot, \cdot) :$ $\mathbb{R} \times \mathbb{R}^{j-1} \to \mathbb{R}$. The topological ordering ensures that the parents are earlier indices, i.e., $\mathrm{Pa}_j \subseteq \{1, 2, \cdots, j-1\}$, so that this rewriting is possible w.l.o.g. Given this new notation, the unique representation is given by:

$$f(\epsilon) = \left[ \widehat{f}^{(1)}(\epsilon_1),\; \widehat{f}^{(2)}(\epsilon_2, \underbrace{\widehat{f}^{(1)}(\epsilon_1)}_{\text{recover } z_1}),\; \widehat{f}^{(3)}(\epsilon_3, \underbrace{\widehat{f}^{(1)}(\epsilon_1), \widetilde{f}^{(2)}(\epsilon_2, \widehat{f}^{(1)}(\epsilon_1))}_{\text{recover } \boldsymbol{z}_{<3}}),\; \cdots \right]^\top, \tag{8}$$

where for all $j$,

$$\widehat{f}^{(j)}(\epsilon_j, \boldsymbol{z}_{<j}) = [f(\underbrace{[f^{-1}(\boldsymbol{z}_{<j}, \cdot)]_{<j}}_{\text{recover } \epsilon_{<j} \text{ from } \boldsymbol{z}_{<j}}, \epsilon_j, \cdot)]_j. \tag{9}$$

*Proof.* We first prove one direction. Given an invertible SCM defined by it's causal mechanisms $\{\widehat{f}^{(j)}(\epsilon_j, \boldsymbol{z}_{<j})\}_{j=1}^m$, the observed variables are given recursively as:

$$z_j = \widehat{f}^{(j)}(\epsilon_j, \boldsymbol{z}_{<j}). \tag{10}$$

We now define the corresponding $f$ as in the lemma:

$$f(\epsilon) \triangleq \left[ \widehat{f}^{(1)}(\epsilon_1),\; \widehat{f}^{(2)}(\epsilon_2, \underbrace{\widehat{f}^{(1)}(\epsilon_1)}_{\text{recover } z_1}),\; \widehat{f}^{(3)}(\epsilon_3, \underbrace{\widehat{f}^{(1)}(\epsilon_1), \widetilde{f}^{(2)}(\epsilon_2, \widehat{f}^{(1)}(\epsilon_1))}_{\text{recover } \boldsymbol{z}_{<3}}),\; \cdots \right]^\top. \tag{11}$$

We need to prove that the observed variables are equivalent to the given SCM. Formally, we will prove by induction on $j \in [m]$ the hypothesis that $[f(\epsilon)]_j = \widehat{f}^{(j)}(\epsilon_j, \boldsymbol{z}_{<j}) = z_j, \forall \epsilon \in \mathbb{R}^m$. The base case is trivial from the definition in (11), i.e., $\forall \epsilon \in \mathbb{R}^m$, $[f(\epsilon)]_j = \widehat{f}^{(1)}(\epsilon_1) = z_j$. For the inductive step, we have the following:

$$[f(\epsilon)]_{j+1} = \widehat{f}^{(j+1)}(\epsilon_{j+1}, \underbrace{\widehat{f}^{(1)}(\epsilon_1)}_{z_1}, \underbrace{\widehat{f}^{(2)}(\epsilon_2, \widehat{f}^{(1)}(\epsilon_1))}_{z_2}, \cdots) = \widehat{f}^{(j+1)}(\epsilon_{j+1}, \boldsymbol{z}_{<j+1}) = z_{j+1} \tag{12}$$

where the first equals is by (11), the second is by the inductive hypothesis, and the last is by definition of the SCM.

Now we prove the other direction. Given an invertible autoregressive function $f \in \mathcal{F}_I \cap \mathcal{F}_A$, we define the following recursive set of mechanism functions:

$$\forall j, z_j \equiv \widehat{f}^{(j)}(\epsilon_j, \boldsymbol{z}_{<j}) \triangleq [f([f^{-1}(\boldsymbol{z}_{<j}, \cdot)]_{<j}, \epsilon_j, \cdot)]_j. \tag{13}$$

Again, we will prove that these functional forms are equivalent via induction on $j$ for the hypothesis that $\widehat{f}^{(j)}(\epsilon_j, \boldsymbol{z}_{<j}) = [f(\epsilon)]_j = z_j$. The base case is trivial based on (13):

$$\widehat{f}^{(1)}(\epsilon_1) = [f([f^{-1}(\boldsymbol{z}_{<1}, \cdot)]_{<1}, \epsilon_1, \cdot)]_1 = [f(\epsilon_1, \cdot)]_1 = z_1 \tag{14}$$

For the inductive step, we use the definition of $\bar{f}_{<j}$ and its inverse from Lemma 1 and derive the final result:

$$\widehat{f}^{(j+1)}(\epsilon_{j+1}, \boldsymbol{z}_{<j+1}) = [f([f^{-1}(\boldsymbol{z}_{<j}, \cdot)]_{<j}, \epsilon_j, \cdot)]_j = [f(\bar{f}_{<j}^{-1}(\boldsymbol{z}_{<j}), \epsilon_j, \cdot)]_j = [f(\epsilon_{<j}, \epsilon_j, \cdot)]_j = z_j. \tag{15}$$

$\qquad\square$

### B.2 PROOF OF PROPOSITION 2

**Proposition 2** (Existence of Invertible SCM for Any Distribution). *Given any observed continuous distribution, there exists an invertible SCM with continuous exogenous noise whose observed distribution matches the given observed distribution.*

The proof leverages the invertible Rosenblatt transformation (Rosenblatt, 1952; Melchers and Beck, 2018, Chapter B) that can transform any distribution to the uniform distribution or vice versa using its inverse. Given an ordering of a set of random variables, i.e., $\mathbf{X} = [X_1, X_2, \cdots, X_m]^\top$, the Rosenblatt transformation is defined as follows:

$$
\begin{aligned}
u_1 &:= F_1(x_1) \\
u_2 &:= F_2(x_2|x_1) \\
u_3 &:= F_3(x_3|x_1, x_2) \\
&\vdots \\
u_m &:= F_m(x_m|x_1, x_2, \cdots, x_{m-1}),
\end{aligned}
\tag{16}
$$

where $F_j(x_j|\boldsymbol{x}_{<j})$ is the conditional CDF of $X_j$ given $\mathbf{X}_{<j} = \boldsymbol{x}_{<j}$, i.e., the CDF corresponding to the distribution $p(X_j = x_j|\mathbf{X}_{<j} = \boldsymbol{x}_{<j})$. It's inverse can be written as follows:

$$
\begin{aligned}
x_1 &= F_1^{-1}(u_1) \\
x_2 &= F_2^{-1}(u_2|F_1^{-1}(u_1)) \\
x_3 &= F_3^{-1}(u_3|F_1^{-1}(u_1), F_2^{-1}(u_2|x_1 = F_1^{-1}(u_1))) \\
&\vdots \\
x_m &= F_m^{-1}(u_m|x_1 = F_1^{-1}(u_1)), x_2 = F_2^{-1}(u_2|x_1 = F_1^{-1}(u_1)), \cdots, x_{m-1} = \ldots),
\end{aligned}
\tag{17}
$$

where $F_j^{-1}(u_j|\boldsymbol{x}_{<j})$ is the conditional inverse CDF corresponding to the conditional CDF $F_j(x_j|\boldsymbol{x}_{<j})$.

Let $F_p(\boldsymbol{x})$ denote the Rosenblatt transformation for distribution $p$, and let $F_p^{-1}(\boldsymbol{u})$ denote its inverse as defined above. Assuming the random variables are continuous, the Rosenblatt transformation transforms the samples from any distribution to samples from the Uniform distribution (i.e., the push-forward of the Rosenblatt transformation is the uniform distribution and the push-forward of a uniform distribution through the inverse Rosenblatt is the distribution $p$).

*Proof.* Given any continuous target distribution $p$, we can construct an invertible SCM whose observed distribution is $p$. Specifically, if we let $q$ denote the exogenous noise distribution, then the following invertible and autoregressive function $f$—which defines an invertible SCM via Proposition 1—can be used to match the SCM distribution to $p$:

$$
f(\epsilon) = F_p \circ F_q^{-1}(\epsilon),
\tag{18}
$$

where $F_q^{-1}$ maps to the uniform distribution and then $F_p$ maps to the target distribution per the properties of the Rosenblatt transformation. The function is invertible since both functions are invertible. Additionally, both functions are autoregressive and thus the composition is autoregressive. Therefore, $f$ represents a valid invertible SCM whose observed distribution is $p$. □

### B.3 PROOF OF PROPOSITION 3

**Proposition 3** (Intervention set characterized by SCM global function). *The intervention set between two SCM $f, f' \in \mathcal{F}_{IA}$ is equivalent to the set of variables where the inverse sub functions are different, i.e., $\mathcal{I}(f, f') = \left\{ j : \left[f^{-1}\right]_j \neq \left[f'^{-1}\right]_j \right\}$.*

*Proof.* **Step 1:** Prove $\left\{ j : \left[f^{-1}\right]_j \neq \left[f'^{-1}\right]_j \right\} \subseteq \mathcal{I}\left(\tilde{f}, \tilde{f}'\right)$.

For all $j \in \left\{ j : \left[ f^{-1} \right]_j \neq \left[ f'^{-1} \right]_j \right\}$, there exists some $\boldsymbol{z}$, such that

$$\left[ f^{-1}(\boldsymbol{z}) \right]_j \neq \left[ f'^{-1}(\boldsymbol{z}) \right]_j, \tag{19}$$

given that $f, f'$ are auto-regressive function, we conclude there exists some $(\boldsymbol{z}_{<j}, z_j)$ such that

$$\epsilon_j = [f^{-1}(\boldsymbol{z}_{<j}, z_j, \cdot)]_j \neq [f'^{-1}(\boldsymbol{z}_{<j}, z_j, \cdot)]_j = \epsilon'_j. \tag{20}$$

we have, for $\epsilon_j, \epsilon'_j$ and such $\boldsymbol{z}_{<j}$ there holds

$$
\begin{aligned}
\widehat{f}^{(j)}(\epsilon_j, \boldsymbol{z}_{<j}) &\overset{(20)}{=} z_j \overset{(20)}{=} \widehat{f'}^{(j)}(\epsilon'_j, \boldsymbol{z}_{<j}) \\
&= [f'([f'^{-1}(\boldsymbol{z}_{<j}, \cdot)]_{<j}, \epsilon'_j, \cdot)]_j \\
&\overset{(a)}{\neq} [f'([f'^{-1}(\boldsymbol{z}_{<j}, \cdot)]_{<j}, \epsilon_j, \cdot)]_j \\
&= \widehat{f'}^{(j)}(\epsilon_j, \boldsymbol{z}_{<j}).
\end{aligned} \tag{21}
$$

where (a) comes from the $f' \in \mathcal{F}_I$. Thus it implies $j \in \mathcal{I}\left( \tilde{f}, \tilde{f}' \right)$.

**Step 2:** Prove $\mathcal{I}\left( \tilde{f}, \tilde{f}' \right) \subseteq \left\{ j : \left[ f^{-1} \right]_j \neq \left[ f'^{-1} \right]_j \right\}$.

For all $j \in \mathcal{I}\left( \tilde{f}, \tilde{f}' \right)$, there exists some $(\epsilon_j, \boldsymbol{z}_{<j})$, such that

$$z_j \triangleq \widehat{f}^{(j)}(\epsilon_j, \boldsymbol{z}_{<j}) \neq \widehat{f'}^{(j)}(\epsilon_j, \boldsymbol{z}_{<j}) \triangleq z'_j, \tag{22}$$

Define

$$\boldsymbol{z}_{\leq j} \triangleq [\boldsymbol{z}_{<j}, z_j] \quad \text{and} \quad \boldsymbol{z}'_{\leq j} \triangleq [\boldsymbol{z}_{<j}, z'_j], \tag{23}$$

then we have

$$[f^{-1}(\boldsymbol{z}_{\leq j}, \cdot)]_j = \epsilon_j = [f'^{-1}(\boldsymbol{z}'_{\leq j}, \cdot)]_j, \tag{24}$$

given that $f, f' \in \mathcal{F}_I$, we conclude,

$$[f^{-1}(\boldsymbol{z}_{\leq j}, \cdot)]_j \neq [f'^{-1}(\boldsymbol{z}_{\leq j}, \cdot)]_j, \tag{25}$$

which implies $j \in \left\{ j : \left[ f^{-1} \right]_j \neq \left[ f'^{-1} \right]_j \right\}$. $\qquad \square$

### B.4 Proof of Lemma 2

**Lemma 2** (Equivalence relation of counterfactual equivalence). *Domain counterfactually equivalent, denoted by $(g, \mathcal{F}) \simeq_C (g', \mathcal{F}')$ is an equivalence relation, i.e., the relation satisfies reflexivity, symmetry, and transitivity.*

*Proof.* We only need to prove that it satisfies reflexivity, symmetry, and transitivity.

1. Reflexivity - Letting $g' = g$ and $\mathcal{F}' = \mathcal{F}$ in the definition, it is trivial to see that $\forall d, d'$
$$g \circ f_{d'} \circ f_d^{-1} \circ g^{-1} = g \circ f_{d'} \circ f_d^{-1} \circ g^{-1},$$
and thus $(g, \mathcal{F}) \simeq_C (g', \mathcal{F}')$.

2. Symmetry - Similarly, it is trivial to see that $\forall d, d'$,
$$g \circ f_{d'} \circ f_d^{-1} \circ g^{-1} = g' \circ f'_{d'} \circ f_d'^{-1} \circ g'^{-1}$$
$$\iff \quad g' \circ f'_{d'} \circ f_d'^{-1} \circ g'^{-1} = g \circ f_{d'} \circ f_d^{-1} \circ g^{-1},$$
and thus $(g, \mathcal{F}) \simeq_C (g', \mathcal{F}') \Leftrightarrow (g', \mathcal{F}') \simeq_C (g, \mathcal{F})$.

3. Transitivity - For $(g, \mathcal{F})$, $(g', \mathcal{F}')$ and $(g'', \mathcal{F}'')$, we can derive the transitive property by applying the property twice to the first two and the last two pairs $\forall d, d'$:
$$g \circ f_{d'} \circ f_d^{-1} \circ g^{-1} = g' \circ f'_{d'} \circ f_d'^{-1} \circ g'^{-1} = g'' \circ f''_{d'} \circ f_d''^{-1} \circ g''^{-1},$$
which means that $(g, \mathcal{F}) \simeq_C (g'', \mathcal{F}'')$.

$\qquad \square$

### B.5 PROOF OF THEOREM 1

**Theorem 1** (Characterization of Counterfactual Equivalence). *Two ILDs are domain counterfactually equivalent, i.e., $(g, \mathcal{F}) \simeq_C (g', \mathcal{F}')$ if and only if:*

$$\exists\, h_1, h_2 \in \mathcal{F}_I \text{ s.t. } g' = g \circ h_1^{-1} \in \mathcal{F}_I \text{ and } f'_d = h_1 \circ f_d \circ h_2 \in \mathcal{F}_A\,, \forall d\,, \tag{1}$$

*and moreover, counterfactually equivalent models share the same intervention set size, i.e., if $(g, \mathcal{F}) \simeq_C (g', \mathcal{F}')$, then $|\mathcal{I}(\mathcal{F})| = |\mathcal{I}(\mathcal{F}')|$.*

Theorem 1 contains two parts. The first part characterizes the domain counterfactual equivalence model with two invertible functions. The second part proves that all counterfactual equivalent models share the same intervention set size.

#### B.5.1 PROOF OF THE REPRESENTATION OF DCF EQUIVALENCE

The proof of the domain counterfactual equivalence representation. relies heavily on one the following two lemmas that provides a necessary and sufficient condition for the composition of two invertible functions to be equal.

**Lemma 3** (Invertible Composition Equivalence). *For two pairs of invertible functions $(f_1, f_2)$ and $(f'_1, f'_2)$, the following two conditions are equivalent:*

1. *The compositions are equal:*

$$f_1 \circ f_2 = f'_1 \circ f'_2\,.$$

2. *There exists an intermediate invertible function $h$ s.t.*

$$f'_1 = f_1 \circ h^{-1}, f'_2 = h \circ f_2\,. \tag{26}$$

*Proof of Lemma 3.* For notational simplicity in this proof, we will let $g \triangleq f_1$, $f \triangleq f_2$, $g' \triangleq f'_1$ and $f' \triangleq f'_2$—note that $g$ and $f$ are just arbitrary invertible functions in this proof. Furthermore, without loss of generality, we will prove for the property $\exists\, h : g' = g \circ h, f' = h^{-1} \circ f$ which is equivalent to $\exists\, h : g' = g \circ h^{-1}, f' = h \circ f$. Thus, in the new notation, we are seeking to prove:

$$g \circ f = g' \circ f' \Leftrightarrow \exists\, h : g' = g \circ h, f' = h^{-1} \circ f \tag{27}$$

If $\exists\, h : g' = g \circ h, f' = h^{-1} \circ f$, then it is easy to show that $g \circ f = g' \circ f'$:

$$g' \circ f' = g \circ h \circ h^{-1} \circ f = g \circ f\,. \tag{28}$$

For the other direction, we will prove by contradiction. First, using Lemma 4, we can first rewrite $g'$ and $f'$ using the two uniquely determined invertible functions $h_1$ and $h_2$:

$$g' = g \circ h_1 \tag{29}$$
$$f' = h_2 \circ f. \tag{30}$$

Now, suppose that $g \circ f = g' \circ f'$ but $\nexists\, h$ such that $g' = g \circ h, f' = h^{-1} \circ f$. By the first assumption and the facts above, we can derive the following:

$$g \circ f = g' \circ f' = g \circ h_1 \circ h_2 \circ f \tag{31}$$
$$\Leftrightarrow f = h_1 \circ h_2 \circ f \tag{32}$$
$$\Leftrightarrow h_1^{-1} \circ f = h_2 \circ f \tag{33}$$

From the second assumption, i.e., $\nexists\, h : g' = g \circ h, f' = h^{-1} \circ f$, we have the following:

$$\forall\, h \text{ s.t. } g' = g \circ h, \text{ it holds that } f' \neq h^{-1} \circ f \tag{34}$$
$$\Rightarrow f' \neq h_1^{-1} \circ f \tag{35}$$
$$\Leftrightarrow h_2 \circ f \neq h_1^{-1} \circ f \tag{36}$$
$$\Leftrightarrow h_2 \neq h_1^{-1} \tag{37}$$

$$\Leftrightarrow h_2^{-1} \neq h_1 \,, \tag{38}$$

where (34) is by assumption, (35) follows from (29) because $h_1$ is one particular $h$, (36) is by our rewrite of $f'$ in (30), (37) is by the invertibility of $f$, and (38) is by invertibility of $h_1$ and $h_2$. Thus, there exists $\tilde{\boldsymbol{y}}$, such that $h_1^{-1}(\tilde{\boldsymbol{y}}) \neq h_2(\tilde{\boldsymbol{y}})$. Let us choose $\tilde{\boldsymbol{x}} \triangleq f^{-1}(\tilde{\boldsymbol{y}})$ for the $\tilde{\boldsymbol{y}}$ that satisfies the condition. For this $\tilde{\boldsymbol{x}}$, we then know that:

$$h_1^{-1} \circ f(\tilde{\boldsymbol{x}}) = h_1^{-1}(\tilde{\boldsymbol{y}}) \neq h_2(\tilde{\boldsymbol{y}}) = h_2 \circ f(\tilde{\boldsymbol{x}}) \tag{39}$$

$$\Leftrightarrow h_1^{-1} \circ f \neq h_2 \circ f \,. \tag{40}$$

But this leads to a direct contradiction of (33). Therefore, if $g \circ f = g' \circ f'$, then $\exists\, h : g' = g \circ h, f' = h^{-1} \circ f$. $\qquad\square$

**Lemma 4** (Invertible function rewrite). *Given any two invertible functions $f : \mathcal{X} \to \mathcal{Y}$ and $f' : \mathcal{X} \to \mathcal{Y}$, $f'$ can be decomposed into the composition of $f$ and another invertible function. Specifically, $f'$ can be decomposed in the following two ways:*

$$f' \equiv f \circ h_{\mathcal{X}} \tag{41}$$

$$f' \equiv h_{\mathcal{Y}} \circ f \,, \tag{42}$$

*where $h_{\mathcal{X}} \triangleq f^{-1} \circ f' : \mathcal{X} \to \mathcal{X}$ and $h_{\mathcal{Y}} \triangleq f' \circ f^{-1} : \mathcal{Y} \to \mathcal{Y}$ are both invertible functions.*

*Proof of Lemma 4.* The proof is straightforward. We first note that $h_{\mathcal{X}}$ and $h_{\mathcal{Y}}$ are invertible because they are compositions of invertible functions. Then, we have that:

$$f \circ h_{\mathcal{X}} = f \circ f^{-1} \circ f' = f' \tag{43}$$

$$h_{\mathcal{Y}} \circ f = f' \circ f^{-1} \circ f = f' \,. \tag{44}$$

$$\square$$

*Proof of Theorem 1: Part 1.* The basic idea is to use repeated application of Lemma 3 under the constraint that $h_1$ and $h_2$ must be shared across for all $d$ and $g$ and $g^{-1}$ must be inverses of each other.

For one direction as in Lemma 3, if (1) holds, it is nearly trivial to prove the equation in (3) holds, i.e., for all $d, d'$:

$$g' \circ f'_{d'} \circ f'^{-1}_d \circ g'^{-1} = (g \circ h_1^{-1}) \circ (h_1 \circ f_{d'} \circ h_2) \circ (h_2^{-1} \circ f_d^{-1} \circ h_1^{-1}) \circ (h_1 \circ g^{-1})$$

$$= g \circ f_{d'} \circ f_d^{-1} \circ g^{-1} \,.$$

To prove the other direction, let us define the following functions for a specific $(d, d')$ (we will treat the case of all $(d, d')$ afterwards): $f_1 \triangleq g^{-1}, f_2 \triangleq f_d^{-1}, f_3 \triangleq f_{d'}$, and $f_4 \triangleq g$ and similarly $f'_1, f'_2, f'_3$, and $f'_4$ for the other side. Given these definitions, we can write the property as:

$$f_4 \circ f_3 \circ f_2 \circ f_1 = f'_4 \circ f'_3 \circ f'_2 \circ f'_1 \,.$$

By recursively applying Lemma 3 for each of the three function compositions, we arrive at the following fact:

$$\exists\, h_1, h_2, h_3, \quad \text{s.t.} \quad \begin{cases} f'_1 = h_1 \circ f_1 \text{ and } f'_4 \circ f'_3 \circ f'_2 = f_4 \circ f_3 \circ f_2 \circ h_1^{-1} \\ f'_2 = h_2 \circ f_2 \circ h_1^{-1} \text{ and } f'_4 \circ f'_3 = f_4 \circ f_3 \circ h_2^{-1} \\ f'_3 = h_3 \circ f_3 \circ h_2^{-1} \text{ and } f'_4 = f_4 \circ h_3^{-1} \end{cases}$$

By using the definitions of $f_1, f_2$, etc., we can now derive the following:

$$g' = g \circ h_3^{-1}$$

$$f'_{d'} = h_3 \circ f_{d'} \circ h_2^{-1}$$

$$f'^{-1}_d = h_2 \circ f_d^{-1} \circ h_1^{-1}$$

$$g'^{-1} = h_1 \circ g^{-1} \,.$$

We can connect the first and the last equality to derive that $h_3 = h_1$:

$$g'^{-1} = h_1 \circ g^{-1}$$
$$\Leftrightarrow g' = g \circ h_1^{-1} = g \circ h_3^{-1}$$
$$\Leftrightarrow h_1^{-1} = h_3^{-1}$$
$$\Leftrightarrow h_1 = h_3 \,.$$

Thus, there are only two free functions. Specifically, for any fixed pair of $(d, d')$ there exist $h_{1,d,d'} (\equiv h_{3,d,d'})$ and $h_{2,d,d'}$ such that

$$g' = g \circ h_{1,d,d'}^{-1}, f_d' = h_{1,d,d'} \circ f_d \circ h_{2,d,d'}^{-1}, \text{ and } f_{d'}' = h_{1,d,d'} \circ f_{d'} \circ h_{2,d,d'}^{-1}.$$

Finally, we tackle the case of all $(d, d')$ by assuming that there could be unique functions $h_{1,d,d'}$ and $h_{2,d,d'}$ for all pairs of $(d, d')$ and show that they are in fact equal. Because the condition holds for *all* $(d, d')$, we know that for any particular $(d, d')$ and $(d'', d)$, we have the following two things based on the proof above:

$$g' \circ f_{d'}' \circ f_d'^{-1} \circ g'^{-1} = g \circ f_{d'} \circ f_d^{-1} \circ g^{-1}$$

$$\Leftrightarrow \exists \, h_{1,d,d'}, h_{2,d,d'} \text{ s.t. } \begin{cases} g' = g \circ h_{1,d,d'}^{-1} \\ f_d' = h_{1,d,d'} \circ f_d \circ h_{2,d,d'}^{-1} \\ f_{d'}' = h_{1,d,d'} \circ f_{d'} \circ h_{2,d,d'}^{-1} \end{cases}$$

$$g' \circ f_d' \circ f_{d''}'^{-1} \circ g'^{-1} = g \circ f_d \circ f_{d''}^{-1} \circ g^{-1}$$

$$\Leftrightarrow \exists \, h_{1,d'',d}, h_{2,d'',d} \text{ s.t. } \begin{cases} g' = g \circ h_{1,d'',d}^{-1} \\ f_{d''}' = h_{1,d'',d} \circ f_{d''} \circ h_{2,d'',d}^{-1} \\ f_d' = h_{1,d'',d} \circ f_d \circ h_{2,d'',d}^{-1} \end{cases} \,.$$

By equating the RHS for the $g'$ equations, we can thus derive that:

$$g \circ h_{1,d,d'}^{-1} = g \circ h_{1,d'',d}^{-1}$$
$$\Leftrightarrow \quad h_{1,d,d'} = h_{1,d'',d} \,.$$

Using this fact and similarly by equating the RHS for the $f_d'$ equations, we can derive:

$$f_d' = h_{1,d,d'} \circ f_d \circ h_{2,d,d'}^{-1} = h_{1,d'',d} \circ f_d \circ h_{2,d'',d}^{-1} = h_{1,d,d'} \circ f_d \circ h_{2,d'',d}^{-1}$$
$$\Leftrightarrow \quad h_{2,d,d'}^{-1} = h_{2,d'',d}^{-1}$$
$$\Leftrightarrow \quad h_{2,d,d'} = h_{2,d'',d} \,.$$

By applying these facts to all possible triples of $(d, d', d'')$, we can conclude that $\forall d, d', h_{1,d,d'} = h_1$, $h_{2,d,d'} = h_2$, i.e., these intermediate functions must be independent of $d$ and $d'$. Finally, we can adjust notation so that $\forall d, f_d' = \tilde{h}_1 \circ f_d \circ \tilde{h}_2$ and $g' = g \circ \tilde{h}_1^{-1}$, where $\tilde{h}_1 \triangleq h_1$ and $\tilde{h}_2 \triangleq h_2^{-1}$, which matches the result in the theorem. □

### B.5.2 PROOF OF THE SHARING INTERVENTION SET SIZE BETWEEN DCF EQUIVALENCE MODELS

Now we aim to prove the second part of Theorem 1, which states that all DCF equivalence models share the same intervention set size. The proof requires the concept of canonical form, please refer Definition 5 for the definition of canonical form and Theorem 3 for the existence of a special kind of canonical form we refer as Idendity Canonical, where all the un-internvend nodes are independent standard Gaussian.

For two ILDS $(g, \mathcal{F}) \simeq_{C,D} (g', \mathcal{F}')$, we apply Theorem 3 to get Identity Canonical form $(g_C, \mathcal{F}_C) \simeq_{C,D} (g, \mathcal{F})$ and similarly $(g_C', \mathcal{F}_C') \simeq_{C,D} (g', \mathcal{F}')$. Then we use the following Proposition 4, we show they must have the same intervention set size. Lastly, notice that the DCF equivalence is a equivalence relation Lemma 2, we can show all the DCF equivalence models share the same intervention set size.

Before proving Proposition 4, we first introduce a lemma that will be used in the main proof.

**Lemma 5.** *If* $f : \mathbb{R}^m \to \mathbb{R}^m \in \mathcal{F}_{IA}$, *then* $[f(\boldsymbol{x})]_k$ *must be a non-constant function of* $x_k$.

*Proof.* We prove this by contradiction. Suppose $k$ is the first index that $[f(\boldsymbol{x})]_k = \widetilde{f}(x_1, \ldots, x_{k-1})$. Since $k$ is the smallest index, $[f(\boldsymbol{x})]_{<k}$ is uniquely determined by $[\boldsymbol{x}]_{\leq k}$, The remaining $m - k$ dimension outputs could not be bijective to $m - k + 1$ inputs. $\qquad\square$

**Proposition 4** (Identity Canonical ILD Shares Intervention Sparsity). *Given an ILD* $(g, \mathcal{F})$, *and* $g, f_d \in \mathcal{F}$, *for all* $d \in m$ *are continuous, then all Identity Canonical ILDs that are distributionally and counterfactually equivalent to* $(g, \mathcal{F})$ *have the same intervention set, i.e.,*

$$\mathcal{I}(\mathcal{F}) = \mathcal{I}(\mathcal{F}'), \quad \forall (g', \mathcal{F}') \in \left\{ (\widetilde{g}, \widetilde{\mathcal{F}}) \in \mathcal{C} : (\widetilde{g}, \widetilde{\mathcal{F}}) \simeq_D (g, \mathcal{F}), (\widetilde{g}, \widetilde{\mathcal{F}}) \simeq_C (g, \mathcal{F}) \right\}. \tag{45}$$

*Proof of Proposition 4.* In the proof, we denote $F$ as a non constant function without specifying the expression.

**Step 1: Characterization of counterfactual equivalence for Identity Canonical forms.** Theorem 1 states that there exists $h_1, h_2 \in \mathcal{F}_I$, such that for all $d$,

$$f'_d = h_1 \circ f_d \circ h_2. \tag{46}$$

Furthermore, by the definition of Identity Canonical form, we have

$$f'_1 = \mathrm{Id}, f_1 = \mathrm{Id}. \tag{47}$$

Plugging this into (46), we have

$$\mathrm{Id} = h_1 \circ \mathrm{Id} \circ h_2.$$

Thus,

$$h_1^{-1} = h_2 \triangleq h.$$

Plugging this into (46), for all $d$, we have

$$f'^{-1}_d = h^{-1} \circ f_d^{-1} \circ h. \tag{48}$$

**Step 2: Counterfactual equivalence between Identity Canonical forms maintain the intervention set.** The goal of this step is to prove that $h$ is a bridge satisfying the following property: for any $i \notin \mathcal{I}(f'_1, f'_d)$, for all $x$, there exists an unique $j$, such that $[h^{-1}(\boldsymbol{x})]_i$ only depends on $x_j$. In addition, we can prove such $j$ satisfies $j \notin \mathcal{I}(f_1, f_d)$.

We start with writing the $i$-th output of $f'^{-1}_d(\boldsymbol{x})$ as the following

$$[f'^{-1}_d(\boldsymbol{x})]_i \overset{(48)}{=} [h^{-1}(f_d^{-1}(h(\boldsymbol{x})))]_i \tag{49}$$

$$= [h^{-1} ([f_d^{-1}(h(\boldsymbol{x}))]_1, [f_d^{-1}(h(\boldsymbol{x}))]_2, \ldots, [f_d^{-1}(h(\boldsymbol{x}))]_m)]_i \tag{50}$$

$$\overset{(a)}{=} \left[ h^{-1} \left( \widetilde{f}_{d,1}^{-1}([h(\boldsymbol{x})]_1), \widetilde{f}_{d,2}^{-1}([h(\boldsymbol{x})]_1, [h(\boldsymbol{x})]_2), \ldots, \widetilde{f}_{d,m}^{-1}([h(\boldsymbol{x})]_1, \ldots [h(\boldsymbol{x})]_m) \right) \right]_i, \tag{51}$$

where in step (a), we used autoregresiveness of $f_d^{-1}$, and $\widetilde{f}_{d,k}^{-1}$ is defined as a function from $\mathbb{R}^k$ to $\mathbb{R}$. According to Lemma 5, $\widetilde{f}_{d,k}^{-1}(\boldsymbol{x})$ is a non-constant function of $x_k$.

**Step 2.1: We show $i$ and $j$ must be one-to-one mapping of $h$.** We proof this by contradiction.

Suppose $h^{-1}$ maps more than one index to $i$-th index, w.l.o.g, we could assume $j_1$ and $j_2$. That is, $[h^{-1}(\boldsymbol{u})]_i$ depends on $u_{j_1}$ and $u_{j_2}$. Take $\boldsymbol{u} = f_d^{-1}(h(\boldsymbol{x}))$, then we have

$$[f'^{-1}_d(\boldsymbol{x})]_i = F \left( \widetilde{f}_{d,j_1}^{-1}([h(\boldsymbol{x})]_1, \ldots [h(\boldsymbol{x})]_{j_1}), \widetilde{f}_{d,j_2}^{-1}([h(\boldsymbol{x})]_1, \ldots [h(\boldsymbol{x})]_{j_2}) \right) \tag{52}$$

Due to that $f_d^{-1} \in \mathcal{F}_{IA}$, from Lemma 5, we have

$$\widetilde{f}_{d,j_1}^{-1}([h(\boldsymbol{x})]_1, \ldots [h(\boldsymbol{x})]_{j_1}) = F([h(\boldsymbol{x})]_{j_1}, \cdot) \tag{53}$$

$$\widetilde{f}_{d,j_2}^{-1}([h(\boldsymbol{x})]_1, \ldots [h(\boldsymbol{x})]_{j_2}) = F([h(\boldsymbol{x})]_{j_2}, \cdot). \tag{54}$$

Plug (53), (54) into (52), we have

$$[f_d'^{-1}(\boldsymbol{x})]_i = F([h(\boldsymbol{x})]_{j_1}, [h(\boldsymbol{x})]_{j_2}, \cdot) \tag{55}$$

Given that $h \in \mathcal{F}_{IA}$, we conclude $([h(\boldsymbol{x})]_{j_1}, [h(\boldsymbol{x})]_{j_2})$ depend at least two distinct indices. That is, there exists $i_1, i_2$ such that

$$([h(\boldsymbol{x})]_{j_1}, [h(\boldsymbol{x})]_{j_2}) = F(x_{i_1}, x_{i_2}). \tag{56}$$

That implies $[f_d'^{-1}(\boldsymbol{x})]_i$ is a nontrivial function of $(x_{i_1}, x_{i_2})$. This leads to the contradiction that $i \notin \mathcal{I}(f_1', f_d')$, where for all $\boldsymbol{x}$, there holds

$$[f_d'^{-1}(\boldsymbol{x})]_i = x_i \tag{57}$$

**Step 2.2: We show such $j$ is not in the intervention set between $f_1$ and $f_d$.** We prove this by contradiction as well.

Step 2.1 implies

$$[f_d'^{-1}(\boldsymbol{x})]_i = F\left(\widetilde{f}_{d,j}^{-1}([h(\boldsymbol{x})]_1, \ldots [h(\boldsymbol{x})]_j)\right) = F'([h(\boldsymbol{x})]_j, \cdot), \tag{58}$$

Suppose $j \in \mathcal{I}(f_1, f_d)$, then $f_d^{-1}([h(\boldsymbol{x})]_j)$ Recall that $f_d^{-1} \in \mathcal{F}_A$,

then $[f_d^{-1}(h(\boldsymbol{x}))]_j$ must be a non-constant function of $[h(\boldsymbol{x})]_j$ and $[h(\boldsymbol{x})]_{j'}$ for some $j' < j$, i.e.,

$$[f_d^{-1}(h(\boldsymbol{x}))]_j = \widetilde{f}_{d,j}^{-1}([h(\boldsymbol{x})]_1, \ldots, [h(\boldsymbol{x})]_j) = F([h(\boldsymbol{x})]_{j'}, [h(\boldsymbol{x})]_j, \cdot). \tag{59}$$

Similarly, we know that $[h(\boldsymbol{x})]_{j'}$ and $[h(\boldsymbol{x})]_j$ must be nontrivial functions of $x_{i_3}$ and $x_{i_4}$, which $i_3 \neq i_4$. However, we know $[f_d'^{-1}(\boldsymbol{x})]_i$ is a function of $x_i$ exclusively, which leads to contradiction. This shows that the number of non-intervened node in $f_d'$ must not be greater than that in $f_d$, i.e.,

$$\mathcal{I}(f_1', f_d') \geq \mathcal{I}(f_1, f_d), \forall d. \tag{60}$$

We further notice that the symmetric relationship between $f_d$ and $f_d'$, we could also have

$$\mathcal{I}(f_1', f_d') \leq \mathcal{I}(f_1, f_d), \forall d. \tag{61}$$

Union among on $d$, we have

$$\mathcal{I}(f') = \mathcal{I}(f). \tag{62}$$

$\square$

## B.6    Proof of Theorem 2

**Lemma 6** (Counterfactual Pseudo-Metric for ILD Model is a pseudo-metric)**.**

*Proof.* It is trivial to check that it is always positive, symmetric, and equal to 0 if $(g, \mathcal{F}) = (\bar{g}, \bar{\mathcal{F}})$. Finally, because RMSE satisfies the triangle inequality (Chai and Draxler, 2014), Definition 4 also satisfies the triangle inequality. $\square$

**Theorem 2** (Counterfactual Error Bound Decomposition)**.** *Given a max intervention sparsity $k \geq 0$ and letting $\mathcal{M}(k) \triangleq \{(g, \mathcal{F}) : (g, \mathcal{F}) \simeq_D (g^*, \mathcal{F}^*), |\mathcal{I}(\mathcal{F})| \leq \max\{k, |\mathcal{I}(\mathcal{F}^*)|\}\}$, the counterfactual error can be upper bounded as follows:*

$$\varepsilon(\hat{g}, \hat{\mathcal{F}}) \leq \underbrace{\min_{(g', \mathcal{F}') \in \mathcal{M}(k)} d_C((\hat{g}, \hat{\mathcal{F}}), (g', \mathcal{F}'))}_{\text{(A) Error due to lack of distribution equivalence}} + \underbrace{\max_{(\tilde{g}, \widetilde{\mathcal{F}}) \in \mathcal{M}(k)} d_C((\tilde{g}, \widetilde{\mathcal{F}}), (g^*, \mathcal{F}^*))}_{\text{(B) Worst-case error given distribution equivalence}} . \tag{2}$$

*Furthermore, if we assume that the ILD mixing functions are Lipschitz continuous, we can bound the worst-case error (B) as follows:*

$$(B) \leq \left[ \underbrace{\max_{(\tilde{g}, \widetilde{\mathcal{F}}) \in \mathcal{M}(k)} \widetilde{k} \, L_{\tilde{g}}^2 \max_{i \in [m]} \mathbb{E}\left[[\widetilde{f}_d(\boldsymbol{\epsilon}) - \widetilde{f}_{d'}(\boldsymbol{\epsilon})]_i^2\right]}_{\text{Error depends on } k \text{ since } \widetilde{k} \leq \max\{k, k^*\}} + \underbrace{k^* L_{g^*}^2 \max_{i \in [m]} \mathbb{E}\left[[f_d^*(\boldsymbol{\epsilon}) - f_{d'}^*(\boldsymbol{\epsilon})]_i^2\right]}_{\text{Error only depends on ground truth model}} \right]^{1/2},$$

*where $\widetilde{k} \equiv |\mathcal{I}(\widetilde{\mathcal{F}})|$ and $k^* \equiv |\mathcal{I}(\mathcal{F}^*)|$, $L_g$ is the Lipchitz constant of $g$, and the expectation is over $p(d, d', \boldsymbol{\epsilon}) \triangleq p(d)p(d')p(\boldsymbol{\epsilon})$.*

*Proof of Theorem 2.* We will prove this theorem when both $(\widehat{g}, \widehat{f})$ and $(g^*, \mathcal{F}^*)$ are both canonical forms (See Definition 5). According Theorem 3, any two ILD's counterfactual error are equivalent two their equivalent canonical models, thus this bound holds for all pairs of ILD models.

(2) is by the triangle inequality for *any* intervening $(\tilde{g}, \tilde{\mathcal{F}})$ and in this case we choose to minimize the bound over all possible distributionally equivalent models with a bounded sparsity—we know that at least the true ILD satisfies this, and thus there is at least one feasible solution. (2) is by the fact that choosing the ILD model with the worst counterfactual error is larger than the error incurred by $(\tilde{g}, \tilde{\mathcal{F}})$, under the same constraints—again, by construction, $(\tilde{g}, \tilde{\mathcal{F}})$ satisfies the constraints in the maximization problem and thus at least one ILD model is feasible.

Now we prove the worst-case counterfactual misspecification error bound.

$$\max_{(\widetilde{g},\widetilde{\mathcal{F}})\in\mathcal{M}(k)} d_C^2((\widetilde{g},\widetilde{\mathcal{F}}),(g^*,\mathcal{F}^*))$$

$$= \max_{(\widetilde{g},\widetilde{\mathcal{F}})\in\mathcal{M}(k)} \mathbb{E}_{p(d')}\mathbb{E}_{p(\boldsymbol{x},d)}\left[\left\|\widetilde{g}\circ\widetilde{f}_{d'}\circ\widetilde{f}_d^{-1}\circ\widetilde{g}^{-1}(\boldsymbol{x}) - g^*\circ f_{d'}^*\circ f_d^{*-1}\circ g^{*-1}(\boldsymbol{x})\right\|_2^2\right]$$
(Definition)

$$= \max_{(\widetilde{g},\widetilde{\mathcal{F}})\in\mathcal{M}(k)} \mathbb{E}_{p(d')}\mathbb{E}_{p(\boldsymbol{x},d)}\left[\left\|\widetilde{g}\circ\widetilde{f}_{d'}\circ\widetilde{f}_d^{-1}\circ\widetilde{g}^{-1}(\boldsymbol{x}) - \boldsymbol{x} + \boldsymbol{x} - g^*\circ f_{d'}^*\circ f_d^{*-1}\circ g^{*-1}(\boldsymbol{x})\right\|_2^2\right]$$
(Inflation)

$$= \max_{(\widetilde{g},\widetilde{\mathcal{F}})\in\mathcal{M}(k)} \mathbb{E}_{p(d')}\mathbb{E}_{p(\boldsymbol{x},d)}\Big[\Big\|\widetilde{g}\circ\widetilde{f}_{d'}\circ\widetilde{f}_d^{-1}\circ\widetilde{g}^{-1}(\boldsymbol{x}) - \widetilde{g}\circ\widetilde{f}_d\circ\widetilde{f}_d^{-1}\circ\widetilde{g}^{-1}(\boldsymbol{x})$$
$$+ g^*\circ f_d^*\circ f_d^{*-1}\circ g^{*-1}(\boldsymbol{x}) - g^*\circ f_{d'}^*\circ f_d^{*-1}\circ g^{*-1}(\boldsymbol{x})\Big\|_2^2\Big]$$
(Invertibility of ILD)

$$\leq \max_{(\widetilde{g},\widetilde{\mathcal{F}})\in\mathcal{M}(k)} 2\mathbb{E}_{p(d')}\mathbb{E}_{p(\boldsymbol{x},d)}\left[\left\|\widetilde{g}\circ\widetilde{f}_{d'}\circ\widetilde{f}_d^{-1}\circ\widetilde{g}^{-1}(\boldsymbol{x}) - \widetilde{g}\circ\widetilde{f}_d\circ\widetilde{f}_d^{-1}\circ\widetilde{g}^{-1}(\boldsymbol{x})\right\|_2^2\right]$$
$$+ 2\mathbb{E}_{p(d')}\mathbb{E}_{p(\boldsymbol{x},d)}\left[\left\|g^*\circ f_d^*\circ f_d^{*-1}\circ g^{*-1}(\boldsymbol{x}) - g^*\circ f_{d'}^*\circ f_d^{*-1}\circ g^{*-1}(\boldsymbol{x})\right\|_2^2\right]$$
(AM-QM Inequality)

$$\triangleq \max_{(\widetilde{g},\widetilde{\mathcal{F}})\in\mathcal{M}(k)} 2\mathbb{E}_{p(d)}\mathbb{E}_{p(d')}\mathbb{E}_{p(\boldsymbol{\epsilon})}\left[\left\|\widetilde{g}\circ\widetilde{f}_{d'}(\boldsymbol{\epsilon}) - \widetilde{g}\circ\widetilde{f}_d(\boldsymbol{\epsilon})\right\|_2^2\right]$$
$$+ 2\mathbb{E}_{p(d)}\mathbb{E}_{p(d')}\mathbb{E}_{p(\boldsymbol{\epsilon'})}\left[\left\|g^*\circ f_{d'}^*(\boldsymbol{\epsilon'}) - g^*\circ f_d^*(\boldsymbol{\epsilon'})\right\|_2^2\right]$$

$\square$

then we aim to bound the both term related to worse-case ILD term and ground-truth ILD term.

**Lemma 7.** *If $g$ is Lipschitz continuous with constant $L_g$ and $k = |\mathcal{I}(\mathcal{F})|$, then the following holds:*

$$\mathbb{E}_{p(d)p(d')p(\boldsymbol{\epsilon})}[\|g\circ f_{d'}(\boldsymbol{\epsilon}) - g\circ f_d(\boldsymbol{\epsilon})\|_2^2] \leq L_g^2 \cdot k \cdot \max_{i\in[m]} \mathbb{E}_{p(d)p(d')p(\boldsymbol{\epsilon})}\left[[f_d(\boldsymbol{\epsilon}) - f_{d'}(\boldsymbol{\epsilon})]_i^2\right] \quad (63)$$

*Proof.*

$$\mathbb{E}_{p(d)p(d')p(\boldsymbol{\epsilon})}[\|g\circ f_{d'}(\boldsymbol{\epsilon}) - g\circ f_d(\boldsymbol{\epsilon})\|_2^2] \quad (64)$$
$$\leq L_g^2\mathbb{E}_{p(d)p(d')p(\boldsymbol{\epsilon})}[\|f_{d'}(\boldsymbol{\epsilon}) - f_d(\boldsymbol{\epsilon})\|_2^2]$$
(Lipschitz)
$$= L_g^2\mathbb{E}_{p(\boldsymbol{z},d)p(d')}[\|f_{d'}(f_d^{-1}(\boldsymbol{z})) - \boldsymbol{z}\|_2^2] \quad (65)$$
$$= L_g^2 d_C^2((\text{Id},\mathcal{F}),(\text{Id},\text{Id}))$$
(Interpretation as counterfactual error)
$$= L_g^2\mathbb{E}_{p(\boldsymbol{z},d)p(d')}[\sum_{i=1}^m [f_{d'}(f_d^{-1}(\boldsymbol{z})) - \boldsymbol{z}]_i^2] \quad (66)$$
$$= L_g^2\mathbb{E}_{p(\boldsymbol{z},d)p(d')}[\sum_{i=0}^{k-1} [f_{d'}(f_d^{-1}(\boldsymbol{z})) - \boldsymbol{z}]_{m-i}^2]$$
(Canonical form)

$$= L_g^2 \sum_{i=0}^{k-1} [\mathbb{E}_{p(\boldsymbol{z},d)p(d')}[f_{d'}(f_d^{-1}(\boldsymbol{z})) - \boldsymbol{z}]_{m-i}^2] \tag{67}$$

$$\leq L_g^2 \cdot k \cdot \max_{i:i<k} \mathbb{E}_{p(\boldsymbol{z},d)p(d')}[f_{d'}(f_d^{-1}(\boldsymbol{z})) - \boldsymbol{z}]_{m-i}^2 \tag{68}$$

$$= L_g^2 \cdot k \cdot \max_{i\in[m]} \mathbb{E}_{p(\boldsymbol{z},d)p(d')}[f_{d'}(f_d^{-1}(\boldsymbol{z})) - \boldsymbol{z}]_i^2 \tag{69}$$

$$= L_g^2 \cdot k \cdot \max_{i\in[m]} \mathbb{E}_{p(\boldsymbol{z},d)p(d')}[\boldsymbol{z} - f_{d'}(f_d^{-1}(\boldsymbol{z}))]_i^2 \tag{rearrange}$$

$$= L_g^2 \cdot k \cdot \max_{i\in[m]} \mathbb{E}_{p(d)p(d')p(\boldsymbol{\epsilon})}\big[[f_d(\boldsymbol{\epsilon}) - f_{d'}(\boldsymbol{\epsilon})]_i^2\big] \tag{change back to $\boldsymbol{\epsilon}$}$$

where the distribution for $d_C$ in this case is the one induced by $\mathcal{F}$. $\qquad\square$

### B.7 PROOF OF THEOREM 3

The proof of Theorem 3 relies on the Swapping Lemma (Lemma 9), and before proving the swapping lemma, we first introduce a lemma which will be used in the swapping lemma to show that if one domain in an ILD is identity, then we could check intervention set using $f_d$ instead of $f_d^{-1}$.

**Lemma 8.** *For an ILD with $f_1 = \mathrm{Id}$, $\mathcal{I}(f_d, f_1) = \left\{j : [f_d]_j \neq [f_1]_j\right\}$.*

*Proof of Lemma 8.* Suppose $f_d^{-1}(\boldsymbol{x}) = \boldsymbol{x}'$ where $x_j' \neq x_j$, then $f_d(\boldsymbol{x}') = \boldsymbol{x}$ becasue that $f_d$ is bijective. Then

$$[f_d(\boldsymbol{x}')]_j = x_j \neq x_j' = f_1(\boldsymbol{x}').$$

For any $j \notin \mathcal{I}(\mathcal{F})$, for any $\boldsymbol{x} = f_d(\boldsymbol{x}')$, we have $x_j' = [f_d^{-1}(\boldsymbol{x})]_j = [f_1^{-1}(\boldsymbol{x})]_j = x_j \Rightarrow x_j = x_j'$, thus

$$x_j = [f_d(\boldsymbol{x}')]_j = [f_d(\boldsymbol{x}')]_j = x_j'.$$

$\qquad\square$

**Lemma 9** (Swapping Lemma). *Given that the first canonical counterfactual property is satisfied, i.e., $f_1 = \mathrm{Id}$, denote $f'$ as SCM constructed by $f' = h_1 \circ f \circ h_2(x)$, where $h_1 = h_2$ denote swapping the $j$-th feature with $j'$-th feature. Then there exists $g'$ such that*

$$(g, \mathcal{F}) \simeq_C (g', \mathcal{F}'), \quad f_1' = \mathrm{Id}, \quad \mathcal{I}(\mathcal{F}') = (\mathcal{I}(\mathcal{F}) \setminus \{j\}) \cup \{j'\}.$$

*if the following conditions hold*

$$j \in \mathcal{I}(\mathcal{F}) \text{ and } \forall \tilde{j} : j < \tilde{j} \leq j', \ \tilde{j} \notin \mathcal{I}(\mathcal{F}).$$

*Proof of Lemma 9.* First, note that because $j'$ is not intervened, then we can derive that it's corresponding conditional function is independent of all but the $j'$-th value:

$$[f_d]_{j'} = [f_1]_{j'} \tag{70}$$

$$\Leftrightarrow \quad f_{d,j'}(\boldsymbol{x}_{\leq j'}) = f_{1,j'}(\boldsymbol{x}_{\leq j'}) = x_{j'}. \tag{71}$$

For the new model, we choose the invertible functions as swapping the $j$-th and $j'$-th feature values, i.e.,

$$h_1(\boldsymbol{x}) \triangleq [x_1, x_2, \cdots, x_{j-1}, x_{j'}, x_{j+1}, \cdots, x_{j'-1}, x_j, x_{j'+1}, \cdots, x_m]^T \tag{72}$$

and similarly for $h_2$, i.e., $h_2 \triangleq h_1$. Because $h_1$ and $h_2$ are invertible, we know that the new model will be in the same counterfactual equivalence class by Theorem 1. Construct $g' \triangleq g \circ h_1^{-1}$, and then for all $d$,

$$\begin{aligned} f_d'(x) =& h_1 \circ f_d \circ h_2(x) \\ =& h_1 \circ f_d([x_1, x_2, \cdots, x_{j-1}, x_{j'}, x_{j+1}, \cdots, x_{j'-1}, x_j, x_{j'+1}, \cdots, x_m]^T]) \\ =& h_1 \circ f_d([y_1, y_2, \cdots, y_{j-1}, y_j, y_{j+1}, \cdots, y_{j'-1}, y_{j'}, y_{j'+1}, \cdots, y_m]^T]) \\ =& h_1 \circ [f_{d,i}(\boldsymbol{y}_{\leq i})]_{i=1}^m \end{aligned}$$

$$= \Big[ f_{d,1}(\boldsymbol{y}_1), \cdots, f_{d,j-1}(\boldsymbol{y}_{\leq j-1}), f_{d,j'}(\boldsymbol{y}_{\leq j'}), f_{d,j+1}(\boldsymbol{y}_{\leq j+1}), \cdots,$$

$$f_{d,j'-1}(\boldsymbol{y}_{\leq j'-1}), f_{d,j}(\boldsymbol{y}_{\leq j}), f_{d,j'+1}(\boldsymbol{y}_{\leq j'+1}), \cdots, f_{d,m}(\boldsymbol{y}_{\leq m}) \Big],$$

where we define $\boldsymbol{y} \triangleq h_2^{-1}(\boldsymbol{x})$.

We now need to check that the first canonical counterfactual property still holds.

$$f_1' = h_1 \circ f_1 \circ h_2 = h_1 \circ \mathrm{Id} \circ h_2 = h_1 \circ h_2 = \mathrm{Id}, \tag{73}$$

where the last equals is because swap operations are self-invertible.

We move to check that the autoregressive property still holds for other domain SCMs.

**1)** For the $j$-th feature, we have that:

$$[f_d'(\boldsymbol{x})]_j = f_{d,j'}(\boldsymbol{y}_{\leq j'}) = f_{d,j'}(x_1, \cdots, x_{j-1}, x_{j'}, x_{j+1}, \cdots, x_{j'-1}, x_j) = x_j$$

where the last equals is because the $f_{d,j'}(\boldsymbol{y}_{\leq j'}) = y_{j'} = x_j$. This clearly satisfies the autoregressive property as $[f_d']_j$ only depends on $x_j$.

**2)** For the $j'$-th feature, we have that:

$$[f_d'(\boldsymbol{x})]_{j'} = f_{d,j}(\boldsymbol{y}_{\leq j}) = f_{d,j'}(x_1, \cdots, x_{j-1}, x_{j'})$$

where again this satisfies the autoregressive property because all input indices are less than $j'$ because $j < j'$. Now we handle the cases for other variables. If $\tilde{j} < j$, then we have the following:

$$[f_d']_{\tilde{j}} = [h_1 \circ f_d \circ h_2]_{\tilde{j}} = [f_d \circ h_2]_{\tilde{j}} = f_{d,\tilde{j}}([h_2(\boldsymbol{x})]_{\leq \tilde{j}}) = f_{d,\tilde{j}}(x_1, \ldots, x_{\tilde{j}}) \tag{74}$$

**3)** Similarly if $j < \tilde{j} < j'$:

$$[f_d']_{\tilde{j}} = f_{d,\tilde{j}}(x_1, \ldots, x_{j-1}, x_{j'}, x_{j+1}, \cdots, x_{\tilde{j}}) = x_{\tilde{j}}, \tag{75}$$

where we use the fact that there are no intervening nodes in between $j$ and $j'$.

**4)** Finally, for $\tilde{j} > j'$, we have:

$$[f_d']_{\tilde{j}} = f_{d,\tilde{j}}(x_1, \cdots, x_{j-1}, x_{j'}, x_{j+1}, \cdots, x_{j'-1}, x_j, x_{j'+1}, \cdots, x_{\tilde{j}}), \tag{76}$$

which is still autoregressive because $\tilde{j} > j'$ and $\tilde{j} > j$. Thus, the new $f_d'$ is autoregressive and is thus a valid model.

It remains to prove that $\mathcal{I}(\mathcal{F}') = (\mathcal{I}(\mathcal{F}) \setminus \{j\}) \cup \{j'\}$.

**1)** When $k < j$, we have for all $d$,

$$[f_d']_k = f_{d,k}(\boldsymbol{y}_{\leq k}) = f_{d,k}(\boldsymbol{x}_{\leq k}) = [f_d]_k,$$

then for all $k \in \mathcal{I}(\mathcal{F})$, there exists $d_0$, such that

$$[f_{d_0}'^{-1}]_k = [f_{d_0}']_k \neq [f_1]_k = [f_1'^{-1}]_k.$$

Thus, $k \in \mathcal{I}(\mathcal{F}')$.

If $k \notin \mathcal{I}(\mathcal{F})$, we have for all $d$,

$$[f_d'^{-1}]_k = [f_d']_k = [f_1]_k = [f_1'^{-1}]_k.$$

Thus $k \notin \mathcal{I}(\mathcal{F}')$.

**2)** When $j \leq k < j'$, we have $\forall d, [f_d'(\boldsymbol{x})]_k = x_k \Rightarrow [f_d'^{-1}(\boldsymbol{x})]_k = x_k$. Thus we have $\forall d$, $[f_d'^{-1}]_k = [f_1'^{-1}]_k$, which means for all $j \leq k < j'$, $k \notin \mathcal{I}(\mathcal{F}')$.

**3)** When $k = j'$, we have $\forall d, [f_d']_{j'} = f_{d,j}(x_1, \cdots, x_{j-1}, x_{j'})$. Furthermore, since, $j \in \mathcal{I}(\mathcal{F})$, we have $\exists d_0, [f_{d_0}]_j \neq [f_1]_j$ by Lemma 8. Thus $[f_{d_0}']_{j'} = [f_{d_0}]_j \neq [f_1]_j = [f_1']_{j'} \Rightarrow j' \in \mathcal{I}(\mathcal{F}')$ also by Lemma 8.

**4)** When $k > j'$, if $k \in \mathcal{I}(\mathcal{F}), \exists d_1, d_2, [f_{d_1}]_k \neq [f_{d_2}]_k$, Chaining with (76), we have $[f_{d_1}']_k \neq [f_{d_2}']_k$. Thus, $k \in \mathcal{I}(\mathcal{F}')$ by Lemma 8. Similarly, if $k \notin \mathcal{I}(\mathcal{F})$, then $k \notin \mathcal{I}(\mathcal{F}')$

To summarize, $\mathcal{I}(\mathcal{F}') = (\mathcal{I}(\mathcal{F}) \setminus \{j\}) \cup \{j'\}$. □

Built upon swapping Lemma, we move to our main result on the existence of equivalent Canonical ILD.

**Theorem 3** (Existence of Equivalent Canonical ILD). *Given an ILD $(g, \mathcal{F})$, there exists a canonical ILD that is both counterfactually and distributionally equivalent to $(g, \mathcal{F})$ while maintaining the size of the intervention set, i.e., $\forall (g, \mathcal{F}), \exists (g', \mathcal{F}') \in \mathcal{C}$ s.t. $(g', \mathcal{F}') \simeq_{C,D} (g, \mathcal{F})$ and $|\mathcal{I}(\mathcal{F})| = |\mathcal{I}(\mathcal{F}')|$.*

*Proof of Theorem 3.* At high level the proof is organized in the following three steps.

**(Step 1)** we use Theorem 1 to construct an equivalent counterfactual $(g^{(0)}, \mathcal{F}^{(0)}) \simeq_C (g, \mathcal{F})$ by choosing two invertible functions $h_1 = f_1^{-1}$ and $h_2 = \text{Id}$. In this way, Theorem 1 implies

$$f_1^{(0)} = h_1 \circ f_1 \circ h_2 = f_1^{-1} \circ f_1 \circ \text{Id} = \text{Id}$$

$$\forall d > 1, \quad f_d^{(0)} = h_1 \circ f_d \circ h_2 = f_1^{-1} \circ f_d \circ \text{Id} = f_1^{-1} \circ f_d, \quad \text{and} \quad g^{(0)} = g \circ h_1^{-1} = g \circ f_1.$$

Equipped with $(g^{(0)}, \mathcal{F}^{(0)})$, we can show that part I of Definition 5 is satisfied, i.e., $f_1^{(0)} = \text{Id}$. Choosing $h_2 = \text{Id}$, we could prove this operation could guarantee the distribution equivalence.

**(Step 2)** we can further construct a series of equivalent counterfactuals iteratively applying Lemma 9 to gradually satisfy part II of Definition 5. Specifically, in this step, we recursively construct, for all iteration $k \in \{1, 2, \ldots, k^{\text{last}}\}$,

$$\mathcal{F}^{(k)} \triangleq h_{j(k) \leftrightarrow j'(k)} \circ \mathcal{F}^{(k-1)} \circ h_{j(k) \leftrightarrow j'(k)},$$

and

$$g^{(k)} \triangleq g^{(k-1)} \circ h_{j(k) \leftrightarrow j'(k)}^{-1} = g^{(k-1)} \circ h_{j(k) \leftrightarrow j'(k)},$$

where $h_{j(k) \leftrightarrow j'(k)}$ denotes swapping the $j(k)$-th and $j'(k)$-th feature values, i.e.,

$$h_{j \leftrightarrow j'}(\boldsymbol{x}) \triangleq [x_1, x_2, \cdots, x_{j-1}, x_{j'}, x_{j+1}, \cdots, x_{j'-1}, x_j, x_{j'+1}, \cdots, x_m]^T, \tag{77}$$

and further define

$$j'(k) \triangleq \max \left\{ j, j \notin \mathcal{I}\left(\mathcal{F}^{(k)}\right) \right\}, \text{ and } j(k) \triangleq \max \left\{ j < j'(k), j \in \mathcal{I}\left(\mathcal{F}^{(k)}\right) \right\}. \tag{78}$$

In high level, at each iteration, we seek the largest index $j'(k)$ which does not lies in the previous intervention set $\mathcal{I}\left(\mathcal{F}^{(k)}\right)$, and swap it with the largest index $j(k)$ which is smaller than $j'(k)$. We terminate at $k$ when $\{j < j'(k), j \in \mathcal{I}\left(\mathcal{F}^{(k)}\right)\} = \emptyset$.

By the definition of $j'(k), j(k)$ in (78), we can show that

**1)** for each swap step $k$, there holds

$$j(k) \in \mathcal{I}\left(\mathcal{F}^{(k)}\right), \text{ and } \forall \tilde{j} : j(k) < \tilde{j} \leq j'(k), \, \tilde{j} \notin \mathcal{I}\left(\mathcal{F}^{(k)}\right), \tag{79}$$

which implies Lemma 9 can be applied to ensure the counterfactual equivalence at each step.

**2)** When meeting the stopping criterion at step $k^{\text{last}}$, i.e.,

$$\left\{ j < j'(k^{\text{last}}), j \in \mathcal{I}\left(\mathcal{F}^{(k^{\text{last}}-1)}\right) \right\} = \emptyset, \tag{80}$$

there holds

$$\forall j \in \mathcal{I}\left(\mathcal{F}^{(k^{\text{last}}-1)}\right), \quad j > m - \left| \mathcal{I}\left(\mathcal{F}^{(k^{\text{last}}-1)}\right) \right|,$$

i.e., $\left(g^{(k^{\text{last}}-1)}, \mathcal{F}^{(k^{\text{last}}-1)}\right)$ is in canonical form. Chaining **1)** and **2)**, we conclude

$$\exists (g', \mathcal{F}') \triangleq \left(g^{k^{\text{last}}-1}, \mathcal{F}^{k^{\text{last}}-1}\right) \in \mathcal{C} \text{ s.t. } (g', \mathcal{F}') \simeq_C (g, \mathcal{F}).$$

Note that $g^{(k)} \circ f_d^{(k)} = g^{(k-1)} \circ f_d^{(k-1)} \circ h_{j(k) \leftrightarrow j'(k)}$, and linear operator $h_{j(k) \leftrightarrow j'(k)}$ is orthogonal, then iteratively, we conclude $(g', \mathcal{F}') \simeq_D (g, \mathcal{F})$.

To prove **1)**, observe in (78), $j(k)$ is the largest index in the intervention set which is smaller than $j'(k)$. This simply implies (79).

To prove **2)**, suppose when meeting the stopping criterion at step $k^{\text{last}}$, there holds

$$\exists j \in \mathcal{I}\left(\mathcal{F}^{(k^{\text{last}}-1)}\right) \text{ such that } j \leq m - \left|\mathcal{I}\left(\mathcal{F}^{(k^{\text{last}}-1)}\right)\right|. \tag{81}$$

It implies that

$$\exists \hat{j} \notin \mathcal{I}\left(\mathcal{F}^{(k^{\text{last}}-1)}\right) \text{ and } \hat{j} \in \left\{m - \left|\mathcal{I}\left(\mathcal{F}^{(k^{\text{last}}-1)}\right)\right| + 1, \ldots, m\right\}.$$

Then we can choose $j'(k) = \hat{j}$, implying $j \in \left\{j < j'(k), j \in \mathcal{I}\left(\mathcal{F}^{(k^{\text{last}}-1)}\right)\right\} \neq \emptyset$, contradict to (80).

**(Step 3)** We use the same techinique as in **Step 1**, where instead we choose $h_1 = f_1$ and $h_2 = \text{Id}$. This concludes the proof of part I in Theorem 3.

It remains to prove that the construction of $f^{(0)}$ in the **step 1** does not change the intervention set.

**1)** For any $j \notin \mathcal{I}(\mathcal{F})$, for any pairs $d, d'$, we have $\left[f_d^{-1}\right]_j = \left[f_{d'}^{-1}\right]_j$, based on the construction of $f^{(0)}$, we have

$$\left[f_d^{(0)^{-1}}\right]_j = [f_d^{-1} \circ f_1]_j = [f_{d'}^{-1} \circ f_1]_j = \left[f_{d'}^{(0)^{-1}}\right]_j \tag{82}$$

thus, $\mathcal{I}(f_d^{(0)}, f_{d'}^{(0)}) \subseteq \mathcal{I}(f_d, f_d')$.

**2)** For any $j \in \mathcal{I}(\mathcal{F})$, there exists $d, d'$ and $z$, such that $[f_d^{-1}(z)]_j \neq [f_{d'}^{-1}(z)]_j$. Note that $f_1$ is a bijective function, there exists $z'$ such that $z = f_1(z')$, we have

$$[f_d^{-1}(z)]_j \neq [f_{d'}^{-1}(z)]_j$$
$$\Leftrightarrow \left[f_d^{-1}(f_1(z'))\right]_j \neq \left[f_{d'}^{-1}(f_1(z'))\right]_j$$
$$\Leftrightarrow \left[f_d^{(0)^{-1}}(z')\right]_j \neq \left[f_{d'}^{(0)^{-1}}(z')\right]_j$$
$$\Leftrightarrow j \in \mathcal{I}\left(f_d^{(0)}, f_{d'}^{(0)}\right)$$

thus $\mathcal{I}\left(f_d^{(0)}, f_{d'}^{(0)}\right) \supset \mathcal{I}(f_d, f_{d'})$. Combining **1)** and **2)**, we have $\mathcal{I}\left(f_d^{(0)}, f_{d'}^{(0)}\right) = \mathcal{I}(f_d, f_{d'})$. This show that the construction of **step 1** and **step 3** do not change the intervention set, combining the fact in **step 1**, we iteratively used swapping Lemma 9, and swapping Lemma 9 does not change the intervention set size, i.e., $\mathcal{I}(\mathcal{F}') = (\mathcal{I}(\mathcal{F}) \setminus \{j\}) \cup \{j'\}$, we conclude that $\left|\mathcal{I}\left(f_d^{(0)}, f_{d'}^{(0)}\right)\right| = |\mathcal{I}(f_d, f_{d'})|$ This completes the proof. $\square$

To help understanding, we design a simple linear ILD model to demonstrate the theorem construction procedure.

**Example 1.** *Suppose we have a $4$-dimensional ILD model $(g, \mathcal{F})$ containing $2$ domains, where*

$$f_1 \triangleq \begin{bmatrix} 1 & 0 & 0 & 0 \\ 1 & 1 & 0 & 0 \\ 1 & 1 & 1 & 0 \\ 1 & 1 & 1 & 1 \end{bmatrix}, f_2 \triangleq \begin{bmatrix} 1 & 0 & 0 & 0 \\ 2 & 2 & 0 & 0 \\ 1 & 1 & 1 & 0 \\ 1 & 1 & 1 & 1 \end{bmatrix}, g \text{ invertible.}$$

Following the proof of Theorem 3, we have Following **Step 1** in the proof of Theorem 3, we have $h_1 = f_1^{-1}$,

$$f_1^{(0)} = f_1^{-1} \circ f_1, f_2^{(0)} = f_1^{-1} \circ f_2, g^{(0)} = g \circ f_1,$$

$$f_1^{(0)} = \begin{bmatrix} 1 & 0 & 0 & 0 \\ 0 & 1 & 0 & 0 \\ 0 & 0 & 1 & 0 \\ 0 & 0 & 0 & 1 \end{bmatrix}, f_2^{(0)} = \begin{bmatrix} 1 & 0 & 0 & 0 \\ 1 & 2 & 0 & 0 \\ -1 & -1 & 1 & 0 \\ 0 & 0 & 0 & 1 \end{bmatrix},$$

$$g^{(0)} = g \circ \begin{bmatrix} 1 & 0 & 0 & 0 \\ 1 & 1 & 0 & 0 \\ 1 & 1 & 1 & 0 \\ 1 & 1 & 1 & 1 \end{bmatrix}$$

Notice that $\mathcal{I}(f^{(0)}) = \{2, 3\}$. Following **Step 2** in the proof of Theorem 3, we first swap $j = 3$ and $j' = 4$,

$$h_{3 \leftrightarrow 4} = \begin{bmatrix} 1 & 0 & 0 & 0 \\ 0 & 1 & 0 & 0 \\ 0 & 0 & 0 & 1 \\ 0 & 0 & 1 & 0 \end{bmatrix}, g^{(1)} = g \circ \begin{bmatrix} 1 & 0 & 0 & 0 \\ 1 & 1 & 0 & 0 \\ 1 & 1 & 1 & 1 \\ 1 & 1 & 1 & 0 \end{bmatrix}$$

We have $f^{(2)} \triangleq h_{3 \leftrightarrow 4} \circ f^{(1)} \circ h_{3 \leftrightarrow 4}$

$$f_1^{(2)} = \begin{bmatrix} 1 & 0 & 0 & 0 \\ 0 & 1 & 0 & 0 \\ 0 & 0 & 1 & 0 \\ 0 & 0 & 0 & 1 \end{bmatrix}, f_2^{(2)} = \begin{bmatrix} 1 & 0 & 0 & 0 \\ 1 & 2 & 0 & 0 \\ 0 & 0 & 1 & 0 \\ -1 & -1 & 0 & 1 \end{bmatrix}.$$

Notice that $\mathcal{I}(f^{(1)}) = \{2, 4\}$. Following **Step 2** in the proof of Theorem 3, we first swap $j = 2$ and $j' = 3$,

$$h_{2 \leftrightarrow 3} = \begin{bmatrix} 1 & 0 & 0 & 0 \\ 0 & 0 & 1 & 0 \\ 0 & 1 & 0 & 0 \\ 0 & 0 & 0 & 1 \end{bmatrix}, g^{(2)} = g \circ \begin{bmatrix} 1 & 0 & 0 & 0 \\ 1 & 1 & 1 & 1 \\ 1 & 1 & 0 & 0 \\ 1 & 1 & 1 & 0 \end{bmatrix}$$

We have $f^{(3)} \triangleq h_{2 \leftrightarrow 3} \circ f^{(2)} \circ h_{2 \leftrightarrow 3}$

$$f_1^{(2)} = \begin{bmatrix} 1 & 0 & 0 & 0 \\ 0 & 1 & 0 & 0 \\ 0 & 0 & 1 & 0 \\ 0 & 0 & 0 & 1 \end{bmatrix}, f_2^{(2)} = \begin{bmatrix} 1 & 0 & 0 & 0 \\ 0 & 1 & 0 & 0 \\ 1 & 0 & 2 & 0 \\ -1 & 0 & -1 & 1 \end{bmatrix}.$$

$$g^{(2)} = g \circ \begin{bmatrix} 1 & 0 & 0 & 0 \\ 1 & 1 & 1 & 1 \\ 1 & 1 & 0 & 0 \\ 1 & 1 & 1 & 0 \end{bmatrix}$$

Then we follow **Step 3**, i.e.,

$$f_1^{(3)} = f_1 \circ f_1^{(2)}, f_2^{(3)} = f_1 \circ f_2^{(2)}, g^{(3)} = g^2 \circ f_1^{-1},$$

$$f_1^{(3)} = \begin{bmatrix} 1 & 0 & 0 & 0 \\ 1 & 1 & 0 & 0 \\ 1 & 1 & 1 & 0 \\ 1 & 1 & 1 & 1 \end{bmatrix}, f_2^{(3)} = \begin{bmatrix} 1 & 0 & 0 & 0 \\ 1 & 1 & 0 & 0 \\ 2 & 1 & 2 & 0 \\ 1 & 1 & 1 & 1 \end{bmatrix}.$$

Notice that $(g^{(2)}, f^{(2)})$ is the Identity Canonical form, and $(g^{(3)}, f^{(3)})$ is in general canonical form. They are counterfactually equivalent to each other by checking definition.

## C SIMULATED EXPERIMENTS

### C.1 EXPERIMENT DETAILS

**Dataset** The ground truth latent SCM $f_d^* \in \mathcal{F}_{IA}$ takes the form $f_d^*(\epsilon) = F_d^* \, \epsilon + b_d^* \mathbb{1}_{\mathcal{I}}$ where $F_d^* = (I - L_d^*)^{-1}, L_d^* \in \mathbb{R}^{m \times m}$ is domain-specific lower triangular matrix that satisfies sparsity constraint, $b_d^* \in \mathbb{R}$ is a domain-specific bias and $\mathbb{1}_{\mathcal{I}}$ is an indicator vector where any entries corresponding to the intervention set are 1. To be specific, $[L_d^*]_{i,j} \sim \mathcal{N}(0, 1)$ and $b_d^* \sim \text{Uniform}(-2\sqrt{m/|\mathcal{I}|}, 2\sqrt{m/|\mathcal{I}|})$. The observation function takes the form $g^*(x) = G^* \, \text{LeakyReLU}(x)$ where $G^* \in \mathbb{R}^{m \times m}$ and the slope of LeakyReLU is $0.5$. To allow for similar scaling across problem settings, we set the determinant of $G^*$ to be 1 and standardize the intermediate output of the LeakyReLU. The generated $F_d^*, b_d^*, G^*$ all vary with random seeds and all experiments are repeated for 10 different seeds. We generate 100,000 samples from each domain for the training set and 1,000 samples from each domain in the validation and test set.

**Model**   We test with two ILD models: *ILD-Can* as introduced in Section 3.3 and a baseline model, *ILD-Dense* which has no sparsity restrictions on its latent SCM. To be specific, the latent SCM of *ILD-Dense* could be any model in $\mathcal{F}_{IA}$. We use $\mathcal{I}$ and $\mathcal{I}^*$ to represent the intervention set of the model and dataset, respectively. We note that for *ILD-Dense*, $\mathcal{I}$ contains all nodes and for *ILD-Can*, $\mathcal{I}$ contains only the last few nodes. Both models follow a similar structure as the ground truth. To be specific, the latent SCM takes the form $f_d(\boldsymbol{\epsilon}) = F_d\,\boldsymbol{\epsilon} + \boldsymbol{b}_d$ where $F_d = (I - L_d)^{-1}S_d$, $L_d \in \mathbb{R}^{m \times m}$, $S_d \in \mathbb{R}^{m \times m}$, and $\boldsymbol{b}_d \in \mathbb{R}^m$. The observation takes the form $g(\boldsymbol{x}) = G\,\text{LeakyReLU}\,(\boldsymbol{x}) + \boldsymbol{b}$ where $G \in \mathbb{R}^{m \times m}$, $\boldsymbol{b} \in \mathbb{R}^m$, and the slope of LeakyReLU is 0.5.

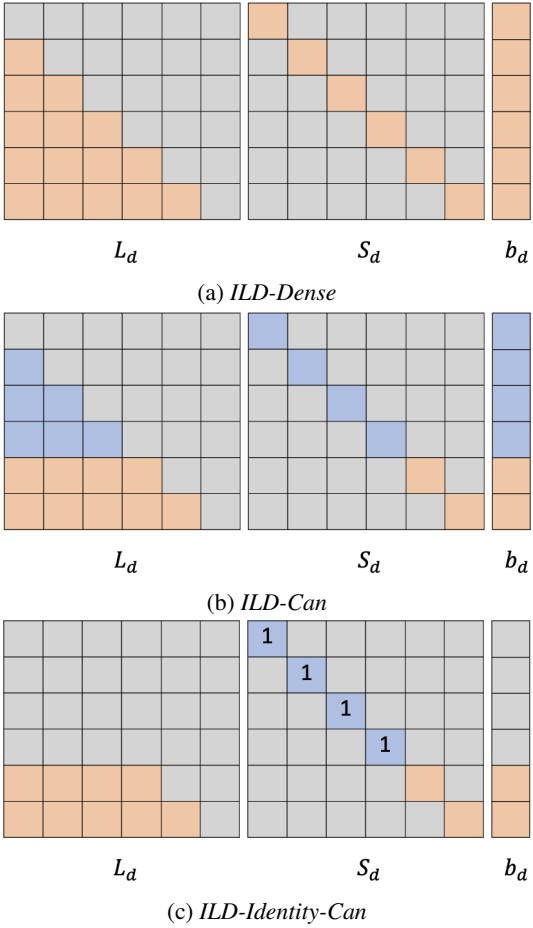

(a) *ILD-Dense*

(b) *ILD-Can*

(c) *ILD-Identity-Can*

Figure 3: An illustration of the matrices/vector used to create $f_d$ across the three ILD models when $m = 6$ and $|\mathcal{I}| = 2$. These are used such that $f_d(\boldsymbol{\epsilon}) = F_d\,\boldsymbol{\epsilon} + \boldsymbol{b}_d$ where $F_d = (I - L_d)^{-1}S_d$. The grey elements are 0, the orange elements are parameters that are different for different domains, and the blue elements are parameters shared across domains. We specify the value if it is a fixed number other than 0. Note that we don't implement *ILD-Identity-Can* in our experiments. We include it here only for illustration of our theory.

In Figure 3a and Figure 3b, we add an illustration of the latent SCM for *ILD-Dense* and *ILD-Can* respectively. We emphasize a few main differences between the dataset and models here: (1) For *ILD-Can*, $\mathcal{I}$ only contains the last few nodes while for the dataset while $\mathcal{I}^*$ could contain any node we specify. We note that *ILD-Dense* is equivalent to a *ILD-Can* with all nodes in its intervention set. (2) There is no constraint on the determinant of $G$ and standardization in $g(\boldsymbol{x})$. (3) The bias added to all dimensions in the ground truth model is the same scalar value, but the bias in the model is allowed to vary for each axis. (4) In the model, $g$ is allowed a learnable bias.

**Metric**   To evaluate the models, we compute the mean square error between the estimated counterfactual and ground truth counterfactual, i.e. Error $= \frac{2}{N_d(N_d-1)}\sum_{d'\neq d}\sum_d \|g^* \circ f_{d'}^* \circ (f_d^*)^{-1} \circ$

$(g^*)^{-1}(\boldsymbol{x}_d) - g \circ f_{d'} \circ f_d^{-1} \circ g^{-1}(\boldsymbol{x}_d)\|^2$. As in practice, we can only check data fitting instead of counterfactual estimation, and we report the counterfactual error computed with the test dataset when the likelihood computed with the validation set is highest.

**Training details** We use Adam optimizer for both $f$ and $g$ with a lr $= 0.001, \beta_1 = 0.5, \beta_2 = 0.999$, and a batch size of is 500. We run all experiments for 50,000 iterations and compute validation likelihood and test counterfactual error every 100 steps. $f$ is randomly initialized. Regarding $g$, $G$ is initialized as an identity matrix and $\boldsymbol{b}$ is initialized as $\boldsymbol{0}$.

## C.2 ADDITIONAL SIMULATED EXPERIMENT RESULTS

For better organization here, we split our experiment into three cases as introduced below. The first two cases point to the question: given the fact that we use the correct sparsity, does sparse canonical form model designing provide benefits in generating domain counterfactuals? The third case investigates the more practical scenario where we don't have any knowledge of the ground truth sparsity and we explore what would be a better model design practice in this case.

**Case 0: Exact match between dataset and models** In this section, we investigate the performance of *ILD-Dense* and *ILD-Can* while assuming that the ground truth intervention set only contains the last few nodes and we choose the correct size of the intervention set.

To understand how the true intervention set affects the gap between *ILD-Dense* and *ILD-Can*, we varied the size of the ground truth intervention. In Figure 4, we observe that the performance gap tends to be largest when the true intervention set is the most sparse and the performance of *ILD-Can* approaches to the performance of *ILD-Dense* as we increase the size. This makes sense as *ILD-Can* is a subset of *ILD-Dense* and they are equivalent when $\mathcal{I} = \{1, 2, 3, 4, 5, 6\}$. Additionally, even when the ground truth model is relatively dense (when $|\mathcal{I}^*|$ is close to $m$), *ILD-Can* is still better than *ILD-Dense*. Then we test how our algorithm scales with dimension when the number of domains is different. In Figure 5, we notice that *ILD-Can* is significantly better than *ILD-Dense* in 9 out of 12 cases. In the next paragraphs, we further investigate the 3 cases that do not outperform *ILD-Dense* to understand if it seems to be a theoretic or algorithmic/optimization problem.

We take a further investigation on the three cases where *ILD-Can* is close to or worse than *ILD-Dense*. As shown in Figure 6, when the latent dimension is 10 and the number of domains is 2, i.e. $m = 10$ and $N_d = 2$, the validation likelihood of *ILD-Can* is much lower than *ILD-Dense* especially in comparison to that with $m = 4, 6$. We conjecture that the performance drop in terms of counterfactual error could be a result of the worse data fitting, i.e., the model does not fit the data well in terms of log-likelihood. As further evidence, we show the counterfactual error and corresponding validation log-likelihood in Table 3. We observe that the log-likelihood of *ILD-Dense* tends to be much lower when it has a larger counterfactual error than that of *ILD-Dense*. As for the relatively worse performance of *ILD-Can* when $m = 4, N_d = 2$ and $m = 4, N_d = 3$, we report the counterfactual error corresponding to each seed in Table 4 and Table 5 respectively. When the latent dimension is 4 and the number of domains is 2, i.e., $m = 4, N_d = 2$, *ILD-Can* is better than *ILD-Dense* with 9 out of 10 seeds. However, it fails significantly with seed 0 and thus leads to a larger average of counterfactual error. When $m = 4, N_d = 3$, *ILD-Can* is better than *ILD-Dense* with 7 out of 10 seeds but *ILD-Can* is not significantly better than *ILD-Dense* in terms of average error. We think this is more likely an optimization issue with lower dimensions, which is not explored by our theory. We conjecture that larger models with smoother optimization landscapes will perform better as we see in the imaged-based experiments. We also note that these models are not significantly overparametrized and thus may not benefit from the traditional overparameterization that aids the performance of deep learning in many cases. Further investigation into overparameterized models may alleviate this algorithmic issue.

Despite some corner cases in which the optimization landscape may be difficult for these simple models, all the results point to the same trend that the sparse constraint and canonical form motivated by our theoretic derivation indeed aids in counterfactual performance—despite not explicitly training for counterfactual performance.

**Case 1: Correct $|\mathcal{I}|$ but mismatched intervention indices** In this section, we include more results in the more practical scenario where we choose the correct number of the intervened nodes

but they are not necessarily the last few nodes in the latent SCM. This experiment is related to our canonical ILD theory, i.e., that there exists a canonical counterfactual model (where the intervened nodes are the last ones) corresponding to any true non-canonical ILD that has the same sparsity. As a starting point, we first illustrate the existence of a canonical model we try to find in Figure 10.

To investigate the effect of different indices of the intervened nodes, in Figure 7, we change the true intervention set $\mathcal{I}^*$ while keeping the number of intervened nodes $|\mathcal{I}^*|$ the same. We observe that *ILD-Can* is consistently better than *ILD-Dense* regardless of which nodes are intervened except for one case. When the number of domains is 2 and $\mathcal{I}^* = \{4, 5\}$, we find the gap is much smaller mainly because *ILD-Can* fails to fit the observed distribution in one case as shown in Table 6. We then test the effect of the number of domains with different latent dimensions in Figure 8. We observe that our model performs consistently well with different numbers of domains and latent dimensions. In Figure 9, we visualize how *ILD-Can* leads to a lower counterfactual error in comparison to *ILD-Dense*. As shown in Figure 9a and Figure 9b, *ILD-Can* clearly does better in counterfactual estimation. In Figure 9c and Figure 9d, both of them have a relatively larger error. However, *ILD-Can* tends to find a closer solution while *ILD-Dense* matches distribution more randomly. This could result from the large search space of *ILD-Dense* and it can easily encodes a transformation such as rotation which will not hurt distribution fitting but will lead to a significant counterfactual error.

Even though we do not know the specific nodes being intervened on, similar to Case 1, we show that sparse constraint leads to better counterfactual estimation.

**Case 2: Intervention set size mismatch**  In this section, we include more results in the most difficult cases where we have no knowledge of the dataset. To investigate what will happen if there is a mismatch of the number of intervened nodes between the true model and the approximation, i.e., $|\mathcal{I}| \neq |\mathcal{I}^*|$, we first change $\mathcal{I}^*$ while keeping the model unchanged, i.e., $\mathcal{I}$ is fixed. As shown in Figure 11, the performance gap between *ILD-Can* and *ILD-Dense* become smaller as the dataset becomes less sparse while *ILD-Can* outperforms *ILD-Dense* in all cases. We then change $\mathcal{I}$ while keeping $\mathcal{I}^*$ unchanged. As shown in Figure 12, the performance of *ILD-Can* approaches to that of *ILD-Dense* as we increase $|\mathcal{I}|$. A somewhat surprising result is that *ILD-Can* has the lowest counterfactual error when $|\mathcal{I}| = 1$. However, as we check data fitting in Figure 13, we can tell *ILD-Can* fails to fit the observed distribution in this case. We conjecture the main reason for this is that our theory does not guarantee the existence of a distributionally and counterfactually equivalent canonical model in those cases as we are using a model that is more sparse than the ground truth dataset. Hence, we cannot rely on the counterfactual estimation when the observed distribution is not fitted.

In summary, we observe that *ILD-Can* always tends to get a lower counterfactual error even though we choose a wrong size of intervention set, i.e. $|\mathcal{I}| \neq |\mathcal{I}^*|$. However, we also observe that in the cases where our model is more sparse than ground truth, the data fitting performance of *ILD-Can* would drop more significantly. We believe this could also be a good indicator of whether we find a reasonable $|\mathcal{I}|$.

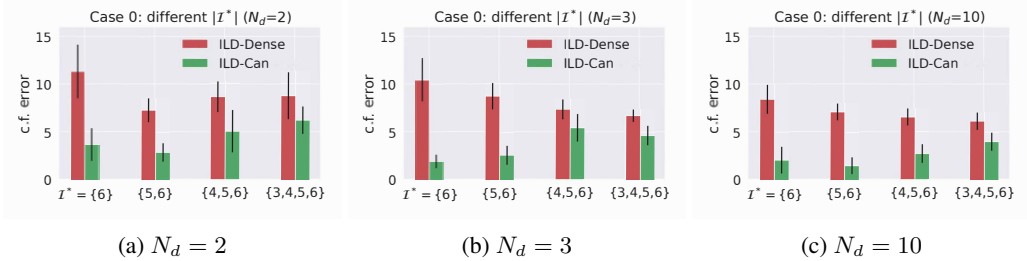

(a) $N_d = 2$        (b) $N_d = 3$        (c) $N_d = 10$

Figure 4: Case 0: Test counterfactual error with different $\mathcal{I}^*$. To understand how the true intervention set affects the gap between *ILD-Dense* and *ILD-Can*, we varied the size of the ground truth intervention. It can be observed that the performance gap tends to be largest when the true intervention set is the sparsest and the performance of *ILD-Can* approaches to the performance of *ILD-Dense* as we increase the size.

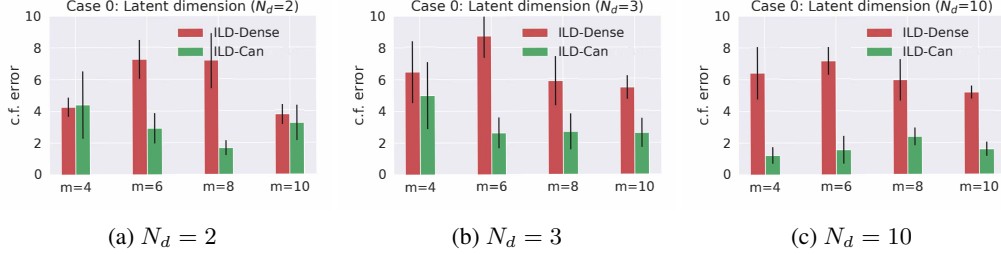

(a) $N_d = 2$        (b) $N_d = 3$        (c) $N_d = 10$

Figure 5: Case 0: Test counterfactual error with different dimension. We investigate how our algorithm scales with dimension. We observe that *ILD-Can* is significantly better than *ILD-Dense* in 9 out of 12 cases, and we also notice that there 3 cases where their performance is close to that of each other. Here the intervention set contains the last two nodes. For example, when $m = 4$, $\mathcal{I} = \{3, 4\}$, and when $m = 10$, $\mathcal{I} = \{9, 10\}$.

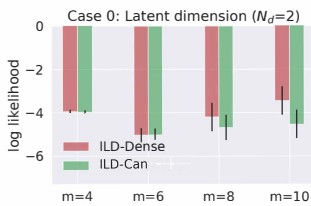

Figure 6: Case 0: Lowest validation log likelihood (same as when we report the test counterfactual error) when testing different dimension with $N_d = 2$. We observe that the likelihood gap between *ILD-Can* and *ILD-Dense* is largest when $m = 10$.

Table 3: Case 0: Test counterfactual error and validation log likelihood for each seed when $m = 10, N_d = 2$. We observe that the log likelihood of *ILD-Dense* tends to be much lower when it has a larger counterfactual error than that of *ILD-Dense*.

|  | Seed | 0 | 1 | 2 | 3 | 4 | 5 | 6 | 7 | 8 | 9 |
|---|---|---|---|---|---|---|---|---|---|---|---|
| Counterfactual error | *ILD-Can* | 4.625 | 0.111 | 0.120 | 0.072 | 4.572 | 10.617 | 4.360 | 6.809 | 0.099 | 0.479 |
|  | *ILD-Dense* | 23.821 | 0.611 | 2.178 | 5.823 | 4.779 | 0.694 | 0.487 | 1.653 | 3.170 | 6.365 |
| Log likelihood | *ILD-Can* | -6.873 | -7.066 | -5.672 | -4.637 | -0.572 | -3.261 | -6.062 | -4.552 | -1.367 | -5.170 |
|  | *ILD-Dense* | -4.034 | -6.434 | -5.679 | -4.197 | 0.711 | -1.908 | -4.180 | -2.413 | -1.483 | -4.796 |

Table 4: Case 0: Test counterfactual error for each seed when $m = 4, N_d = 2$. *ILD-Can* is better than *ILD-Dense* except when seed is 0. However, there is a significant failure for *ILD-Can* with seed 0.

| Seed | 0 | 1 | 2 | 3 | 4 | 5 | 6 | 7 | 8 | 9 |
|---|---|---|---|---|---|---|---|---|---|---|
| *ILD-Can* | 23.790 | 2.309 | 1.747 | 3.180 | 1.265 | 0.864 | 0.779 | 0.227 | 3.325 | 6.362 |
| *ILD-Dense* | 3.321 | 3.435 | 2.838 | 4.209 | 5.356 | 6.456 | 1.615 | 2.165 | 5.195 | 7.937 |

Table 5: Case 0: Test counterfactual error for each seed when $m = 4, N_d = 3$. *ILD-Can* is better than *ILD-Dense* with seed $1, 2, 3, 5, 6, 7, 8$.

| Seed | 0 | 1 | 2 | 3 | 4 | 5 | 6 | 7 | 8 | 9 |
|---|---|---|---|---|---|---|---|---|---|---|
| *ILD-Can* | 23.821 | 0.611 | 2.178 | 5.823 | 4.779 | 0.694 | 0.487 | 1.653 | 3.170 | 6.365 |
| *ILD-Dense* | 24.472 | 3.658 | 2.925 | 5.785 | 3.260 | 5.795 | 3.878 | 4.560 | 4.009 | 5.965 |

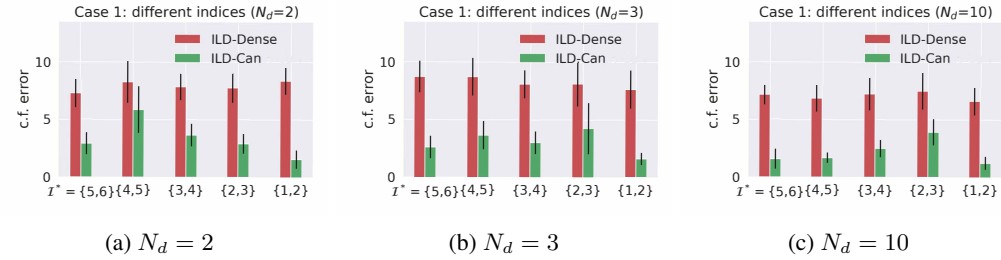

(a) $N_d = 2$      (b) $N_d = 3$      (c) $N_d = 10$

Figure 7: Case 1: Test counterfactual error with different indices. Here we observe that *ILD-Can* performs consistently better than *ILD-Dense*. When $N_d = 2$ and $\mathcal{I} = \{4, 5\}$, the performance of *ILD-Can* gets relatively higher because it fails significantly in one case as shown in Table 6.

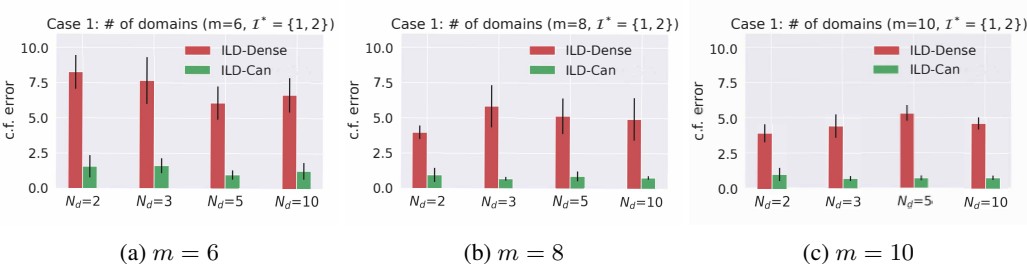

(a) $m = 6$      (b) $m = 8$      (c) $m = 10$

Figure 8: Case 1: Test counterfactual error with different number of domains when $\mathcal{I} = \{1, 2\}$. *ILD-Can* performs consistently well with different number of domains and latent dimension.

Table 6: Case 1: Test counterfactual error and validation log likelihood for each seed when $N_d = 2$ and $\mathcal{I} = \{4, 5\}$. When seed is 5, the error of *ILD-Can* is much larger than that of *ILD-Dense*. In the meanwhile, we notice that the log likelihood of *ILD-Can* is much lower than that of *ILD-Dense* which indicates *ILD-Can* fails to fit the observed distribution well. When seed is 6, there is also a gap in log likelihood. But both models perform very badly in terms of counterfactual error in this case, and we conjecture this results from a very hard dataset.

| | Seed | 0 | 1 | 2 | 3 | 4 | 5 | 6 | 7 | 8 | 9 |
|---|---|---|---|---|---|---|---|---|---|---|---|
| Counterfactual error | *ILD-Can* | 1.395 | 0.862 | 1.338 | 0.193 | 7.557 | 12.422 | 21.762 | 3.879 | 2.352 | 0.479 |
| | *ILD-Dense* | 8.610 | 5.979 | 4.134 | 2.983 | 9.795 | 4.719 | 24.232 | 5.327 | 8.497 | 8.500 |
| Log likelihood | *ILD-Can* | -4.441 | -5.737 | -4.448 | -5.504 | -4.393 | -3.376 | -5.187 | -5.073 | -4.033 | -4.102 |
| | *ILD-Dense* | -4.170 | -5.632 | -4.316 | -5.458 | -4.174 | -2.181 | -4.052 | -5.010 | -5.270 | -4.302 |

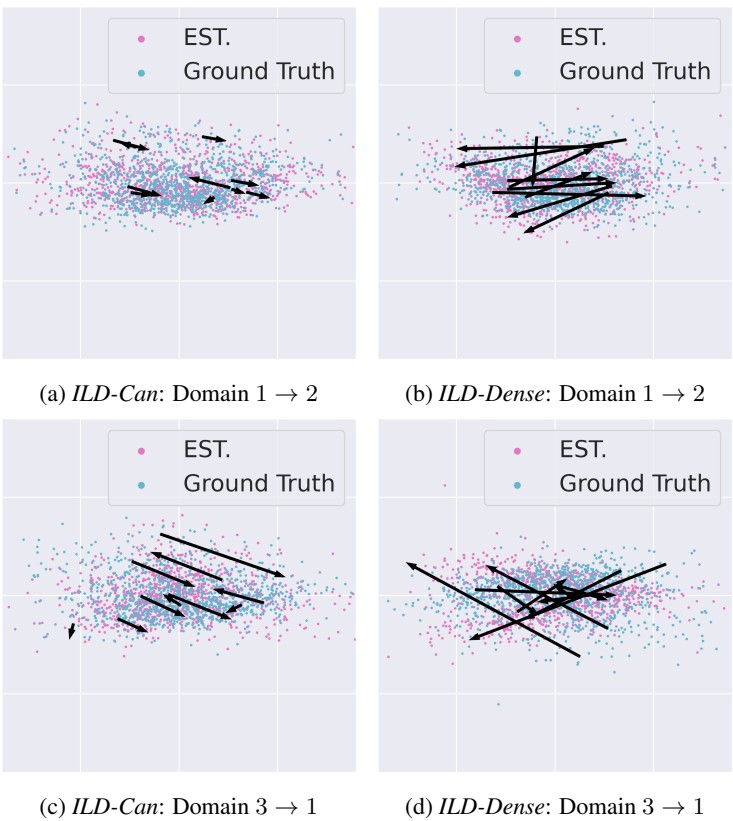

(a) *ILD-Can*: Domain $1 \rightarrow 2$   (b) *ILD-Dense*: Domain $1 \rightarrow 2$

(c) *ILD-Can*: Domain $3 \rightarrow 1$   (d) *ILD-Dense*: Domain $3 \rightarrow 1$

Figure 9: Visualization of counterfactual error when $m = 6, N_d = 3, |\mathcal{I}| = 2, \mathcal{I}^* = \{1, 2\}$. In each plot, we find the first two principle components and project the data along that direction. We select 10 points, then find the corresponding ground truth counterfactual and estimated counterfactual. The black arrow points from ground truth to estimated counterfactual.

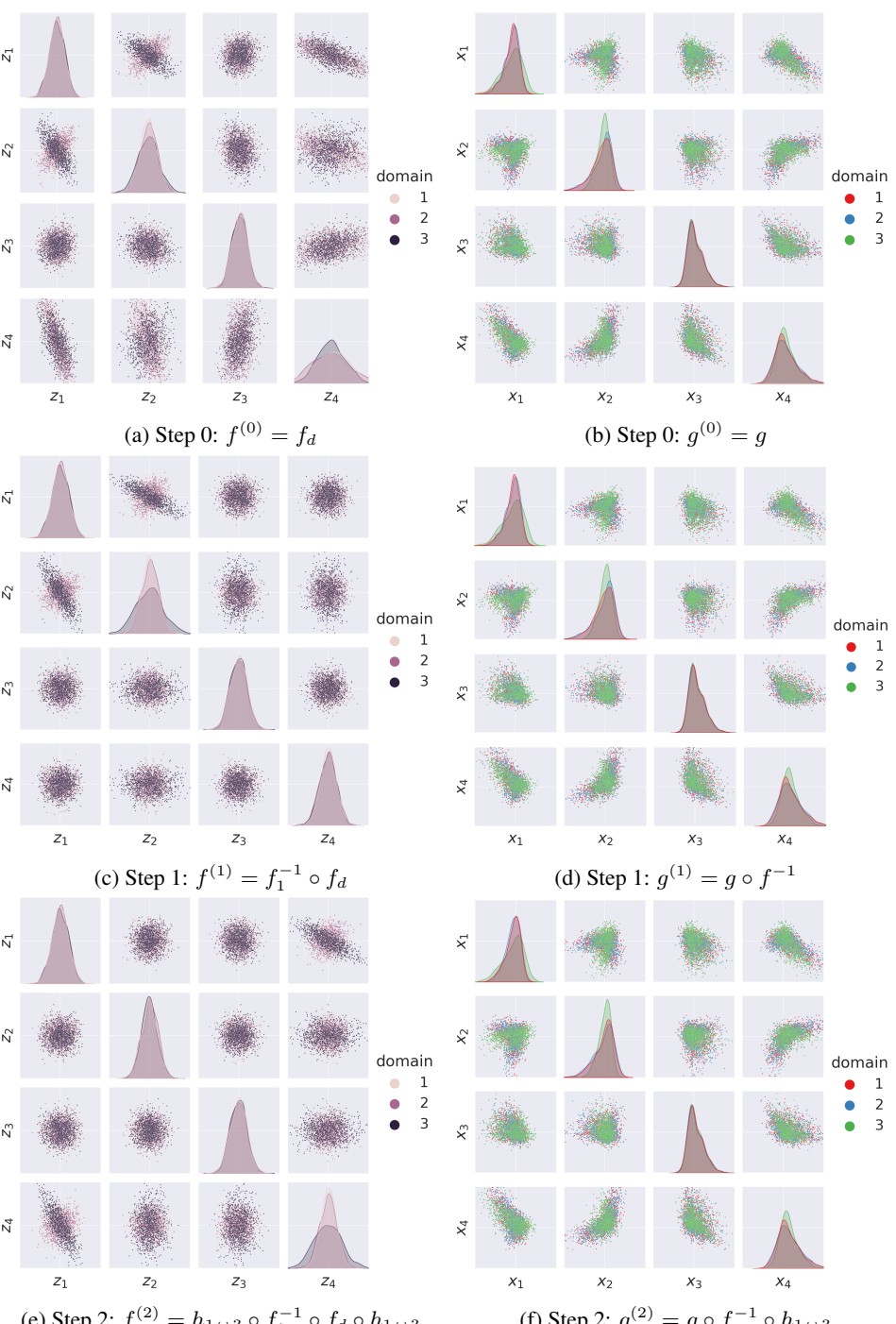

Figure 10: An illustration of the existence of a distributionally and counterfactually equivalent model in canonical form when $m = 4$ and $\mathcal{I} = \{2\}$. $h_{1\leftrightarrow3}$ represents a swapping matrix. $g^{(2)} \circ f^{(2)}$ is one of the caonical model we try to find. Note that the observed distributions in the right column are always the same while the latent distributions on the left change. In particular, the canonical ILD model on the bottom left has independent distributions for the first three variables and is only the non-identity on the last node.

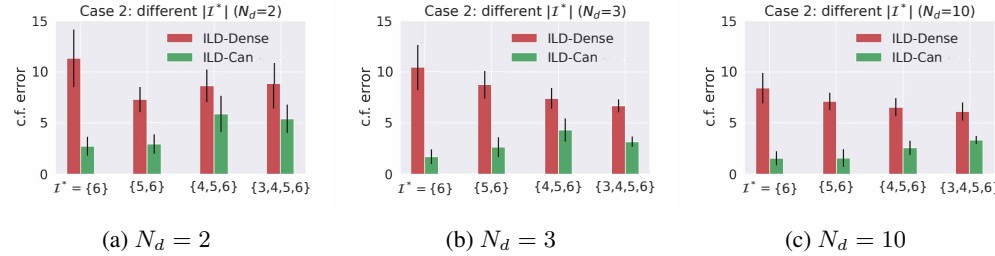

Figure 11: Case 2: Test counterfactual error with different $|\mathcal{I}^*|$ and fixed $|\mathcal{I}| = 2$. The performance of *ILD-Can* gets worse as the dataset becomes less sparse. But it is still better than *ILD-Dense*. Note that when $|\mathcal{I}| = 2$ and $\mathcal{I}^* = \{6\}$, the ground truth canonical model is still a subset of the models we search over.

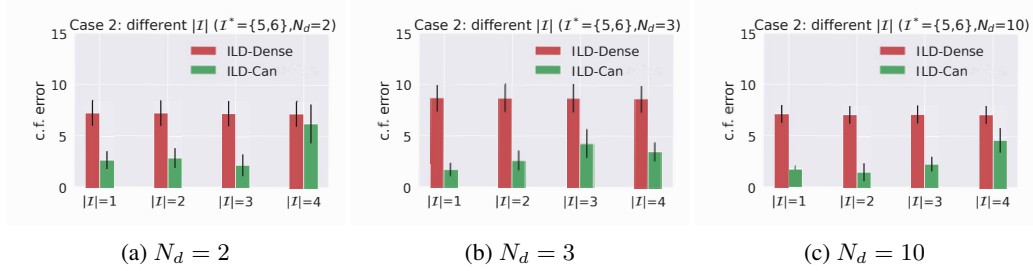

Figure 12: Case 2: Test counterfactual error with different $|\mathcal{I}|$ and fixed $\mathcal{I}^*$. The performance of *ILD-Can* approaches to that of *ILD-Dense* as we increase $|\mathcal{I}|$.

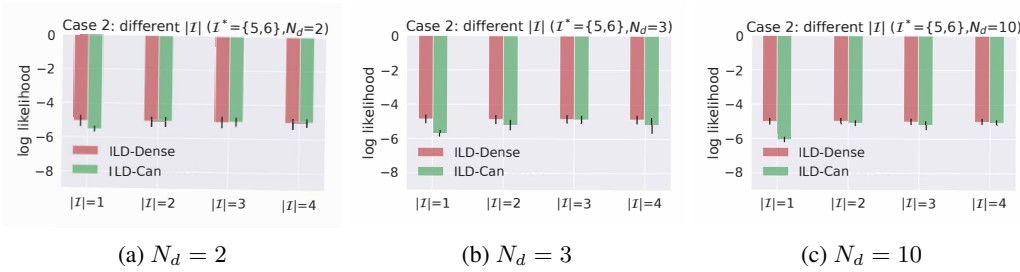

Figure 13: Case 2: Lowest validation log likelihood with different $|\mathcal{I}|$ and fixed $\mathcal{I}^*$. When $|\mathcal{I}| = 1$, there is a more significant gap between *ILD-Can* and *ILD-Dense* with all $N_d$ which indicates *ILD-Can* might fail to fit the observed distribution.

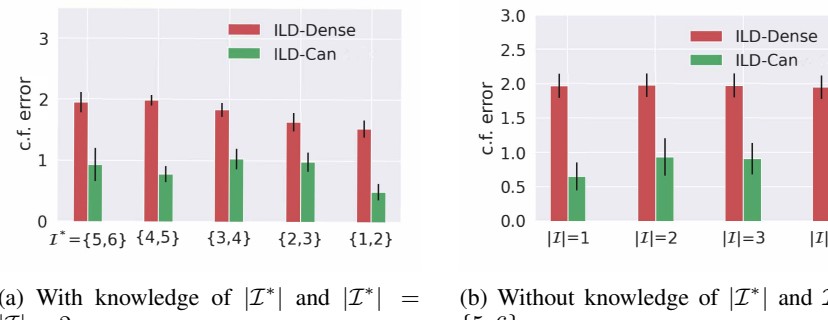

(a) With knowledge of $|\mathcal{I}^*|$ and $|\mathcal{I}^*| = |\mathcal{I}| = 2$

(b) Without knowledge of $|\mathcal{I}^*|$ and $\mathcal{I}^* = \{5, 6\}$

Figure 14: Simulated experiment results (latent dimension $m = 6$, number of domains $N_d = 3$) where observation function of both the model and ground truth $g$ are implemented based on RealNVP. The model $g$ consists of 4 affine coupling layers and the ground truth $g$ consists of 4 affine coupling layers. Results are averaged over 10 runs with different ground truth SCMs and the error bar represents the standard error.

## C.3 SIMULATED EXPERIMENT WITH MORE COMPLICATED STRUCTURES

As an extended study of our simulated experiment, we implement a more complicated observation function for the ground truth $g^*$. We build $g^*$ based on RealNVP (Dinh et al., 2016), where $g^*$ is composed of four sequences of affine coupling layers followed by permutation. Following the notation used in Appendix C.1, the ground truth ILD now takes the form $g \circ f(\epsilon) = \text{Permutation}(\text{AffineCoupling}(\ldots \text{Permutation}(\text{AffineCoupling}(\text{LeakyReLU}(\boldsymbol{x}) + \boldsymbol{b}))\ldots))$.

We first investigate the setting where our model $g$ exactly matches the ground truth ILD, i.e. we use the same number of affine coupling layers for the ground truth $g^*$ and our model $g$. In Figure 14a, we observe that, similar to Figure 1a, *ILD-Can* outperforms *ILD-Dense* no matter which nodes are intervened when $|\mathcal{I}| = |\mathcal{I}^*|$. We then test our model in the setting where $|\mathcal{I}| \neq |\mathcal{I}^*|$, and as shown in Figure 14b, the performance of *ILD-Can* drops as we $|\mathcal{I}|$ grows larger than $|\mathcal{I}^*|$, but it is always better than *ILD-Dense*. This trend is similar to what we observed in Figure 1b. We then test when the *ILD-Can* architecture does not match *ILD-Dense*. To do this, we increased the number of affine coupling layers in the model $g$ to be 8, and in Figure 15a and Figure 15b, we observe similar trends when comparing *ILD-Dense* and *ILD-Can* in all cases. This gives evidence that *ILD-Can* helps with domain counterfactual estimation even when the structures of the model and ground truth do not exactly match.

To further our investigation with mismatching models, we set $g$ to be a VAE-based model composed of three fully connected layers while keeping the ground truth the same as above. To keep the latent dimension relatively comparable, we set the ground truth dimension to be 10 and the latent dimension of VAE to be 6. In Figure 16a and Figure 16b, we observe a similar trend as that with flow model. In Figure 17, only when $|\mathcal{I}| < |\mathcal{I}^*|$, there is a significant difference in the pursuit of distribution equivalence. This again supports our finding that the distribution equivalence performance could be a good indicator of choosing $|\mathcal{I}|$ in practice.

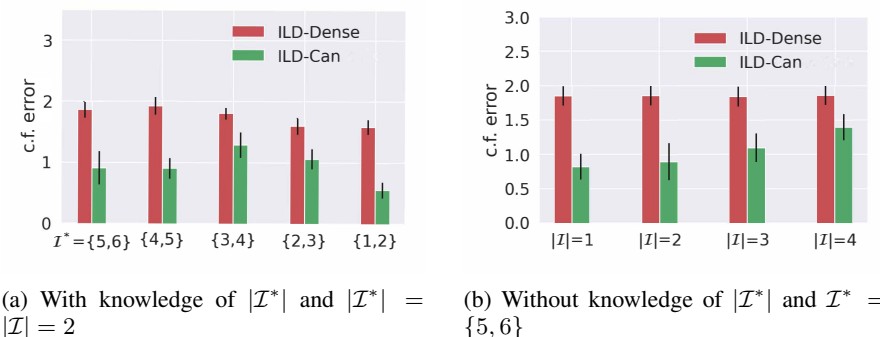

(a) With knowledge of $|\mathcal{I}^*|$ and $|\mathcal{I}^*| = |\mathcal{I}| = 2$

(b) Without knowledge of $|\mathcal{I}^*|$ and $\mathcal{I}^* = \{5, 6\}$

Figure 15: Simulated experiment results (latent dimension $m = 6$, number of domains $N_d = 3$) where the observation function of both the model and ground truth $g$ are implemented based on RealNVP. The model $g$ consists of 8 affine coupling layers and the ground truth $g$ consists of 4 affine coupling layers. Results are averaged over 10 runs with different ground truth SCMs and the error bar represents the standard error.

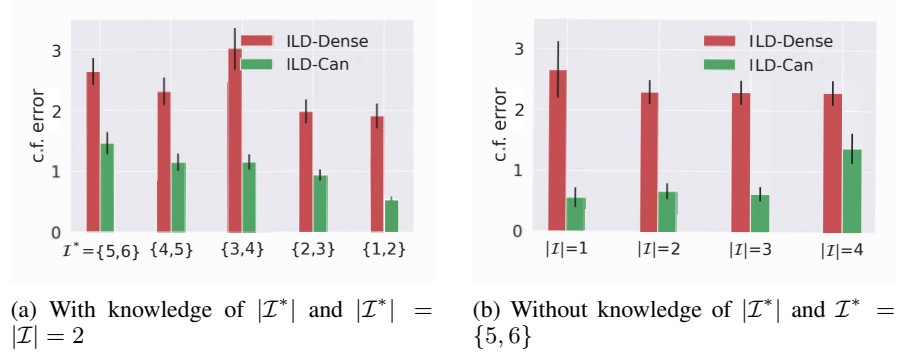

(a) With knowledge of $|\mathcal{I}^*|$ and $|\mathcal{I}^*| = |\mathcal{I}| = 2$

(b) Without knowledge of $|\mathcal{I}^*|$ and $\mathcal{I}^* = \{5, 6\}$

Figure 16: Simulated experiment results (in the ground truth latent dimension $m = 10$ and in the VAE model $m = 6$, number of domains $N_d = 3$) where the observation function of the ground truth $g$ consists of 4 affine coupling layers and the observation function of the model is a VAE composed of three fully connected layers. Results are averaged over 10 runs with different ground truth SCMs and the error bar represents the standard error.

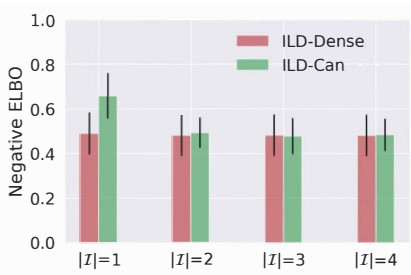

Figure 17: Lowest validation negative ELBO (in the ground truth latent dimension $m = 10$ and in the VAE model $m = 6$, number of domains $N_d = 3$) without knowledge of $|\mathcal{I}^*|$ and $\mathcal{I}^* = \{8, 9\}$. Results are averaged over 10 runs with different ground truth SCMs and the error bar represents the standard error. The number here corresponds to the validation negative ELBO of the checkpoints we use to report test counterfactual error.

# D IMAGE COUNTERFACTUAL EXPERIMENTS

## D.1 DATASET DESCRIPTIONS

**Rotated MNIST and FashionMNIST**  We split the MNIST trainset into 90% training data, 10% validation, and for testing we use the MNIST test set. Within each dataset, we create the domain-specific data by replicating all samples and applying a fixed $\theta_d$ counterclockwise rotation to within that domain. Specifically we generate data from 5 domains by applying rotation of $0°, 15°, 30°, 45°, 60°$. For Rotated FashionMNIST, we use the same setup as the RMNIST setup, except we used the Fashion MNIST (Xiao et al., 2017) dataset. This dataset is structured similar to the MNIST dataset but is designed to require more complex modeling (Xiao et al., 2017).

**3D Shapes**  This is a dataset of 3D shapes that are procedurally generated from 6 independent latent factors: floor hue, wall hue, object hue, scale, shape, and orientation (Burgess and Kim, 2018). In our experiment, we only choose samples with one fixed scale. We then split the four object shapes into four separate domains and set the 10 object colors as the class label. The causal graph for this dataset can be seen in Fig. 18c, and following this, we should expect to see only the object shape change when the domain is changed. Similar to the RMNIST experiment, we use 90% of the samples for training and 10% of the samples for validation.

**Color Rotated MNIST (CRMNIST)**  This is an extension of the RMNIST dataset which introduces a latent color variable whose parents are the latent domain-specific variable and latent class variable. Similar to RMNIST, the latent domain variable corresponds to the rotation of the given digit, except here $d_1 = 0°$ rotation, $d_2 = 90°$ rotation, and the class labels are restricted to digits $y \in \{0, 1, 2\}$. For each sample, there is a 50% chance the color is determined by the combination of class and digit label and a 50% chance the color is randomly chosen. For example if $\epsilon \sim \mathcal{N}(0, 1)$,

$$f_{z_c}(y, d, \epsilon) = \begin{cases} \text{red,} & \text{if } y = 0, d = 1, \epsilon > 0 \\ \text{green,} & \text{if } y = 0, d = 2, \epsilon > 0 \\ \text{blue,} & \text{if } y = 1, d = 1, \epsilon > 0 \\ \dots \\ \text{Random(red, green, blue, yellow, cyan, pink),} & \text{if } \epsilon < 0 \end{cases}$$

The causal graph for this dataset can be seen in Fig. 18b. Similar to the RMNIST experiment, we use 90% of the samples for training and 10% of the samples for validation.

**Causal3DIdent Dataset**  This is a benchmark dataset from (Von Kügelgen et al., 2021) which contains rendered images of realistic 3d objects on a colored background that contain hallmarks of natural environments (e.g. shadows, different lighting conditions, etc.) which are generated via a causal graph imposed over latent variables (the causal graph can be seen in Figure 18d). Similar to (Von Kügelgen et al., 2021), we chose the shape of the 3D object to be the class label, and we defined the background color as the domain label. In the original dataset, the range of the background hue was $\left[\frac{-\pi}{2}, \frac{\pi}{2}\right]$ and to convert it to a binary domain variable, we binned the background hue values into bins $\left[\frac{-\pi}{2}, -0.8\right]$ and $\left[\frac{\pi}{2}, 0.8\right]$. These ranges were chosen to be distinct enough that we can easily distinguish between the domains yet large enough to keep the majority of original samples in this altered dataset. We split the 18k binned samples into 90% training data and 10% validation data for our experiment.

## D.2 METRICS

Inspired by the work in Monteiro et al. (2023), we define four metrics (Effectiveness, Preservation, Composition, and Preservation) specifically for the image-based counterfactuals with latent SCMs. The key idea is to check if the correct latent information is changed when generating domain counterfactuals (e.g., domain-specific information is changed, while all else is preserved). Since we don't have direct access to the ground truth value of latent variables (nor their counterfactual values), we use a domain classifier $h_{\text{domain}}$ and class classifier $h_{\text{class}}$ to measure if the intended change has taken place.

**Effectiveness**: The idea is to check if the domain-specific variables change as wish in the counterfactual samples.

$$\mathbb{P}\left(h_{\text{domain}}\left(\hat{x}_{d \to d'}\right) = d'\right)$$

**Preservation**: This checks if the semantically meaningful content (i.e. the class information) that is independent of the domain is left unchanged while the domain is changed.

$$\mathbb{P}\left(h_{\text{class}}\left(\hat{x}_{d \to d'}\right) = y\right)$$

**Composition**: We check if our model is invertible on the image manifold, thus satisfying the pseudoinvertibility criteria.

$$\mathbb{P}\left(h_{\text{class}}\left(\hat{x}_{d \to d}\right) = y\right)$$

**Reversibility**: This metric checks if our model is cycle-consistent, or in other words, checking if the mapping between the observation and the counterfactual is deterministic.

$$\mathbb{P}\left(h_{\text{class}}\left(\hat{x}_{d \to d' \to d}\right) = y\right)$$

For the domain classifier $h_{\text{domain}}$ and class classifier $h_{\text{class}}$, we used pretrained ResNet18 models (He et al., 2016) that were fine-tuned by classifying *clean* samples (i.e. not counterfactuals) for 25 epochs with the Adam optimizer, a learning rate of 1e-3, and a random data augmentation with probabilities: $50\%$: no augmentation, $17\%$: sharpness adjustment (factor=2), $17\%$: gaussian blur (kernel size=3), $17\%$: gaussian blur (kernel size=5). A reminder that for MNIST/FMNIST/ColorRotated MNIST, the domain is rotation and the label is the original label of images (digits/type of clothes), for 3D shapes, the domain is object shape and the label is object color, and for Causal3DIdent, the domain is hue of the background and the label is the object shape.

### D.3 Causal Interpretation of our Experiments

In this section, we introduce the causal interpretation of our experiments. To evaluate the model's capability of generating good domain counterfactuals, for each dataset, we have one domain latent variable and choose one class latent variable that are generated independently of the domain latent variable. As an example, for RMNIST, we choose rotation as the domain latent variable and digit class as the class latent variable. As indicated in Figure 18a, for the counterfactual query "Given we observed an image in this domain, what would have happened if it was generated by a different domain?", we should expect the image to be rotated while the class remain unchanged. Specifically, we want to check

$$\mathbb{P}(Z_{\text{rot}_{D=d'}}|X=x, D=d) \neq \mathbb{P}(Z_{\text{rot}_{D=d''}}|X=x, D=d) \quad \forall d, d', d'' \in \mathcal{D}, \, d' \neq d''$$
$$\mathbb{P}(Z_{y_{D=d'}}|X=x, D=d) = \mathbb{P}(Z_{y_{D=d''}}|X=x, D=d) \quad \forall d, d', d'' \in \mathcal{D}$$

where $\mathcal{D}$ is the set of all domains. However, in practice we cannot directly get the value of those latent variables. This motivates our choice of evaluation metric of training a domain classifier and class classifier to detect if the domain latent variable is changed and class latent variable (which we call class) is preserved in the counterfactuals.

For RMNIST/RFMNIST, we choose rotation as the domain variable and digit/clothes class as the class variable. For 3D Shapes, we choose object shape as the domain variable and hue of objects as the class variable. For CRMNIST, we choose rotation as the domain variable and digit class as the class variable. For CausalIdent, we choose the hue of the background as the domain variable and object class as the class variable. In the case of 3D Shapes, we can technically choose anything other than object shape as the class variable. However, for simplicity, we choose one of them. In the case of CRMNIST, we cannot choose $Z_{\text{color}}$ because it will change after we change the domain. In the case of Causal3DIdent, we can choose anything but the hue of the object, though we figure $Z_y$ is easier to check and can reduce error caused by classifier proxies.

We also want to note that other than observational image, access to domain information is also important for answering this query. For example, in the case of RMNSIT, given an image that looks like digit "9", for the question "what would have happened if it is in domain $90°$", the fact that the current digit is in domain $0°$ (which means it is indeed digit "9") or the current digit is in domain $180°$ domain $0°$ (which means it is digit "6") would lead to different answer.

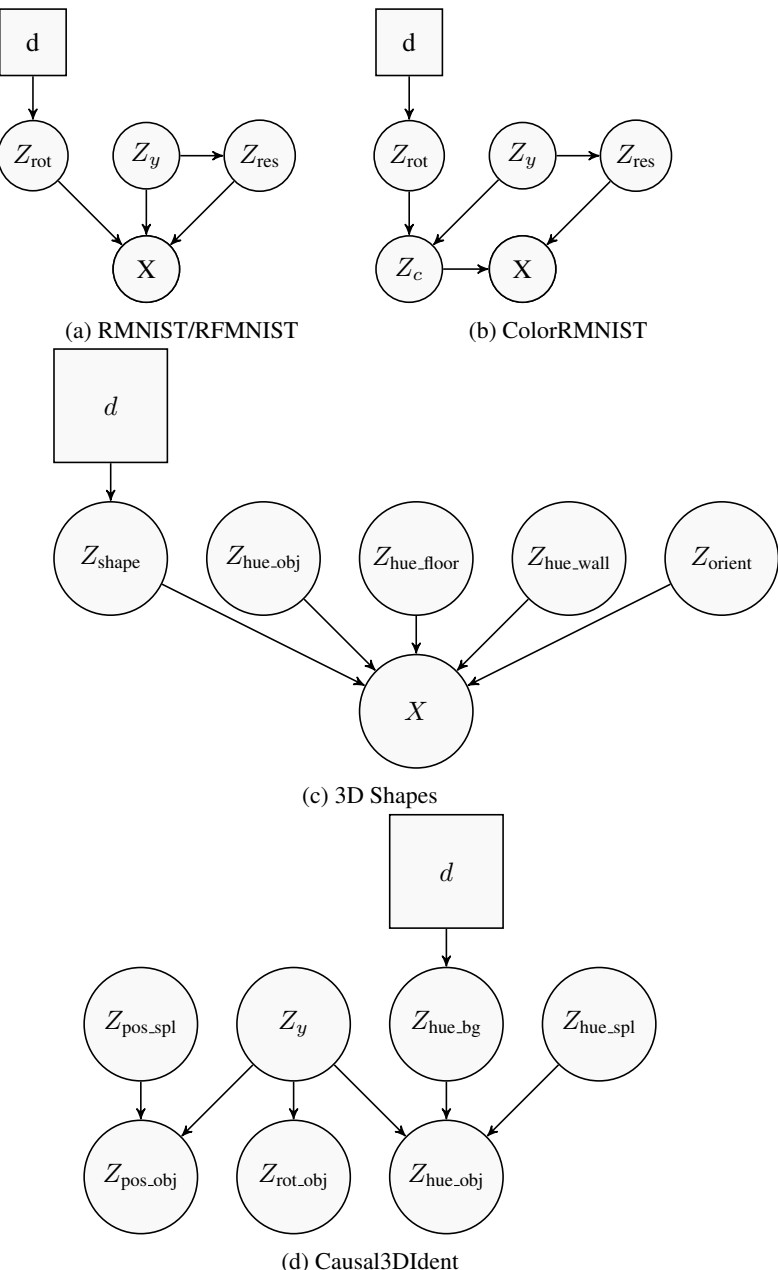

Figure 18: (a) RMNIST/RFMNIST. Here $Z_{\text{rot}}$ represents the rotation of the image, $Z_y$ represents the original RMNIST/RFMNIST class, $Z_{\text{res}}$ contains other detail information such as writing style, which is controlled by how MNIST dataset was originally created. (b) $Z_c$ represents the color of the digit while others are the same as (a). (c) 3D Shapes. $Z_{\text{shape}}$ represents the object shape. $Z_{\text{hue\_obj}}, Z_{\text{hue\_floor}}, Z_{\text{hue\_wall}}$ represent the hue of the object, floor and wall respectively. $Z_{\text{orient}}$ represents the orientation of the object. (d) Causal3DIdent. $Z_y$ represents the object class. $Z_{\text{hue\_obj}}, Z_{\text{hue\_bg}}, Z_{\text{hue\_spl}}$ represent the hue of the object, background and spotlight respectively. $Z_{\text{pos\_obj}}, Z_{\text{pos\_spl}}$ represent the position of the object and spotlight respectively. $Z_{\text{rot\_obj}}$ represents the rotation of the object. $X$ is not shown in the graph but all nodes should point to it.

## D.4 EXPERIMENT DETAILS

**Model setup** The relaxation to pseudo invertibility allows us to modify the ILD models to fit a VAE (Kingma and Welling, 2013) structure. The overall VAE structure can be seen in Fig. 19,

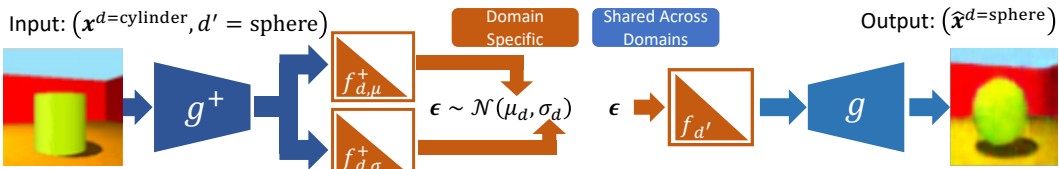

Figure 19: The model structure for the pseudo-invertible ILD model used in the high-dimensional experiments. The overall structure matches that of a VAE where the encoder (left) first projects to the latent space via $g^+$ (the pseudoinverse of the observation function $g$). This latent encoding is then passed to two domain-specific autoregressive models $f_{d,\mu}^+, f_{d,\sigma}^+$ which produce the mean and variance parameters (respectively) of the Gaussian posterior distribution. During training, the exogenous noise variable $\epsilon$ is then found via sampling from the posterior distribution ($\epsilon \sim \mathcal{N}(\mu_d, \sigma_d)$) which can be viewed as a stochastic SCM, however, during inference the exogenous variable is set to the mean of the latent posterior distribution (i.e. $\epsilon := \mu_d$) to reduce noise when producing counterfactuals. The decoder (right) follows the usual VAE decoder structure, with the exception that the initial linear layer is an autoregressive function of the $\epsilon$ input. The structure of all the $f$ models is determined by the type of ILD model used (e.g., dense, canonical, or identity canonical) and matches that seen in Fig. 3.

where the variational encoder first projects to the latent space via $g^+$ to produce the latent encoding $z$, which is then passed to two domain-specific autoregressive models $f_{d,\mu}^+, f_{d,\sigma}^+$ which produce the mean and variance parameters (respectively) of the Gaussian posterior distribution. The decoder of the VAE follows the structure typical ILD structure: $g \circ f_d$. Here, $g^+$ can be viewed as the pseudoinverse of the observation function $g$ and $f_d$ can be viewed as a pseudoinverse of $f_{d,\mu}^+$ During training, the exogenous noise variable $\epsilon$ is then found via sampling from the posterior distribution ($\epsilon \sim \mathcal{N}(\mu_d, \sigma_d)$) which can be viewed as a stochastic SCM, however, to reduce noise when producing counterfactuals, when performing inference the exogenous variable is set to the mean of the latent posterior distribution (i.e. $\epsilon = \mu_d$). In all experiments, $g$ and $g^+$ follow the $\beta$-VAE architecture seen in Higgins et al. (2017) (with the exception that in the Causal3DIdent experiment, $g$ and $g^+$ follow the base VQ-VAE architecture (Van Den Oord et al., 2017) without the quantizer), and the structure of the $f$ models is determined by the type of ILD model used (e.g., independent, dense, or relaxed canonical) and matches that seen in the simulated experiments and visualized in Fig. 3. For the $f$ models which enforce sparsity (i.e. *ILD-Can*), we use a sparsity level, $|\mathcal{I}|$, of 5. We also introduce an additional baseline, *ILD-Independent*, which has an architecture similar to the *ILD-Dense* baseline, with the exception that the $g$ and $g^+$ functions are no longer shared across domains. The *ILD-Independent* baseline can be seen as training an independent $\beta$-VAE for each domain, where each $\beta$-VAE an autoregressive $f_{dense}$ model as its last (first) layer for the encoder (decoder), respectively. For experiment with RMNSIT, RFMNIST, 3D Shapes and Causal3DIdent, we choose $m = 20$ and for CRMNIST, we choose $m = 10$.

**Training** We train each ILD model for 300K, 300K, 300K, 500K, and 200K steps for RMNIST, RFMNIST, CRMNIST, 3D Shapes and Causal3DIdent respectively using the Adam optimizer (Kingma and Ba, 2014) with $\beta_1 = 0.5, \beta_2 = 0.999$, and a batch size of 1024. The learning rate for $g$ and $g^+$ is $10^{-4}$, and all $f$ models use $10^{-3}$. During training, we calculate two loss terms: a reconstruction loss $\ell_{recon} = |x - \hat{x}|_2^2$ where $\hat{x}$ is the reconstructed image of $x$ and the $\ell_{align}$ alignment loss which measures the KL-divergence between the posterior distribution $Q_d(\epsilon|x)$ and the prior $P(\epsilon)$. Following the $\beta$-VAE loss calculation in Higgins et al. (2017), we apply a $\beta_{KLD}$ upscaling to the alignment loss such that $\ell_{total} = \ell_{recon} + \beta_{KLD} * \ell_{align}$. For all MNIST-like experiments, we use $\beta_{KLD} = 1000$, which we found leads to the lowest counterfactual error on the validation datasets across all models; this also matches the $\beta_{KLD}$ used in Burgess et al. (2018), and for 3DShape and Causal3DIdent we found $\beta_{KLD} = 10$ leads to the lowest counterfactual error.

### D.5 ADDITIONAL RESULTS

The quantitative results in Table 7, Table 8, and Table 9 match the visual result seen in Figure 20, Figure 21, Figure 22, where almost across all datasets the *ILD-Can* model seems to find a proper latent causal structure that can disentangle the domain information from the class information –

| | RMNIST | | | | RFMNIST | | | |
|---|---|---|---|---|---|---|---|---|
| | Comp. | Rev. | Eff. | Pre. | Comp. | Rev. | Eff. | Pre. |
| *ILD-Independent* | $99.79 \pm 0.44$ | $32.56 \pm 0.20$ | $94.97 \pm 4.71$ | $32.49 \pm 0.22$ | $69.75 \pm 1.86$ | $22.36 \pm 0.76$ | $99.62 \pm 0.37$ | $22.54 \pm 1.19$ |
| *ILD-Dense* | $99.76 \pm 0.28$ | $32.60 \pm 0.21$ | $80.92 \pm 2.21$ | $32.64 \pm 0.23$ | $71.20 \pm 3.39$ | $24.23 \pm 2.51$ | $98.51 \pm 0.93$ | $23.98 \pm 2.18$ |
| *ILD-Can* | $99.85 \pm 0.27$ | $79.84 \pm 17.54$ | $96.72 \pm 1.89$ | $64.99 \pm 9.83$ | $71.79 \pm 4.55$ | $70.44 \pm 3.54$ | $98.82 \pm 0.73$ | $62.15 \pm 6.65$ |

Table 7: Quantitative result with RMNIST and RFMNIST, where higher is better. They are both averaged over 20 runs.

| | CRMNIST | | | | 3D Shapes | | | |
|---|---|---|---|---|---|---|---|---|
| | Comp. | Rev. | Eff. | Pre. | Comp. | Rev. | Eff. | Pre. |
| *ILD-Independent* | $87.24 \pm 11.98$ | $59.88 \pm 6.46$ | $94.65 \pm 15.34$ | $60.39 \pm 6.95$ | $99.79 \pm 0.44$ | $32.56 \pm 0.20$ | $94.97 \pm 4.71$ | $32.49 \pm 0.22$ |
| *ILD-Dense* | $88.18 \pm 17.84$ | $62.29 \pm 10.51$ | $92.72 \pm 15.52$ | $59.60 \pm 8.92$ | $99.76 \pm 0.28$ | $32.60 \pm 0.21$ | $80.92 \pm 2.21$ | $32.64 \pm 0.23$ |
| *ILD-Can* | $92.10 \pm 13.24$ | $85.74 \pm 13.33$ | $94.48 \pm 10.71$ | $72.95 \pm 12.42$ | $99.85 \pm 0.27$ | $79.84 \pm 17.54$ | $96.72 \pm 1.89$ | $64.99 \pm 9.83$ |

Table 8: Quantitative result with CRMNIST and 3D Shapes, where higher is better. CRMNIST are averaged over 20 runs and 3D Shapes are averaged over 5 runs.

unlike the baseline models which seem to commonly change the class during counterfactual. We again note that the training process for all of the models only include the typical VAE invertibility loss (i.e. reconstruction loss) and latent alignment loss (i.e. the KL-divergence between the latent prior and posterior distributions) and do not specifically include any counterfactual training. Thus, we conjecture the enforcing of sparsity in the canonical models correctly biased these models in a manner that preserved important non-domain-specific information when performing counterfactuals. In Figure 25, Figure 26, Figure 27 and Figure 28, we track the change of our metrics w.r.t $|\mathcal{I}|$ (we did not do this investigation for Causal3DIdent because that the training of that model takes much longer time). We observe that as we increase $|\mathcal{I}|$, the reversibility and preservation tends to decrease while the effectiveness tends to increase. We conjecture that this is because as $|\mathcal{I}|$ increases, there is less constraint on the original optimization problem (fitting the observational distribution) which could potentially increase the performance. However, it leads to lower chance in finding a proper latent causal structure for domain counterfactual generation, which results in the decrease in preservation. *ILD-Dense* can be regarded as an extreme case of this. In summary, we validate the practicality of our model design in the pseudoinvertible setting with extensive study on 5 image-based datasets.

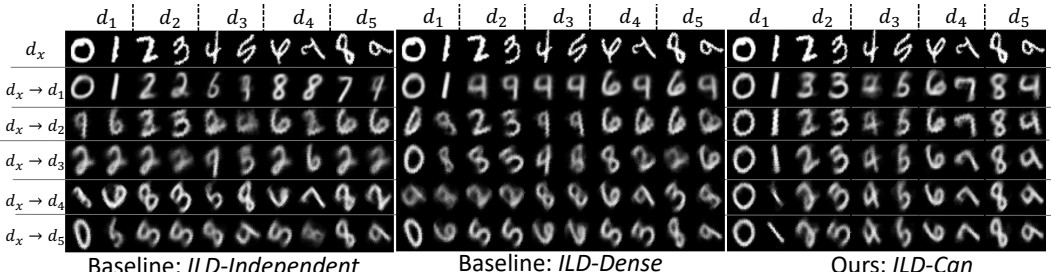

Figure 20: Counterfactual plots for the three relaxed ILD models, where across the columns we show examples of two clothing classes (e.g., "handbag" or "boot") from each domain and each row corresponds to the counterfactual to a different domain. It can be seen that while all models correctly recover the rotation for each domain counterfactual, the baseline models usually change the class label during counterfactual, while *ILD-Can* tends to preserve the clothing label, despite not being privy to any label information during training.

| | Causal3DIdent | | | |
|---|---|---|---|---|
| | Comp. | Rev. | Eff. | Pre. |
| *ILD-Independent* | $88.15 \pm 5.0$ | $51.43 \pm 2.7$ | $91.05 \pm 17.7$ | $51.94 \pm 3.0$ |
| *ILD-Dense* | $83.59 \pm 5.4$ | $49.17 \pm 2.5$ | $92.17 \pm 13.6$ | $48.83 \pm 3.0$ |
| *ILD-Can* | $86.00 \pm 5.6$ | $79.73 \pm 6.6$ | $84.15 \pm 23.5$ | $79.73 \pm 8.6$ |

Table 9: Quantitative result with Causal3DIdent, where higher is better. Causal3DIdent are averaged over 10 runs.

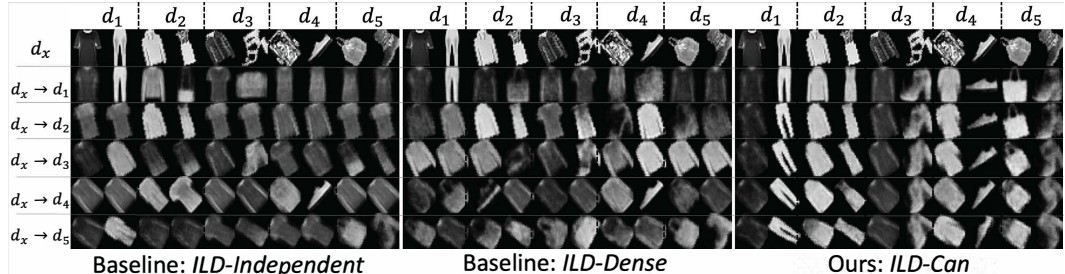

Figure 21: Counterfactual plots for the three ILD models, where across the columns we show examples of two classes from each domain and each row corresponds to the counterfactual to a different RMNIST domain. It can be seen that while all four models correctly recover the rotation for each domain counterfactual, the baseline models usually change the digit label during counterfactual, while *ILD-Can* tends to preserve the digit label, despite not being privy to any label information during training.

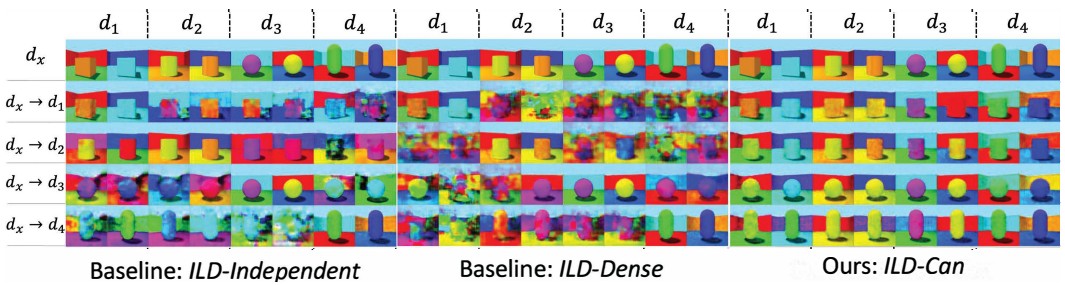

Figure 22: Counterfactual plots for the three ILD models, where across the columns we show examples of two classes from each domain and each row corresponds to the counterfactual to a different object shape domain.

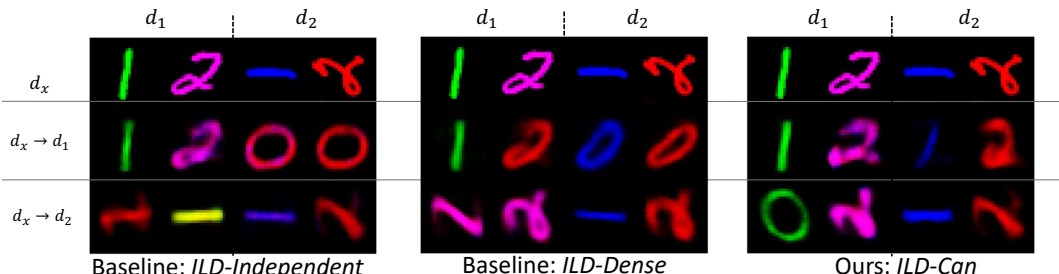

Figure 23: Counterfactual plots for the three ILD models, where across the columns we show examples of two classes from each domain and each row corresponds to the counterfactual to a different rotation domain.

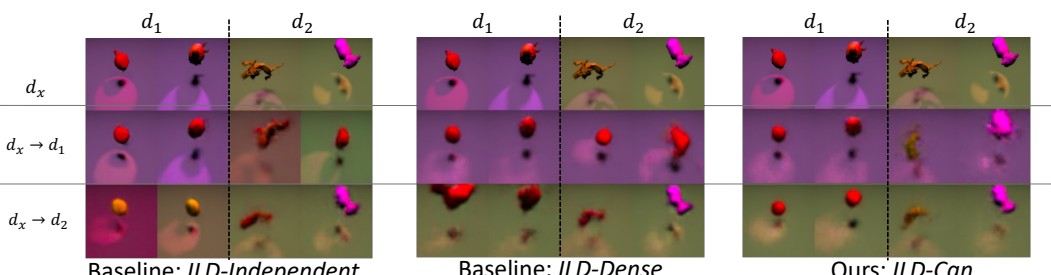

Figure 24: Counterfactual plots for the three ILD models, where across the columns we show examples of two classes from each domain and each row corresponds to the counterfactual to a different background hue domain.

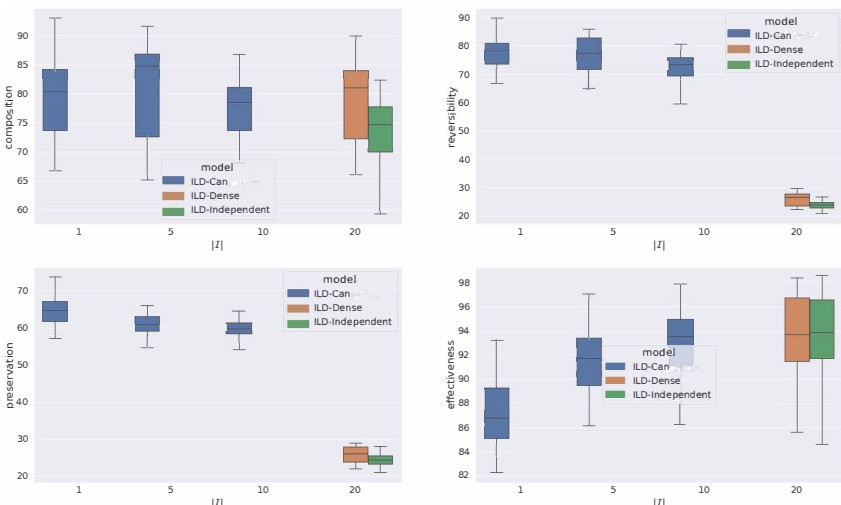

Figure 25: Change of metrics w.r.t $|\mathcal{I}|$ for RMNIST. Results are with 20 runs and we remove outliers when plotting.

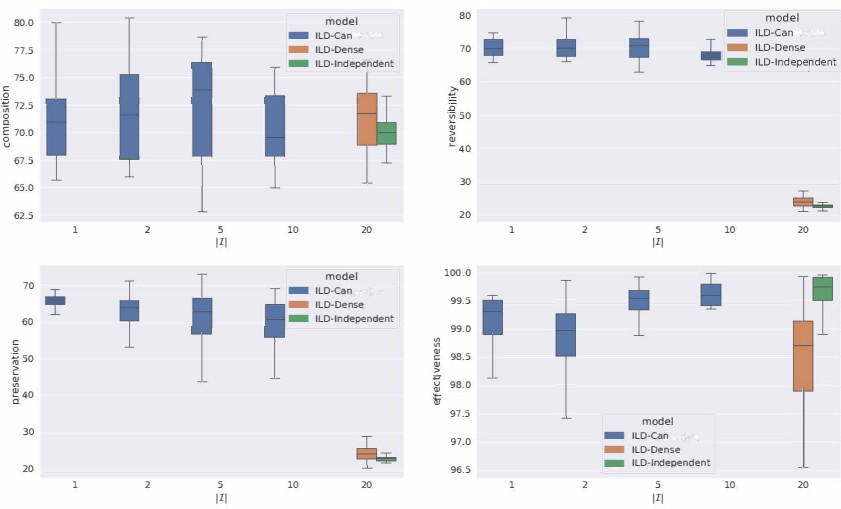

Figure 26: Change of metrics w.r.t $|\mathcal{I}|$ for RFMNIST. Results are with 20 runs and we remove outliers when plotting.

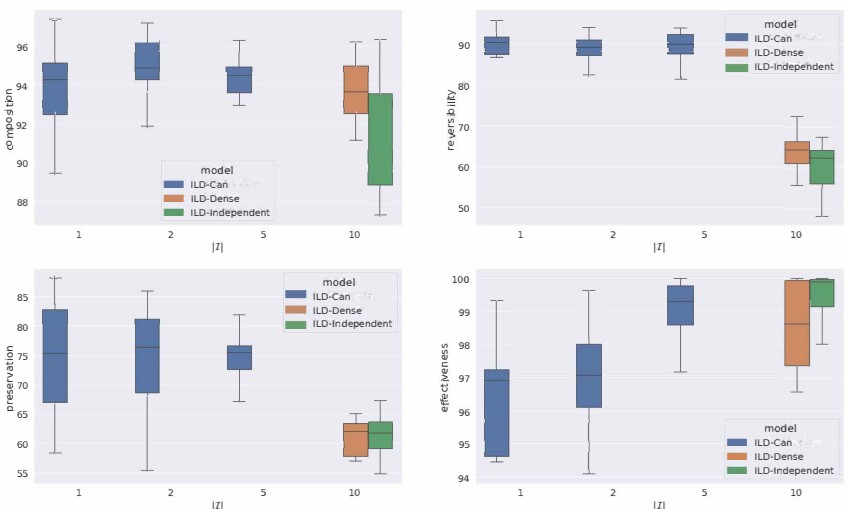

Figure 27: Change of metrics w.r.t $|\mathcal{I}|$ for CRMNIST. Results are with 20 runs and we remove outliers when plotting.

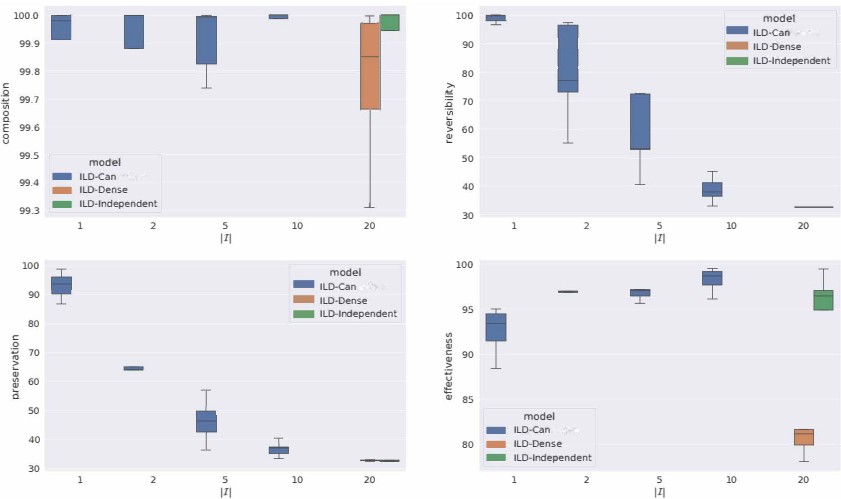

Figure 28: Change of metrics w.r.t $|\mathcal{I}|$ for 3D Shapes. Results are with 5 runs and we remove outliers when plotting.

# E  LIMITATIONS

A practical problem we noticed in our simulated experiments is that sometimes the sparse model is harder to fit, i.e., its log-likelihood is worse than the dense model, even if we only consider the cases where the true model is in the model class being optimized (e.g., the sparsity of the model is at least as large as the sparsity of the ground truth model). We conjecture that this results from a harder loss landscape as we add more constraints to the model. We believe a more careful investigation of the model and algorithm could be an interesting and important future work. For example, if we use a more significantly overparameterized model, there are chances that the training of *ILD-Can* would become easier. Additionally, the addition of further loss terms could aid in the training of these models, such as, assuming access to some ground truth domain counterfactuals (e.g., the same patient received imaging at multiple hospitals) could be used to penalize our model when it changes latent variables which do not change under the ground truth counterfactuals.

In our experiments, we aimed to test the effects of breaking some of our assumptions (e.g., "what if our model is not strictly invertible"), and while our models still performed better in these cases, there are likely cases where the breaking of our assumptions can cause our models to fail to produce faithful counterfactuals. For example, in a case where there is a very large difference between domains and there is no sparsity in the domain shifts, then it is likely that the constraints constituted by our sparsity assumption will make the sparse models struggle to fit the observed distributions.

# F  EXPANDED RELATED WORK

**Causal Representation Learning**   Causal Representation Learning (CRL) is a rapidly developing field that aims to discover the underlying causal mechanisms that drive observed patterns in data and learn representations of data that are causally informative (Schölkopf et al., 2021). This is in contrast to traditional representation learning, which does not consider the causal relationships between variables. An extensive review can be found in Schölkopf et al. (2021). As this is a highly difficult task, most works make assumptions on the problem structure, such as access to atomic interventions, the graph structure (e.g., pure children assumptions), or model structure (e.g., linearity) (Xie et al., 2023; Yang et al., 2022; Huang et al., 2022; Liu et al., 2022b; Xie et al., 2022; Chen et al., 2022; Squires et al., 2023; Zhang et al., 2023; Sturma et al., 2023; Jiang and Aragam, 2023; Liu et al., 2022a; Varici et al., 2023). Other works such as (Brehmer et al., 2022; Ahuja et al., 2022; Von Kügelgen et al., 2021) assume a weakly-supervised setting where one can train on counterfactual pairs $(x, \tilde{x})$ during training. Lachapelle et al. (2023) address the identifiablity of a disentangled representation leveraging multiple sparse task-specific linear predictors. In our work, we aim to maximize the practicality of our assumptions while still maintaining our theoretical goal of equivalent domain counterfactuals (as seen in Table 1).

**Counterfactual Generation**   Counterfactual examples are answers to hypothetical queries such as "What would the outcome have been if we were in setting $B$ instead of $A$?". A line of works focus on the identifiability of counterfactual queries (Nasr-Esfahany et al., 2023; Shah et al., 2022). For example, given knowledge of the ground-truth causal structure, Nasr-Esfahany et al. (2023) are able to recover the structural causal models up to equivalence. However, they do not consider the latent causal setting and they assume some prior knowledge of underlying causal structures such as the backdoor criterion. There is a weaker form of counterfactual generation which does not use causal reasoning but instead uses generative models to generate counterfactuals (Nemirovsky et al., 2022; Zhu et al., 2017; Zhou et al., 2022; Choi et al., 2018; Zhou et al., 2023; Kulinski and Inouye, 2023). These typically involve training a generative model which has a meaningful latent representation that can be intervened on to guide a counterfactual generation (Ilse et al., 2020). As these works do not directly incorporate causal learning in their frameworks, we consider them out of scope for this paper. Another branch of works try to estimate causal effect without trying to learn the underlying causal structure, which typically assume all variables are observable(Louizos et al., 2017).

**Causal Discovery and nonlinear ICA**   Causal discovery focus on identifying the causal relationships from observational data. Peters et al. (2016); Heinze-Deml et al. (2018) achieve this via the invariant mechanism between certain variable and and its direct causes. Most of these works do not assume the setting of latent variables. Similar to CRL, nonlinear ICA typically aims at finding

the mixing function. For example, some works try to identify it with access to auxiliary variables (Hyvarinen et al., 2019; Khemakhem et al., 2020), by adding constraint on the mixing functions (Gresele et al., 2021; Moran et al., 2021) or under specific scenario such as bivariate setting (Wu and Fukumizu, 2020). Zheng et al. (2022) relax the constraint of auxiliary variable and impose structure sparsity to achieve identifiability result, where structure sparsity is less general than the mechanism sparsity discussed in our work. In contrast with CRL, most nonlinear ICA works do not consider latent variables that are causally related.

