# OpenReview forum: "Towards Characterizing Domain Counterfactuals for Invertible Latent Causal Models"
_ICLR.cc/2024/Conference — ICLR 2024 poster_

### Official Review · Reviewer_KSip · 2023-10-26

**Soundness:** 3 good
**Presentation:** 2 fair
**Contribution:** 2 fair
**Rating:** 6
**Confidence:** 4

**Summary:**

This work focuses on domain counterfactuals: *What a sample would have looked like if it had been generated in a different environment*.
It provides a characterization of *domain counterfactually equivalent* models. They show that sparsity of domain interventions as an inductive bias can help reduce the search space and generate more accurate counterfactuals.

**Strengths:**

The problem that this work considers has practical significance, e.g., in combining datasets in domains where data collection is expensive.

**Weaknesses:**

* I think the writing and structure of the paper can be improved. A large fraction of space is used for many theorems, some of which are not that important. I suggest moving these theorems to the appendix and only keeping Theorem 2 in section 3.2. Furthermore, the simulated experiments look very contrived to me. The data generation mechanism used for creating data in these experiments is too simplistic. It consists of linear transformations followed by a leaky relu (I had to dig into a very long appendix to find this). The model also seems to have knowledge of the precise form of data generation mechanism (linear + relu), and only fits that form.
To me, the interesting experiments come in section 5.2. However, I can't find an explanation of how sparsity is enforced in these experiments.
In summary, I think authors can restructure the paper to provide more room for experiments in 5.2., and explain the methodology in detail.

**Questions:**

* How does one set the number of intervention (k)?
* If we don't have the right k, or if the form of the ILD is not precisely known, how off could be our counterfactual estimates? Do we have identifiability in this case?
* How does this work compare with a prior line of work in using sparsity for identifiable representations, e.g., [Synergies between Disentanglement and Sparsity: Generalization and Identifiability in Multi-Task Learning](https://proceedings.mlr.press/v202/lachapelle23a.html) or [On the Identifiability of Nonlinear ICA: Sparsity and Beyond](https://proceedings.neurips.cc/paper_files/paper/2022/hash/6801fa3fd290229efc490ee0cf1c5687-Abstract-Conference.html), just to name a few?

---

> ### Author Response · Authors · 2023-11-17
> **Response to Reviewer KSip (1/2)**
>
> We thank the reviewer for the constructive feedback. We respond to the questions and suggestions individually below. We have also modified the paper accordingly.
>
>
> > “The simulated experiments look very contrived to me. The data generation mechanism used for creating data in these experiments is too simplistic.”
>
> We apologize for any lack of clarity regarding the data-generating process in the main paper, and we have revised the simulated experiment section to clarify the form of the observation function (marked in blue). To address the concern about the simplicity of our simulated experiments, we have added additional experiments where we use a more complicated normalizing flow (based on RealNVP) for the observation function in the ground truth dataset, and our model is misspecified as either a different normalizing flow structure or a VAE structure. More details and relevant figures can be found in Appendix J.2. In summary, we observe a similar trend as that in Figure 1, using the ILD-Relax-Can model results in lower counterfactual error than the ILD-Dense model, regardless of whether the structures of $g^*$ and $g$ match.
>
> > Motivation of the simulated experiment.
>
> The primary aim of our experimental section is to demonstrate the practical utility of our theoretical framework. We begin with scenarios where all our assumptions are met, then progressively explore cases where these assumptions are not fully satisfied, such as model misspecifications in the intervention size. Following the reviewer's suggestion, we have further extended our investigation to include scenarios with misspecified functional classes of $g$. Finally, in Section 5.2, we test our model design in a more practical setting where we relax the invertibility assumption.
>
> We also wish to highlight that our simulated experiments encompass a variety of distinct latent causal mechanisms. We have investigated 10 different SCMs for all of our experiments as discussed in Appendix G.1.
>
> > “However, I can't find an explanation of how sparsity is enforced in these experiments. In summary, I think authors can restructure the paper to provide more room for experiments in 5.2., and explain the methodology in detail.”
>
> In Section 5.2, sparsity is enforced by using the sparse $f_\cdot$ structure (i.e. only $k$ mechanisms in $f_d$ are allowed to change across different domains) as the final layer of the VAE encoder and the initial layer of the VAE decoder. Informally, this can be thought of as the encoder of the VAE generating the Gaussian parameters in a sparse fashion and the decoder of the VAE is sparse due to its initial layer being sparse. An overview of this VAE setup can be seen in the encoder/decoder graphics in Figure 17.
>
>
>
>
> >“I think the writing and structure of the paper can be improved. A large fraction of space is used for many theorems, some of which are not that important. I suggest moving these theorems to the appendix and only keeping Theorem 2 in section 3.2.”
>
> Thank you for your feedback. We agree that the writing of the theoretical sections can be simplified and have substantially revised Section 2 accordingly. Regarding the importance of Theorem 1 and 4, we summarize how they play critical roles in our paper below.
>
> **Theorem 1**: This theorem establishes both necessary and sufficient conditions for counterfactual equivalence. It forms the foundational basis for subsequent theorems in our paper as it allows for easy verification of counterfactual equivalence in ILDs and can facilitate the rapid construction of counterfactual equivalence models for any given ILD.
>
> **Theorem 4**: Together with Corollary 5, Theorem 4 demonstrates that all counterfactually equivalent ILDs share the same sparsity in intervention. This insight is crucial as it shows that constraints on the intervention set size do not exclude any equivalent ILDs, and it significantly narrows the functional space in practical applications, which leads to our model design in practice.

---

> > ### Author Response · Authors · 2023-11-17
> > **Response to Reviewer KSip (2/2)**
> >
> > [continued from above reponse]
> >
> > > Discussion on the size of the intervention set
> >
> > We kindly refer to the response to all reviewers for an in-depth discussion on how choices of $k$ will matter in theory and how to choose it in practice.
> >
> > > How does this work compare with a prior line of work in using sparsity for identifiable representations, e.g., Synergies between Disentanglement and Sparsity: Generalization and Identifiability in Multi-Task Learning or On the Identifiability of Nonlinear ICA: Sparsity and Beyond, just to name a few?
> >
> > Thank you for highlighting these important works. Given this, we have updated the discussions and citations of several sparsity-related studies in the revised manuscript, marked in blue in Appendix A. To summarize the main differences between our paper and the papers mentioned by the reviewer:
> >
> > [1] considers independent latent source variables while in our paper latent variables $z$ are generated by a SCM. Besides, [1] focuses on structured sparsity which describes if certain observables are generated by specific sources. In contrast, our paper discusses the sparsity of mechanism which is more general. This is also discussed in Section 6 in their paper as a limitation of their work.
> >
> > [2] mainly addresses the identifiability of a disentangled representation leveraging multiple sparse task-specific linear predictors (i.e. identifying $g^{*^{-1}}$) and does not study causal estimation (i.e. they ignore $f$), and thus can not directly apply to our task of counterfactual estimation.
> >
> > [1] Zheng, Y., Ng, I., & Zhang, K. (2022). On the identifiability of nonlinear ica: Sparsity and beyond. Advances in Neural Information Processing Systems, 35, 16411-16422.
> >
> > [2] Lachapelle, S., Deleu, T., Mahajan, D., Mitliagkas, I., Bengio, Y., Lacoste-Julien, S., & Bertrand, Q. (2023, July). Synergies between disentanglement and sparsity: Generalization and identifiability in multi-task learning. In International Conference on Machine Learning (pp. 18171-18206). PMLR.

---

> > > ### Comment · Reviewer_KSip · 2023-11-21
> > > **Reviewer response**
> > >
> > > Thanks for the detailed answers. I increased my score.

---

### Official Review · Reviewer_jRXL · 2023-10-31

**Soundness:** 2 fair
**Presentation:** 2 fair
**Contribution:** 3 good
**Rating:** 5
**Confidence:** 3

**Summary:**

**Post rebuttal update** I maintain my score (and uncertainty) but add my final reply to the author(s). Overall, I think both the original submission and the rebuttal revision are too clustered; the submission has many defs and results, and the revision adds more.

*Generalized causal mechanisms*

I still cannot understand. I have no problems with “reordering wlog”, however, if there exists a non-invertible subsystem $s$ in an invertible SCM (now Def 1), the reordering only changes the labels of the input nodes of $s$, not the *functional form* of $s$. Did I miss anything? Do you change the functional form of $s$ via $g$? But $g$ is invertible, I do not think we can change the invertibility of $s$ via $g$.

*Rosenblatt transformation*

I still hold my original view. To be clear, I use $J$ to denote the univariate CDFs with fixed conditional values. For a sample $\mathcal{D}$, we have

$$u_2=J_2(x_2):=F_2(x_2|x_1),$$

but, for another sample $\mathcal{D}’$, we have

$$u_2’=J_2’(x_2'):=F_2(x_2'|x_1’).$$

$J_i \neq J_i'$ except for $i=1$.

Or, we say there does not exist a single set of invertible CDFs for all possible values of RV X.

**End update**

This work considers the situation that different domains have the same causal graph but different structural causal functions; the “intervention” in this paper is the “*domain difference of causal mechanisms*”, and the “counterfactual” is the “*domain adaptation of observations*” under a pre-defined domain difference. More specifically, the “intervention” means changing a structural function (from one domain to another) but leaving any other things untouched, and the “counterfactual” means simply the new observation generated by the “intervention”.

An Invertible Latent Domain Causal Model (ILD) is a set of SCMs whose structural functions are invertible and autoregressive plus an Invertible Observation Function which connects the latent and observable. The equivalence class of ILDs is characterized, and it is proved that, for any CLDs, there exist equivalent “canonical forms” whose domain difference is described by the last k structural functions under the topological ordering. Based on this, the paper makes the point that “if we know the number k of different structural functions between the domains, then we can improve the performance”.

There are experimental supports for the effectiveness of the above idea.

**Strengths:**

It is an interesting idea to see the difference between domains as invertible transformations.

The theoretical analysis seems serious (but it is quite impossible to check the proof in detail as a conference submission).

If a real-world situation satisfies the theoretical setting, then it is possible to largely reduce the complexity of finding domain differences.

**Weaknesses:**

*The idea of “intervention” and “counterfactual” in this paper is nonstandard and confusing*. As standard concepts in causal inference, an “intervention” roughly means setting a variable to a specific value but leaving any other things untouched, and “counterfactual” means imagining an intervention with the *past values* of exogenous variables untouched. In the above, *the structural functions are unchanged*. However, as indicated in my summary, the “intervention” and “counterfactual” in this paper refer to totally different things, in particular, the “counterfactual” does *not* consider the past values of exogenous variables which are the gist of “counterfactual”. Explanations regarding these are necessary, and I strongly suggest *not* using the terms “intervention” and “counterfactual” in this paper.

*The “generalized causal mechanisms” in Prop 2 are only a subclass of “invertible SCMs”*. There is no formal definition of “invertible SCMs” in this paper; I think the definition should be like: writing the whole system of SCMs as X=f(\epsilon) and function f is invertible. In the proof of Prop 2, it assumed that each $\hat{f}^{(j)}$ is invertible. But this is not implied by the general concept/definition of “invertible SCMs” I mentioned above, because there is no guarantee that the *subsystem* involving only $z_{<j}$ is invertible; information from other variables might be needed to invert the whole system, and missing any piece of information might render the subsystems non-invertible. Hereafter, I name this class of models *“invertible autoregressive SCM (IASCM)“*.

*The claim on the generality of the model class (Prop 1) seems problematic*. First, here the model class is IASCM. In the proof of Prop 1, the Rosenblatt transformation should be constructed for a fixed *sample point* of p(X), because each conditional CDF F_j should depend on fixed x_{<j}. Thus, we need a Rosenblatt transformation for each sample point, and we cannot construct the F_p and F_q for the whole distributions p(X) and q(\epsilon).

I have spent 3 hours to get around the confusion in the listed Weaknesses, and I can only skim the rest of the paper and ask some questions as below. *I will read the rebuttal and the revised paper again and update my review accordingly*.

**Questions:**

*Do we have any ideas on the identifiability of the model*? This is an important question because we discuss causality. Although the theory in the paper converts IASCMs on a set of domains into an easy-to-deal-with form that is the canonical ILD. There are no discussions on the identifiability of canonical ILDs, that is, when we really try to learn the canonical ILDs, can we identify the single eq class of canonical ILDs, which contains the true one? The practical application of the proposed idea depends critically on this question.

*How can we understand if a practical dataset satisfies the theoretical setting*? Both high-level discussions and examples are highly desirable. For example, for MNIST datasets, how can we understand that the observable (image) is related to the latent by an injective mapping? Why the latent variables in different domains are related by invertible mappings? And in the end, what are the latent variables? Note that I do not require any precise claims on the causal structure (it is latent anyway), but even reasonable “guesses” would be very helpful.

I think the 2nd equality of eq2 is also by definition?

In Def 3, point 3, I am not sure Gaussian exogenous noises are without loss of generality, any references and/or reasonings?

Shared observation function g should be explained right after Def 3.

The claim of “both counterfactually and distributionally equivalent” in Theorem 2 looks strange to me. Can’t we prove that counterfactual equivalence implies distributional equivalence?

There is a related work [1] that is worth mentioning; it also considers invertible causal mechanisms, shared function between domains, and transferability between domains. See Sec 2.3, 2.4 and 3.1 in that paper.

[1] Wu, Pengzhou, and Kenji Fukumizu. "Causal mosaic: Cause-effect inference via nonlinear ICA and ensemble method." International Conference on Artificial Intelligence and Statistics. PMLR, 2020.

Minor

There are two different defs of autoregressive functions, at the end of page 1 and in Def 1. Def 1 seems to be correct.

In Prop 2, generalize → generalized

In Corollary 3, you mentioned “Prop 8”, but I think it should be just Def 8.

---

> ### Author Response · Authors · 2023-11-17
> **Response to Reviewer jRXL (1/3)**
>
> We thank the reviewer for the insightful review. We respond to the questions and suggestions individually below. We have also modified the paper accordingly (highlighted in blue).
>
> >The idea of “intervention” and “counterfactual” in this paper is nonstandard and confusing … in particular, the “counterfactual” does not consider the past values of exogenous variables which are the gist of “counterfactual”.
>
> Thanks for the question.
>
> *Intervention*: We would like to first clarify the intervention considered in this paper. Here, we mainly consider soft intervention which corresponds to the change of functions in the structural assignment [1] [2]. We note this is different than a perfect intervention (e.g., a *do* intervention), as this does not completely eliminate the causal effect of the parents.
>
> *Counterfactual*: In this work, we are specifically interested domain counterfactual queries (i.e. what would $\mathbf{x}$ look like if it had come from domain $d’$ instead of domain $d$?), which are a specific type of the general counterfactual query. Similar to the general counterfactual query, a domain counterfactual query exactly follows the three steps: abduction, action, and prediction [3]. For example, given an image $\mathbf{x}$ from domain $d$, we want to answer the question what this image would look like if it is from domain $d’$. We first use the inverse of our observation function $g$ and our latent SCM for domain $d$ to recover the exogenous noise $\epsilon$ given the evidence (abduction) as follows: $\epsilon =  f^{-1}\_{d}\circ g^{-1}(\mathbf{x})$. Then, we perform the domain intervention (action) by exchanging the original $f_{d}$ with $f\_{d’}$. Finally, we use the recovered noise and intervened SCM to predict the counterfactual $\mathbf{x}\_{d\rightarrow d’}$ (prediction). Together this approach follows: $\mathbf{x}\_{d \rightarrow d’} = g\circ f\_{d’}\circ f^{-1}\_{d}\circ g^{-1}(\mathbf{x})$. Additionally, in our empirical analysis, we pay particular attention to the variables that should remain constant during counterfactual estimation. For instance, in our ColorRMNIST experiment, a key focus is on evaluating whether the color attribute is preserved when we alter the rotation aspect.
>
> [1] Schölkopf, B., Locatello, F., Bauer, S., Ke, N. R., Kalchbrenner, N., Goyal, A., & Bengio, Y. (2021). Toward causal representation learning. Proceedings of the IEEE, 109(5), 612-634.
>
> [2] Peters, J., Janzing, D., & Schölkopf, B. (2017). Elements of causal inference: foundations and learning algorithms (p. 288). The MIT Press.
>
> [3] Pearl J Glymour M Jewell NP. Causal Inference in Statistics : A Primer. Chichester West Sussex UK: John Wiley & Sons; 2016. , p96.
>
>
>
> > The “generalized causal mechanisms” in Prop 2 are only a subclass of “invertible SCMs” … missing any piece of information might render the subsystems non-invertible.
>
> Thanks for pointing this out. We agree that if we only assume the invertibility of $f$, we could not guarantee that the subsystem of $\widehat{f}^{(i)}$ is invertible to $\epsilon_i$ given only $z_{<i}$. However, without loss of generality, we could assume the partial order in $z$ such that parent nodes have smaller indices, then the overall function $f$ is invertible autoregressive. We believe the confusion is that we define the invertible SCM on the subfunction $\widehat{f}$, rather than on $f$ directly.
>
>
> > The claim on the generality of the model class (Prop 1) seems problematic…Thus, we need a Rosenblatt transformation for each sample point, and we cannot construct the F_p and F_q for the whole distributions p(X) and q(\epsilon).
>
> We might be misunderstanding this concern, please forgive any misinterpretation if that is the case. The Rosenblatt transformation is constructed based on a *joint distribution*, not a single fixed point [1]. But perhaps the concern stems from the fact that in prop 1, $\mathbf{x}$ represents a random vector rather than a specific instantiation. It is under this case, where $\mathbf{x}$) is a *random* vector, that the Rosenblatt transformation is well-defined. Additionally, we note that the conditional CDFs are invertible w.r.t. their first argument given previous values $x_{<j}$, but the whole function is invertible because it is autoregressive. Intuitively, this means that you can recover the first variable using the 1D invertible CDF, and then recover the 2nd variable given the first recovered value, etc. Thus, we believe our original proof is correct. Does this answer your concern (perhaps we misunderstood)?
>
> [1] Rosenblatt, Murray. “Remarks on a Multivariate Transformation.” The Annals of Mathematical Statistics, vol. 23, no. 3, 1952, pp. 470–72. JSTOR, http://www.jstor.org/stable/2236692. Accessed 17 Nov. 2023.

---

> > ### Author Response · Authors · 2023-11-17
> > **Response to Reviewer jRXL (2/3)**
> >
> > > Do we have any ideas on the identifiability of the model? … can we identify the single eq class of canonical ILDs, which contains the true one?
> >
> > The identification of the causal model will allow us to infer all causal queries, to do this in practice req. This paper’s goal, however, is to work on a special type of counterfactual query, where the two interventions associated with counterfactual queries are observable. That is true for canonical form as well. Inside the canonical form, the counterfactually equivalent models are not unique. Our theory 4 characterizes that these true ones must be in the class of canonical ILDs with intervention set $k$ equal to the ground truth model.
> >
> > Furthermore, Theorem 6 in Section 3.4 in the revised manuscript , shows that, although we could not find the counterfactual and distribution equivalent model, specifying $k$ would lower the risk.  As with many hyperparameters, there is a natural bias-variance tradeoff in choosing $k$.
> > If $k > k^*$, then the space of distributionally equivalent models is very large and thus the variance in estimation will be high. If $k < k^*$, then the model will be biased, i.e., there does not exist a model that is distributionally and counterfactually equivalent to the true ILD. Using our new bound on the expected counterfactual error, we also suggest multiple methods for choosing $k$ in practice, with the least assuming option being to start with a small $k$ and then increase $k$ until distributional equivalence is achieved (i.e., the data fit term is optimal, which minimizes the first term of the counterfactual error bound). Empirically, we noticed that in our current simulated experiments (Figure 15) and additional simulated experiments (Figure 34), there is a clear gap if the choice of $k$ is smaller than the ground truth $k^*$.
> >
> >
> > > “How can we understand if a practical dataset satisfies the theoretical setting?... For example, for MNIST datasets, how can we understand that the observable (image) is related to the latent by an injective mapping? … And in the end, what are the latent variables?
> >
> > **Practical Dataset Satisfies the theoretical setting**:
> > In practice, there are many settings that satisfy the theoretical setting (if we include the relaxation to pseudo invertibility). For example, in the image regime, it is reasonable to assume that there exists a semantically meaningful latent space that can define the observed data (e.g., for MNIST this can be the label of the digit, the width of the strokes, the writing style, etc.), which is similar to the manifold hypothesis. Furthermore, for datasets with unique domains, it is reasonable to assume the domain only directly affects a subset of these latent variables (e.g., how the digit is rotated) rather than each domain entirely changing the semantic information encoded in each image [1]. Thus, for a given dataset with unique domains, if we believe the observed data can be better characterized in a latent (i.e. semantic) manner *and* there is some semantic content preserved between domains, then that dataset likely satisfies our theoretical setting.
> >
> > **Latent Variables**:
> > We discuss this in detail in Appendix H.3, which contains conjectures on what the latent variables in the image datasets could represent. For instance, for ColorRMNIST, we expect the latent variables to correspond to four semantically meaningful concepts: the rotation of the digit (which is domain-specific), information regarding the digit label(which is domain invariant), information regarding the color label (which is domain invariant), and information regarding the writing style of the digit (e.g., how thick/straight the strokes are) which we believe to be a child of the class node(s)  (e.g., one may have a unique style of writing a 4).
> >
> > [1] Schölkopf, Bernhard, et al. "Toward causal representation learning." Proceedings of the IEEE 109.5 (2021): 612-634., Sec. III.C.(c).
> >
> > > I think the 2nd equality of eq2 is also by definition?
> >
> > Yes, thanks for pointing this out. We have revised the paper accordingly.

---

> > > ### Author Response · Authors · 2023-11-17
> > > **Response to Reviewer jRXL (3/3)**
> > >
> > > > In Def 3, point 3, I am not sure Gaussian exogenous noises are without loss of generality, any references and/or reasonings?
> > >
> > > If the causal variables are also continuous, then we only need to assume the exogenous noise to be continuous random variables. This is because any continuous distribution could be transformed to another continuous distribution by a continuous bijective mapping (by inverse Rosenblatt transformation and Rosenblatt transformation). This follows from the proof of Proposition 1.
> > >
> > > > The claim of “both counterfactually and distributionally equivalent” in Theorem 2 looks strange to me. Can’t we prove that counterfactual equivalence implies distributional equivalence?
> > >
> > > The distribution equivalence and counterfactual equivalence are two different concepts and they do not imply each other. We will use a counterexample of a linear model with two latent variables to illustrate that counterfactual equivalence does not imply distribution equivalence. If one ILD has $g=ID$, $f_1=ID$ and $f_2=2*ID$. Following the characterization in our Theorem 1, we choose $h_1=Id$ and $h_2=\begin{bmatrix}1 & 0 \\\\ 1 & 1 \\\\ \end{bmatrix}$. With this, we construct a counterfactual equivalent ILD where $g’=ID$, $f’_1=\begin{bmatrix}1 & 0 \\\\ 1 & 1 \\\\ \end{bmatrix}$ and $f_2=\begin{bmatrix}2 & 0 \\\\ 2 & 2 \\\\ \end{bmatrix}$. For any $\mathbf{x}$, we have $g\circ f_2 \circ f_1^{-1}\circ g^{-1}(\mathbf{x})=2\mathbf{x}$ and $g’\circ f_2 \circ f_1^{\prime-1}\circ g^{\prime-1}(\mathbf{x})=2\mathbf{x}$. This validates that two ILDs are counterfactually equivalent. But the induced distribution of the first domain of ILD 1, is still standard Gaussian, where the first domain of ILD 2, is not a standard Gaussian anymore.
> > >
> > >
> > > > There is a related work [1] that is worth mentioning; it also considers invertible causal mechanisms, shared function between domains, and transferability between domains. See Sec 2.3, 2.4 and 3.1 in that paper.
> > >
> > > Thanks for mentioning this relevant paper. We have added and discussed it in our revised manuscript (See Appendix A).

---

> > ### Comment · Reviewer_jRXL · 2023-11-21
> >
> > Thanks for the rebuttal. My main concerns remain.
> >
> > **Soft intervention**
> >
> > The definition should be clearly spelled out in a revision. The references you gave are from the ML community; I suspect this concept is widely accepted outside the ML community, or even inside the ML community. I tend to agree that, if we accept the “soft intervention”, then your “counterfactual” can be seen as a counterpart of the standard counterfactual; but all these need careful explanation in the paper.
> >
> > **Generalized causal mechanisms**
> >
> > I cannot understand how a re-ordering of nodes can make non-invertible subsystems invertible.
> >
> > **Rosenblatt transformation**
> >
> > After another look at the original reference and your paper, it seems to me that, in the original reference, F_n is seen as a function of x_{≤n}, thus is *non*-invertible. But in your proof, you see F_n as invertible, then this necessarily means that *your* F_n should be a function of x_n, with x_{<n} *fixed* by a sample value; this was also my thought when reading your paper.

---

> > > ### Author Response · Authors · 2023-11-22
> > > **Response to Comment by Reviewer jRXL**
> > >
> > > Thank you for your reply! We have tried to address the concerns you listed, but if there are any more concerns, please let us know, and we would be happy to address them.
> > >
> > > > Soft interventions
> > >
> > > Thanks for the suggestion. We have revised the paper to clarify this in Proposition 1, we also emphasized it in Definition 3 and added a paragraph in Appendix B.3 to expound on these points.
> > >
> > > > Generalized causal mechanisms
> > >
> > > We thank the reviewer for helpful suggestions on the clarity of this paper! We have further revised Section 2 to be more clear. We added a formal definition of invertible SCM (now Definition 1) according to the reviewer’s suggestion. We would like to note that in Definition 1, an invertible $f$ does not imply anything about the order of $z_i$. Then in Defition 2 (originally Definition 1), we introduce our ILD where we assume an autoregressive structure. We further explain that we could assume this autoregressive structure without hurting the expressiveness of ILDs in Remark 1 (highlighted in blue in Appendix  B.1).  Essentially, this is because given any model that satisfies all conditions of ILDs but autoregressiveness, we can use the observation function $g$ to reorder the latent causal variables. Hence, we can assume autoregressiveness in ILD without loss of generality. Additionally, in proposition 3, (originally proposition 2), we only discuss the autoregressive invertible representation of the invertible SCM *that are partial ordered*. However, due to the page limit, we have moved this discussion to Section B (currently page 16).
> > >
> > > With these edits, have we addressed your concerns appropriately?
> > >
> > >
> > > > Rosenblatt transform
> > >
> > > Thank you for your further comments on this. We would like to note that $F_n$ is a conditional CDF and only invertible w.r.t. $x_n$ when $\mathbf{x}_{<n}$ are fixed values. To make this clear, we denote $\mathbf{X}$ as the random variables and $\mathbf{x}$ as the realizations. We have rewritten equations 15 and 16 using this notation in the revised manuscript. For example,
> > > $$
> > > u_1 = F_1(X_1=x_1)
> > > $$
> > > $$
> > > u_2 = F_2(X_2=x_2| X_1 = x_1)
> > > $$
> > > and
> > > $$
> > > x_1 = F_1^{-1}(U_1=u_1)
> > > $$
> > > $$
> > > x_2 = F_2^{-1}(U_2=u_2 | X_1=F_1^{-1}(u_1))
> > > $$
> > > Does this address your concerns?

---

### Official Review · Reviewer_4MQs · 2023-11-01

**Soundness:** 3 good
**Presentation:** 3 good
**Contribution:** 3 good
**Rating:** 6
**Confidence:** 3

**Summary:**

UPDATE: I thank the authors for their reply, and am happy to see that the case I described was in fact covered by the theory. I have raised my score accordingly.

This paper considers the problem of making counterfactual predictions, in a multi-domain setting of latent causal models: In each domain, the latent variables form an SCM, the observed variables are computed from the latent ones by a deterministic function that is shared by all domains, and the counterfactual query asks, given observed data from one domain, what that data would have been had it been generated in another domain.

**Strengths:**

The paper is clearly structured, guiding the reader through the theoretical setup.

Compared to existing similar work, this paper provides a significant and novel contribution, both in terms of theoretical results and their application in an algorithm.

**Weaknesses:**

I have a concern that there might be an unstated assumption; see my main question below.

The text could benefit from more proofreading.

**Questions:**

One thing that is unclear to me about the problem setting is the following. Suppose $g = Id$, for some domains $f_d = Id$, and for others $f_d = -Id$. Then in all domains, the observed variables are independently distributed as standard normals. (The same would be true for variations where e.g. each domain flips the signs of a subset of the variables.) For a given counterfactual query, how can your method know whether to predict according to $x_{d'} = x_d$, or according to $x_{d'} = -x_d$? There is no signal in the available data to determine this. Is there an (implicit?) assumption somewhere to rule out cases like this?

Other comments / questions:

* Abstract, "all non-intervened variables have non-intervened ancestors": I suggest "... have no intervened ancestors". The current sentence can be interpreted as "have some intervened ancestors".

* Paragraph before C1-4, "Given our assumption ... distribution equivalence": This is apparently without assumptions (1) and (2), which were stated in the preceding sentence. Please make clearer in the text that you're now considering assuming *only* (3).

* In corollary 3, "Prop. 8" should refer to definition 8.

* Section 3.3: "establish on" should be something else. These sentences should emphasize that you'll now be assuming continuity of $f$ and $g$ (so remove "the" and rewrite). In corollary 5, no requirement of continuity is currently stated, but I think it is needed. Also, just above the corollary, what do you mean with "and not ill-defined"?

---

> ### Author Response · Authors · 2023-11-17
> **Response to Reviewer 4MQs**
>
> We appreciate the insightful feedback provided by the reviewer. We responded to the questions and suggestions you made individually below, with pointers to locations in the manuscript where the corresponding changes have been made (highlighted in blue).
>
> > Suppose $g = Id$, for some domains $f_d = Id$ and for others $f_d = -Id$. For a given counterfactual query, how can your method know whether to predict according to $x_{d’}$ or according to $x_{d’}=-x_d$? Is there an (implicit?) assumption somewhere to rule out cases like this?
>
> Thank you for this insightful question. Our work does not rely on unstated assumptions, and your example pinpoints a crucial aspect of our theoretical framework. Our paper's theoretical contribution lies in characterizing the relationship between counterfactually and distributionally equivalent Invertible Latent Domain (ILD) models. We demonstrate that all counterfactually and distributionally equivalent ILDs share a common feature: the size of the intervention set. In the scenario you described, it is indeed impossible to infer the counterfactual query directly from the interventional data. Based on our distribution equivalence definition (Definition 2), in this case, the intervention set size is $m$ because all the variables are intervened on. By fitting the domain distributions, the counterfactual error (defined in the new Section 3.4 of our revised paper) is bounded by a distribution fit term and a worst-case error. In your example, because the intervention set size is $m$, we will incur a worst-case counterfactual error of $O(\sqrt{m})$. However, if we assume the sparse mechanism shift (SMS) hypothesis about the true model, then we can reduce this worst-case counterfactual error to $O(\sqrt{k})$ because only $k$ dimensions can be the negative of the original.
>
> We expect that future work will be able to more fully explore and analyze counterfactual risk to improve domain counterfactual estimation building upon our results in this paper.
>
>
>
> > [In the] abstract, "all non-intervened variables have non-intervened ancestors": I suggest "... have no intervened ancestors".
>
> Thanks for the suggestion. We rewrote this part in the revised manuscript marked in blue.
>
> > "Given our assumption ... distribution equivalence": This is apparently without assumptions (1) and (2), which were stated in the preceding sentence. Please make clearer in the text that you're now considering assuming only (3).
>
> This is based on Assumption 3. We have revised the paper accordingly.
>
> > In corollary 3, "Prop. 8" should refer to definition 8.
>
> We have revised the paper accordingly.
>
> > Section 3.3: "establish on" should be something else…In corollary 5, no requirement of continuity is currently stated, but I think it is needed.
>
> We have edited these sections to be more clear (e.g., we added a constraint that $(g, f_d)$ for all $d$ are continuous in Corollary 5).

---

> ### Author Response · Authors · 2023-11-21
> **Follow-up**
>
> Hi Reviewer 4MQs, thank you for your helpful feedback. Have we adequately addressed your questions and concerns?

---

### Official Review · Reviewer_ji9w · 2023-11-01

**Soundness:** 3 good
**Presentation:** 3 good
**Contribution:** 3 good
**Rating:** 6
**Confidence:** 3

**Summary:**

The paper focuses on the problem of domain counterfactuals in the context of latent causal models and proposes a practical yet theoretically grounded approach to address this problem, aiming to improve the estimation of domain counterfactuals while making minimal assumptions about the true model and available data.  Experiments on extensive simulated and image-based data show the advantages of domain counterfactual estimation both theoretically and practically.

**Strengths:**

The problem setting, which revolves around domain counterfactual estimation, is a novel and highly intriguing area of research. I believe it has the potential to make a valuable contribution to the research community.

---------

My main concerns are as follows:

1) I am experiencing confusion regarding domain counterfactual equivalence. While I agree that it may not be necessary to fully identify the true latent causal model for domain counterfactual estimation, I would appreciate a more intuitive explanation of Theorem 1 and its implications. For instance, it would be helpful to know which specific aspects of the true latent causal model need to be identified to enable domain counterfactual estimation. This could include considerations like the identifiability of latent noise variables, the size of the intervention set, or any other relevant factors.

2) How should we precisely define the size of the intervention set, denoted as 'k'? In discussions about interventions or the number of variables subject to intervention, there is usually a reference to a latent causal model where no interventions have occurred. How do we accurately establish this reference latent causal model?

3) One of the main contributions, in my view, pertains to the definition of the canonical domain counterfactual model. However, this definition might appear somewhat stringent, particularly with its requirement that only the last variables be intervened. Even though it's surprising that any Invertible Latent Domain (ILD) model can be transformed into an equivalent canonical ILD, could you provide some real-world applications to illustrate and justify the relevance and utility of the canonical domain counterfactual model?

4) It appears that in order to enhance domain counterfactual estimation, one needs to have prior knowledge of the intervention sparsity, denoted as 'k.' However, in practical scenarios, obtaining this information can be challenging. While the experiments do offer some insights and analysis regarding the mismatch of sparsity between generative and inference models, could you provide further justification or reasoning for the selection of the appropriate value for 'k'?


I would be willing to increase my rating if these concerns are addressed.

**Weaknesses:**

See above

**Questions:**

See above

---

> ### Author Response · Authors · 2023-11-17
> **Response to Reviewer ji9w (1/2)**
>
> We thank the reviewer for the helpful feedback. We responded to the questions and suggestions you made individually below, with pointers to locations in the manuscript where the corresponding changes have been made (highlighted in blue).
>
> >“I would appreciate a more intuitive explanation of Theorem 1 and its implications.”
>
> Thanks for the question. The significance of theorem 1 is to formally characterize the set of model’s which are counterfactually equivalent to each other. Intuitively, this means that any model within this set produces *exactly the same counterfactual output*  (i.e. $x_{d_1 \rightarrow d_2}$), *despite* different underlying causal mechanisms, such as variations in recovered exogenous noise, observation functions, or implied distributions (e.g., it is possible to be counterfactually equivalent even though not distributionally equivalent, see a section discussing the relationship between distribution equivalence and counterfactual equivalence in the Response to All Reviewers.).
>
> Key implications of Theorem 1 include:
>
> 1. This theorem reveals that, surprisingly, despite the true ILD, vastly different ILDs could generate equivalent counterfactuals even if they might not be distributionally equivalent.
> 2. This theorem proves that recovering the original causal model, which is a much more challenging task, is unnecessary for estimating domain counterfactuals.
> 3. It provides a framework to test whether two Invariant Latent Domain (ILD) models are counterfactual equivalents. It also facilitates the development of new counterfactual equivalent ILDs. Utilizing this construction ability, the proof of Theorem 2, Proposition 3, Theorem 4 and Proposition 5 all directly use this theorem.
>
>
> > “How should we precisely define the size of the intervention set, denoted as 'k'? …there is usually a reference to a latent causal model where no interventions have occurred. How do we accurately establish this reference latent causal model?”
>
> Thank you for your insightful question. To clarify, we categorize a node as 'intervened' when there's a variation in its causal mechanism across different domains in our ILD model. This determination can be made using any domain distribution as a reference point if the intervention is soft [1]. For simplicity and without any loss of generality, we often assume domain 1 as this reference. In our revised manuscript, this concept is elaborated in Equation 9 under Definition 9, where the second equality reflects this reference-based perspective.
>
> Alternatively, we can assess intervention by comparing pairwise causal mechanisms. This means that a latent node $j$ is considered intervened if the functions $\widetilde{f}\_{d}^{(j)}$ and $\widetilde{f}_{d’}^{(j)}$ differ for any domain pair $d,d’$. This approach is represented by the first equality in Equation 9.
>
> [1] Kocaoglu, Murat, et al. "Characterization and learning of causal graphs with latent variables from soft interventions." Advances in Neural Information Processing Systems 32 (2019).

---

> > ### Author Response · Authors · 2023-11-17
> > **Response to Reviewer ji9w (2/2)**
> >
> > > “One of the main contributions, in my view, pertains to the definition of the canonical domain counterfactual mode…could you provide some real-world applications to illustrate and justify the relevance and utility of the canonical domain counterfactual model?”
> >
> > We appreciate your acknowledgment of the importance of our canonical domain counterfactual model. Before addressing your query, we'd like to clarify a key aspect of the canonical ILD model: the primary objective of our research is not to identify the exact ground truth ILD model. Instead, our aim is to develop a more efficient and reliable method for estimating a model that is counterfactually equivalent in real-world scenarios. With this, the practical significance of the canonical model lies in its application to scenarios where we need to generate domain counterfactuals given only interventional data. By adopting the canonical model approach, we can avoid the exhaustive search across the entire model space. For example, if we consider the size of the interventional set to be $k$, the canonical form allows us to presume that the last $k$ nodes in the set are the intervention points. Without the canonical form, a naive method would have to search over the $k \choose m$ sparsity structures of our latent SCM to find the right sparsity structure (even if we know $k$) or resort to more complex sparse optimization techniques, like the L1 norm relaxation of the L0 penalty. Therefore, the canonical form makes practical optimization significantly more efficient and manageable.
> >
> > > It appears that in order to enhance domain counterfactual estimation, one needs to have prior knowledge of the intervention sparsity, denoted as 'k.' …could you provide further justification or reasoning for the selection of the appropriate value for 'k'?
> >
> > Thank you for your question. In our response to all reviewers, we show that, as with many hyperparameters, there is a natural bias-variance tradeoff in choosing $k$. If $k > k^*$, then the space of distributionally equivalent models is very large and thus the variance in estimation will be high. If $k < k^*$, then the model will be biased, i.e., there does not exist a model that is distributionally and counterfactually equivalent to the true ILD. Using our new bound on the expected counterfactual error, we also suggest multiple methods for choosing $k$ in practice, with the least assuming option being to start with a small $k$ and then increase $k$ until distributional equivalence is achieved (i.e., the data fit term is optimal, which minimizes the first term of the counterfactual error bound). Empirically, we noticed that in our current simulated experiments (Figure 15) and additional simulated experiments (Figure 34), there is a clear gap if the choice of $k$ is smaller than the ground truth $k^*$.

---

> ### Author Response · Authors · 2023-11-21
> **Follow-up**
>
> Hi Reviewer ji9w, thank you for your helpful feedback. Have we adequately addressed your questions and concerns?

---

> > ### Comment · Reviewer_ji9w · 2023-11-22
> > **Comments**
> >
> > Thanks for your replies.
> >
> > While I acknowledge that it may not be essential to fully identify the true latent causal model for domain counterfactual estimation, I remain uncertain about which specific aspects of the true latent causal model need to be identified to facilitate domain counterfactual estimation. Could you please provide further clarification?

---

> ### Author Response · Authors · 2023-11-22
> **Response to Comment by Reviewer ji8w**
>
> Thank you for your response. Succinctly, we need to know $k^*$ for better counterfactual estimation if we want to achieve counterfactual equivalence by fitting the observational data distribution.
>
> Theorem 4 suggests that any distributional and counterfactual equivalent models share the same intervention set size $k$. To aid in selecting $k$ in practice, we have introduced Theorem 6 which shows that misspecifying of $k$ will lead to a bias-variance tradeoff scenario, where if the estimated $k$ is larger than $k^*$, there will be a large variance on the estimation. However, if the estimated $k$ is smaller than $k^*$, you will not be able to find a distributionally equivalent model, which leads to a biased estimation. For a detailed explanation regarding the impact of misspecifying $k$ and how to choose it in practice, we refer the reviewer to the **Choosing intervention set size $k$ and new bound** paragraph in our response to all reviewers. Of all of our assumptions, the sparse mechanism shift assumption plays an important role here as it, together with our new Theorem 6, guides us to choose a relatively smaller $k$ unless we fail to achieve distribution equivalence, which is justified by our empirical study.

---

> > ### Comment · Reviewer_ji9w · 2023-11-23
> >
> > Thank you for further response.
> >
> > I have a better understanding of this part now.  I have increased my score accordingly and will engage in discussion with reviewers further.

---

### Author Response · Authors · 2023-11-17
**Response to all reviewers**

We are grateful to all reviewers for their excellent and detailed feedback. We would like to summarize the major changes in the revised manuscript and address the common concern here.

**Reorganization of Section 2**

Following the suggestion of Reviewer KSip, we have simplified Section 2. The new Section 2 only contains the definitions and propositions that are needed in Section 3. The original Section 2 is moved to Section B of the revised manuscript for a comparison, which contains expanded explanations and proofs.

**Additional Simulated Experiments**

To address Reviewer KSip’s concern about the simplicity of the observation function in our simulated experiment, we added a few additional experiments. First, we replace the current ground truth observation function $g^*$, composed of leaky ReLU and a linear transformation, with a complicated normalizing flow model based on the RealNVP model. Then, for our model, we test the use of both normalizing flows and VAE-based models (with varying levels of model misspecification). More details and relevant figures can be found in Appendix J.1. In summary, we observe a similar trend as that in Figure 1: no matter whether the structures of $g^*$ and $g$ match, using the ILD-Relax-Can model results in lower counterfactual error than the ILD-Relax-Can model.

**Choosing intervention set size $k$ and new bound**

Our Corollary 5 shows that for any underlying ground truth ILD, all distributionally and counterfactually equivalent ILDs must share the same intervention set size. A natural question arises: How should we choose $k$ in practice (without making unrealistic assumptions)? In section 3.4, we introduce a new counterfactual error bound (Theorem 6), which exposes how the expected counterfactual distance between a model ILD $(g, \mathcal{F})$ and the ground truth ILD $(g^*, \mathcal{F}^*)$ can be decomposed into a term corresponding to distributional equivalence and a term corresponding to counterfactual equivalence. With this, we show that as with many hyperparameters, there is a natural bias-variance tradeoff in choosing $k$. If $k > k^*$, then the space of distributionally equivalent models is very large and thus the variance in estimation will be high. If $k < k^*$, then the model will be biased, i.e., there does not exist a model that is distributionally and counterfactually equivalent to the true ILD. However, this may be better if the variance is reduced significantly.

To choose $k$ in practice, we suggest a few practical approaches. Ideally, if a small validation set of counterfactual pairs is available, $k$ can be tuned to minimize the expected counterfactual error (defined in Definition 6). If counterfactual pairs are unavailable, $k$ could be chosen by inspecting the estimated counterfactuals for different $k$ values and choosing the smallest $k$ that yields reasonable counterfactuals (which would require domain knowledge). If counterfactuals are being used for some downstream application such as domain generalization, the $k$ that minimizes the validation loss could be selected. Finally, if none of these approaches are applicable, we would suggest starting with a very small $k$ (even $k=1$) and increasing $k$ until distributional equivalence is achieved (i.e., the data fit term is optimal, which minimizes the first term of the counterfactual error bound). Empirically, we noticed that in our current simulated experiments (Figure 15) and additional simulated experiments (Figure 34), there is a clear gap if the choice of $k$ is smaller than the ground truth $k^*$.

---

### Meta-Review · Area_Chair_t9qN · 2023-12-08

**Metareview:**

Counterfactual identification and its use in data augmentation and other data-analytic areas is a relevant area of research within ICLR. In the paper, some interesting advances are presented by linking a special class of structural causal models to counterfactual data generators exemplified by applications in image generation.

I think the paper would benefit from a discussion on the limitations of the invertible model. Although it is well-known that invertible models can represent any density function for continuous multivariate distributions, it is still not true they can represent all counterfactual distributions - the assumption that the error term is one-dimensional is very restrictive regardless of whether or not we use them additively. For instance, if $X$ is binary and $Y$ is continuous, and we have that $Y$ depends on a two-dimensional exogenous error $(U_{y_1}, U_{y_2})$ with entries which are mutually independent, and the structural equation is $Y = I(X = 0) U_{y_1} + I(X = 1)U_{y_2}$, then the counterfactual $Y_1$ is independent of $Y_0$ even under the event $X = x$ - something that the ILD cannot capture. So while an invertible model is a welcome advance over additive error models, it is still very, very restrictive (in the SCM framework, error terms are potentially infinite dimensional. When a structural equation is additive, we can collapse all error contributions into a single scalar. When parents are all discrete, we can use a finite-dimensional potential outcome process. But the moment we claim we don't want to rely on additivity or discrete parents, then any restrictions in dimensionality is a simplification that needs to be appraised).

The paper should provide such a critical discussion so that readers manage their expectations properly. Proposition 1, while not wrong in a strict sense, can be misinterpreted.

As a minor comment: the current manuscript uses the notation $F(X = x)$, which is not standard and very confusing. Use either $F(x)$ or $P(X \leq x)$, but "$F(X = x)$" does not make sense.

**Justification For Why Not Higher Score:**

This is a good advance, but much more could be done regarding a discussion of the limitations of the approach.

**Justification For Why Not Lower Score:**

The problem is hard, but of wide interest. The paper makes some interesting steps to make it more realistic.

---

### Decision · Program_Chairs · 2024-01-16

Accept (poster)